# Sox2 levels regulate the chromatin occupancy of WNT mediators in epiblast progenitors responsible for vertebrate body formation

Robert Blassberg [1], Harshil Patel [1], Thomas Watson[1], Mina Gouti[2], Vicki Metzis[1,3], M. Joaquina Delás [1] and James Briscoe [1]✉

**WNT signalling has multiple roles. It maintains pluripotency of embryonic stem cells, assigns posterior identity in the epiblast and induces mesodermal tissue. Here we provide evidence that these distinct functions are conducted by the transcription factor SOX2, which adopts different modes of chromatin interaction and regulatory element selection depending on its level of expression. At high levels, SOX2 displaces nucleosomes from regulatory elements with high-affinity SOX2 binding sites, recruiting the WNT effector TCF/β-catenin and maintaining pluripotent gene expression. Reducing SOX2 levels destabilizes pluripotency and reconfigures SOX2/TCF/β-catenin occupancy to caudal epiblast expressed genes. These contain low-affinity SOX2 sites and are co-occupied by T/Bra and CDX. The loss of SOX2 allows WNT-induced mesodermal differentiation. These findings define a role for Sox2 levels in dictating the chromatin occupancy of TCF/β-catenin and reveal how context-specific responses to a signal are configured by the level of a transcription factor.**

Producing the variety of cell types that compose a multicellular organism requires the spatial and temporal regulation of gene expression, controlled by extrinsic signals. But there are relatively few signals compared with the number of cell types, and these signals are re-used over the course of ontogeny. The molecular mechanisms responsible for context-dependent responses to signals that generate cellular diversity remain incompletely understood.

WNT signalling, through its transcriptional effector TCF/β-catenin, has multiple functions[1,2]. In mouse embryonic stem cells (mESCs), WNT/β-catenin signalling promotes pluripotency[3,4]. Later, it upregulates CDX transcription factors (TFs) and assigns posterior identity in the forming caudal epiblast[5]. A subset of the cells within the caudal lateral epiblast (CLE) are neuromesodermal progenitors (NMPs) that generate the neural and mesodermal tissue responsible for the elongation of the axis[6]. In NMPs, WNT/β-catenin signalling promotes differentiation to mesodermal tissue at the expense of spinal cord neural differentiation[7–13].

Alongside WNT signalling, the TF SOX2 also plays a central role. SOX2 is expressed at high levels in mESCs and epiblast cells where it maintains the undifferentiated pluripotent state[14,15]. Then, as the embryo regionalizes, SOX2 expression drops in the CLE and remains expressed at low levels in NMPs together with genes conferring primitive streak identity such as T/BRA[16–18]. Upregulation of SOX2 is associated with the allocation of NMPs to neural progenitors[9,19]. By contrast, SOX2 expression is lost upon commitment of NMPs to mesodermal lineages[17].

The correlation between SOX2 levels and the changes in the function of WNT signalling raises the possibility that SOX2 influences the response of cells to WNT signalling. In mESCs, SOX2 acts with β-catenin and the TFs TCF7L1, OCT4 and NANOG to promote the expression of WNT target genes that maintain pluripotency[20–22]. By contrast, SOX2, T/BRA and β-catenin co-occupy

a large number of mesodermal *cis*-regulatory elements (CREs) in NMPs[12]. This suggested that SOX2 contributes to maintaining the undifferentiated state of CLE progenitors by directly counteracting WNT signalling activity and inhibiting mesoderm gene expression. Nevertheless, direct evidence of whether and how SOX2 is responsible for the different developmental responses to WNT signalling has been lacking.

In this Article, to test the causal role of SOX2 in the response of cells to WNT signalling, we decoupled SOX2 expression from its developmental regulation. This revealed that SOX2 controls the WNT response by adopting different modes of chromatin interaction and occupying distinct genomic locations depending on its level of expression. Together the results provide insight into the mechanisms that determine the context-specific response of cells to an extrinsic signal that generates the diversity of outcomes necessary for tissue development.

## Results

**SOX2 levels alter the response of pluripotent cells.** We took advantage of an in vitro model of the caudal epiblast. mESCs differentiated for 48 h in the presence of FGF and LGK974, an inhibitor of WNT secretion (henceforth 'FL medium'), acquire an epiblast-like cell (EpiLC) identity (Fig. 1a), recapitulating post-implantation epiblast gene expression changes (Extended Data Fig. 1a). Similar to their in vivo counterparts, EpiLCs acquired caudal epiblast-like cell (CEpiLC) identity in response to WNT signalling, activated by 24 h exposure to the GSK3 inhibitor CHIR99021 (henceforth 'FLC medium') (Fig. 1a). This resulted in reduced expression of SOX2, upregulation of the primitive streak marker T/BRA (Extended Data Fig. 1b) and expression of *Cdx* and posterior *Hox* genes (Extended Data Fig. 1c,d)[9].

We ablated endogenous *Sox2* and introduced a *Sox2* transgene under the control of doxycycline (Dox) (Fig. 1b). Addition of Dox to

[1]The Francis Crick Institute, London, UK. [2]Stem Cell Modelling of Development & Disease Group, Max Delbrück Center for Molecular Medicine, Berlin, Germany. [3]Present address: Institute of Clinical Sciences, Imperial College London, London, UK. ✉e-mail: james.briscoe@crick.ac.uk

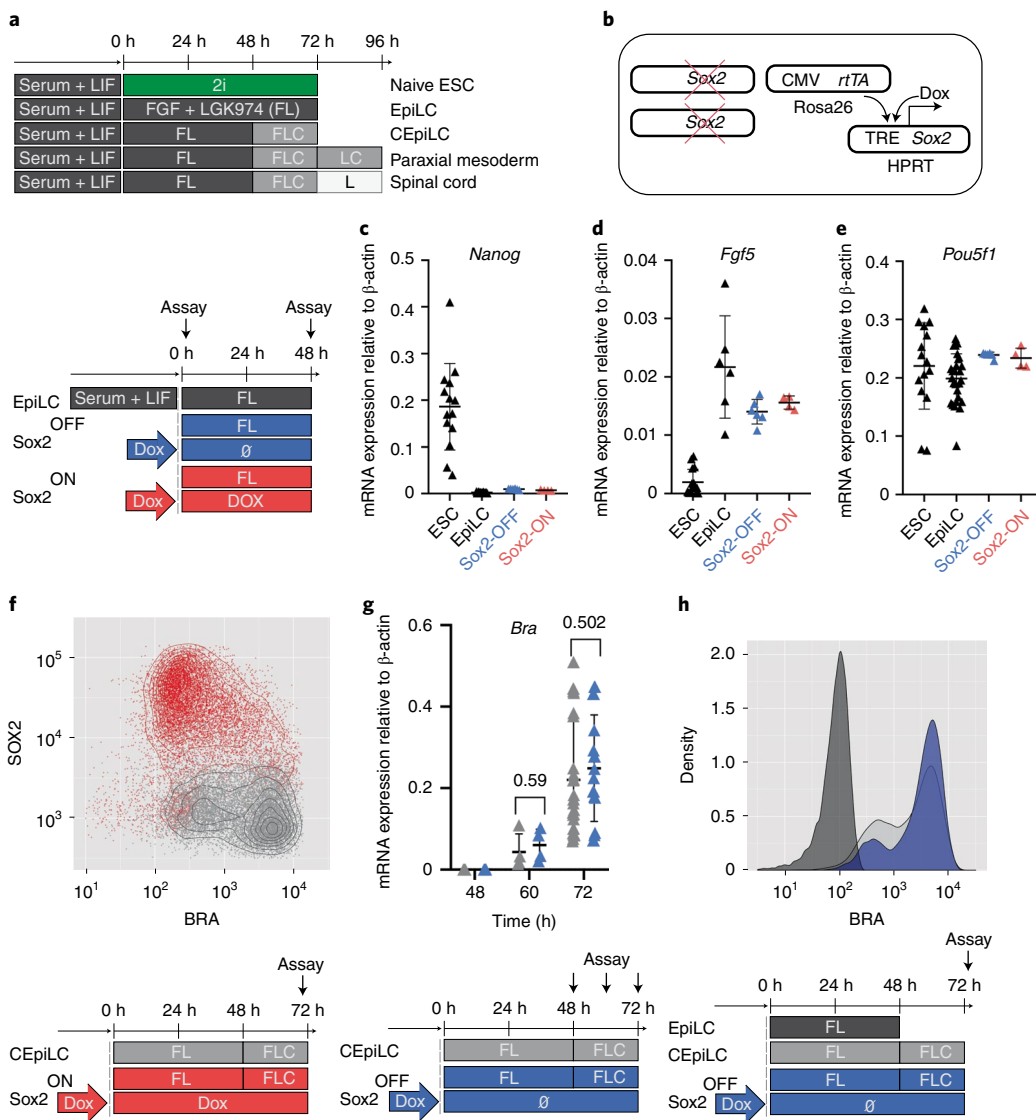

**Fig. 1 | High SOX2 levels inhibit WNT-induced differentiation of epiblast-like cells. a**, Schematic of mESC differentiation. F, FGF; L, LGK974; C, CHIR99021; 2i, CHIR99021 + PD0325901. **b**, To generate SOX2-TetON cells, a Dox-inducible SOX2 transgene was incorporated into the HPRT locus of an mESC line constitutively expressing the rtTA gene from the Rosa26 locus before ablation of endogenous SOX2 expression by gene editing. **c–e**, SOX2-ON and SOX2-OFF cultured in FL medium downregulate Nanog (**c**), upregulate FGF5 (**d**) and maintain Pou5f1 expression (**e**). **f**, SOX2 levels are negatively correlated with BRA expression in SOX2-ON cultured in FLC medium. **g**, Measurement of relative mRNA expression by PCR with reverse transcription (RT–PCR) shows that the kinetics of T/Bra induction is unaltered in SOX2-OFF cultured in FLC medium compared with CEpiLC controls. Each datapoint represents an individual biological replicate. Bars denote mean ± standard error of the mean (s.e.m.). *P* values calculated for differences of mean expression by two-tailed Students *t*-test are shown. *n* = 4 for both CEpiLC and SOX2-OFF at 60 h. *n* = 30 for CEpiLC and *n* = 18 for SOX2-OFF at 72 h. **h**, Representative plot from flow cytometry analysis shows that the distribution of T/BRA levels is unaltered in SOX2-OFF cultured in FLC medium compared with CEpiLC controls (arbitrary units). Source numerical data are available in source data.

these SOX2[TetON] cells generated levels of SOX2 expression similar to wild type (WT) (Extended Data Fig. 2a), and these cells (henceforth SOX2-ON) could be propagated in naive pluripotent '2i' medium[4] (Extended Data Fig. 2b). Removal of Dox from SOX2[TetON] (henceforth SOX2-OFF) resulted in a gradual decrease in SOX2 protein levels (Extended Data Fig. 2c), flattening of colonies (Extended Data Fig. 2d), loss of expression of pluripotency markers OCT4 and NANOG (Extended Data Fig. 2e) and the induction of T/BRA (Extended Data Fig. 2f).

Differentiation of ESCs to CEpiLC identity involves exiting naive pluripotency and transitioning to EpiLC identity before activation of WNT signalling[9] (Fig. 1a). Both SOX2-OFF and SOX2-ON

differentiated in FL medium maintained the gene expression changes characteristic of the transition to EpiLC identity: downregulating *Nanog* and upregulating *Fgf5* while continuing to express *Pou5f1* (Fig. 1c–e). This was due to *Sox3* upregulation in SOX2-OFF (Extended Data Fig. 2g–i), consistent with SOX3 being sufficient to maintain EpiLCs in the absence of SOX2 (ref. [23]). Upon transfer to FLC medium, only limited T/BRA induction was observed in SOX2-ON cells (Fig. 1f), consistent with SOX2 acting as a repressor of primitive streak identity[11,12,24,25]. By contrast, SOX2-OFF cells in FLC medium, which express low levels of both SOX2 and *Sox3* (Extended Data Fig. 2j,k), induced T/BRA to similar levels as WT CEpiLC (Fig. 1g,h).

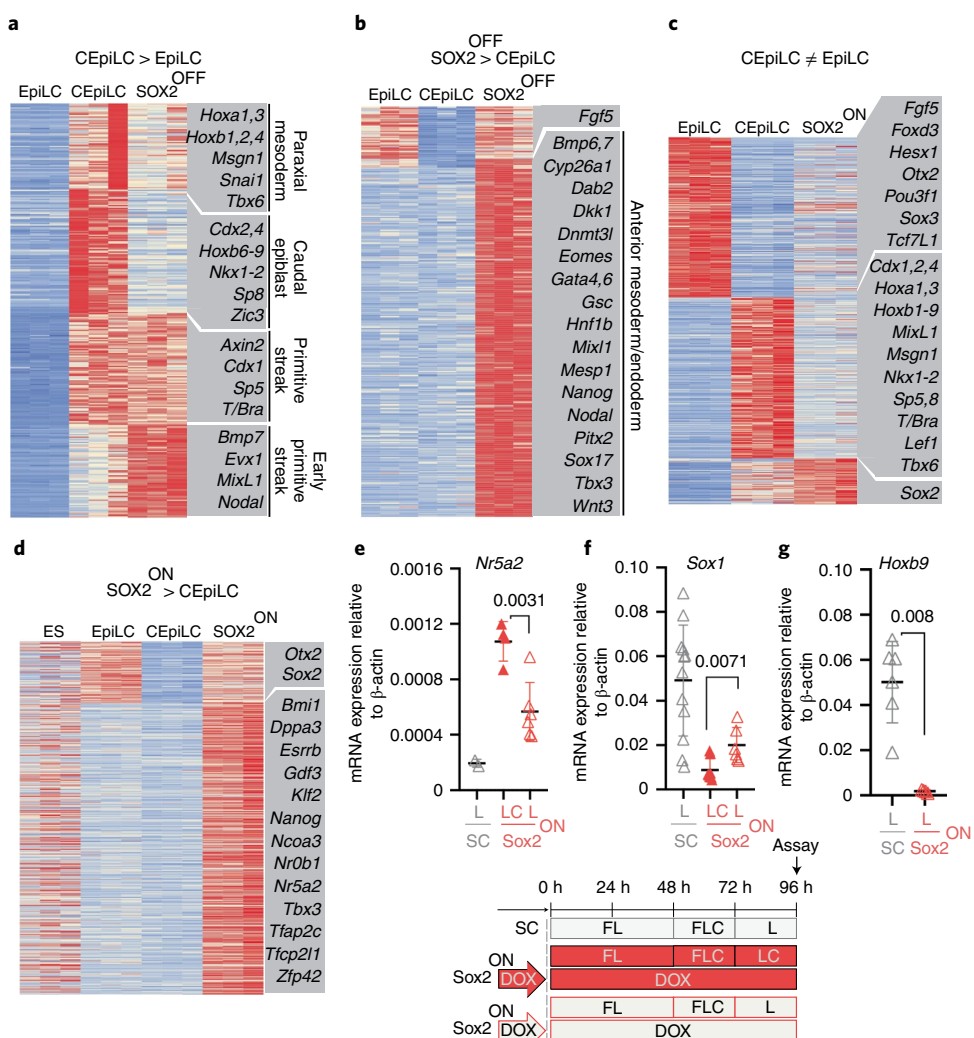

**Fig. 2 | SOX2 dynamics configure the WNT response of epiblast-like cells. a**, Sox2-OFF cultured in FLC medium exhibit altered expression of CEpiLC-specific genes. Genes shown are upregulated in CEpiLC compared with EpiLC (log₂ fold change (log₂FC) >1, false discovery rate (FDR) <0.05). FDR was determined by DESeq2 padj metric from n=3 biological replicates. k-Means clustering (k=3). Illustrative genes for each cluster are highlighted. **b**, Genes upregulated in SOX2-OFF compared with CEpiLCs include early/anterior streak and mesendoderm markers. Genes shown are differentially expressed (log₂FC >1, FDR <0.05) in a comparison between SOX2-OFF and CEpiLC. FDR was determined by DESeq2 padj metric from n=3 biological replicates. k-Means clustering (k=2). Illustrative genes for each cluster are highlighted. **c**, SOX2-ON cultured in FLC medium express low levels of EpiLC- and CEpiLC-specific genes. Genes shown are differentially expressed (mod(log₂FC) >1, FDR <0.05) in a comparison between EpiLCs and CEpiLC. FDR was determined by DESeq2 padj metric from n=3 biological replicates. k-Means clustering (k=3). Illustrative genes for each cluster are highlighted. **d**, SOX2-ON express elevated levels of pluripotency-associated genes compared with CEpiLCs and EpiLCs. Genes shown are differentially expressed (log₂FC >1, FDR <0.05) in a comparison between SOX2-ON and CEpiLCs. FDR was determined by DESeq2 padj metric from n=3 biological replicates. k-Means clustering (k=3). Illustrative genes for each cluster are highlighted. **e–g**, Measurement of relative mRNA expression by RT–qPCR shows that expression of the pluripotency factor Nr5a2 is reduced (**e**), the neural marker Sox1 increases (**f**) and the posterior marker Hoxb9 is not expressed (**g**) when SOX2-ON are transferred from FLC to FL medium. SC, spinal cord. Each datapoint represents an individual biological replicate. Bars denote mean ± s.e.m. P values calculated for differences of mean expression by two-tailed Student's t-test are shown. In **e**, n=6 'LC' and n=4 'L' samples; in **f**, n=6 'LC' and n=8 'L' samples; and in **g**, n=6 'LC' and n=4 'L' samples. mod, absolute value. Source numerical data are available in source data.

Both WT and SOX2-OFF cells differentiated for 24 h in FLC medium induced genes characteristic of the primitive streak, yet a set of genes associated with the caudal epiblast were not upregulated in SOX2-OFF cells (Fig. 2a). These included the caudal epiblast determinants Cdx2 and Cdx4 and posterior Hox genes. Anterior Hox and paraxial mesoderm marker expression was reduced (Fig. 2a and Extended Data Fig. 3a,b) and genes characteristic of earlier more anterior mesoderm and endoderm were increased in SOX2-OFF cells (Fig. 2a,b and Extended Data Fig. 3c). Thus, loss of SOX2 disrupts the induction of the caudal

epiblast gene expression programme in response to WNT signalling and instead leads to the differentiation of an earlier, more anterior primitive streak identity.

We next determined the identity of high-SOX2-expressing SOX2-ON cells cultured in FLC medium. As predicted from the reduction of T/BRA expression (Fig. 1f), paraxial mesoderm differentiation was inhibited (Extended Data Fig. 3d). Moreover, the majority of WNT-induced genes associated with CEpiLC identity were repressed in SOX2-ON cells (Fig. 2c). SOX2-ON cells in FLC medium did not differentiate to neural identity (Extended Data

Fig. 3e), and instead re-expressed genes associated with naïve pluripotency (Fig. 2d and Extended Data Fig. 3f). Thus, sustaining high levels of SOX2 in the presence of WNT signalling appeared to revert cells to a naive-like pluripotent state.

We reasoned that removing the WNT agonist from SOX2-ON cells should destabilize the pluripotent state and permit differentiation to neural identity. Consistent with this, a decline in pluripotency marker expression was accompanied by the onset of neural differentiation following WNT-agonist withdrawal from SOX2-ON cells (Fig. 2e,f). Importantly, posterior *Hox* genes, typical of spinal cord neural progenitors, were not induced (Fig. 2g). Taken together, these data indicate that a reduction of SOX2 levels is necessary to prevent cells from initiating a pluripotent WNT response, whereas premature elimination of SOX2 abrogates the ability of WNT signalling to promote caudal identity. This raises the question of how SOX2 alters the response of epiblast progenitors to WNT signalling.

**SOX2 downregulation reconfigures β-catenin occupancy.** SOX2 is found with WNT signal transducers at a large number of CREs in both CEpiLCs[12] and naive pluripotent ESCs[20–22,26]. To determine whether SOX2 levels configure distinct transcriptional responses to WNT signalling by altering the binding profile of β-catenin, we performed chromatin immunoprecipitation followed by sequencing (ChIP–seq) for SOX2 and β-catenin from naive ESCs cultured in 2i and CEpiLCs, SOX2-ON and SOX2-OFF cells cultured in FLC medium (Fig. 3a). Consensus peaks included 76% of SOX2 and 89% of β-catenin peaks independently identified in WT CEpilCS[12], plus an additional 110,893 SOX2 and 81,832 β-catenin peaks (Extended Data Fig. 4a,b).

Differential analysis between naive pluripotent ESCs and CEpiLCs revealed a dynamic pattern of SOX2 binding (Fig. 3b). SOX2 occupancy was reduced at 5,943 sites in CEpiLCs, but increased at 5,754 locations, despite its lower expression levels. Higher SOX2 occupancy in naive ESCs included peaks associated with pluripotency genes, whereas peaks exhibiting higher SOX2 occupancy in CEpiLCs were associated with primitive streak and trunk identity genes (Fig. 3b). Strikingly, β-catenin exhibited a coordinated reconfiguration in its occupancy at sites differentially occupied by SOX2 (Fig. 3c). The majority of peaks differentially occupied by β-catenin reflected the altered SOX2 occupancy at these sites in CEpiLCs compared with naive ESCs (5,851/5,908; 99%) (Extended Data Fig. 4c). This indicated that the changes in SOX2 levels accompanying the transition from pluripotency to CEpiLC might reconfigure the transcriptional response to WNT signalling by redistributing β-catenin occupancy.

To test whether changes in SOX2 levels account for the reconfiguration of SOX2/β-catenin occupancy during the transition from pluripotency to CEpiLC, we assayed the effect of manipulating SOX2 levels in SOX2[TetON] cells cultured under CEpiLC differentiation conditions. Of the 9,727 β-catenin peaks differentially occupied between CEpiLC, SOX2-ON and SOX2-OFF, 4,399 (45%) overlapped with peaks differentially occupied by SOX2 (Fig. 3d). By contrast, differentially occupied β-catenin peaks showed a markedly lower association with chromatin-associated factors identified by ENCODE (1.6–9.5% overlap) (Extended Data Fig. 4d). Moreover, of the overlapping SOX2 and β-catenin peaks differentially occupied in response to experimental manipulation of SOX2 levels, 1,976 (45%) of the SOX2 and 2,355 (53%) of the β-catenin peaks were also differentially occupied between CEpiLC and naive ESCs. Taken together, these observations suggest that the redistribution of β-catenin that occurs during the transition from pluripotent to CEpiLC identity might be mechanistically related to differential SOX2 occupancy in high- and low-SOX2-expressing cells.

We further investigated the relationship between SOX2 and β-catenin by clustering SOX2-occupied CREs on the basis of their cell-type-specific occupancy. As observed during the transition

from pluripotent to CEpiLC identity, changes in β-catenin occupancy mirrored those of SOX2 (Fig. 3e,f and Extended Data Fig. 4e,f). Moreover, both SOX2 and β-catenin occupancy were similar between both SOX2-ON and naive progenitors (which express high SOX2) ($R = 0.75$ SOX2, $R = 0.76$ β-catenin). Likewise, SOX2 and β-catenin occupancy were similar in CEpiLC and SOX2-OFF cells (which express low SOX2) ($R = 0.87$ SOX2, $R = 0.90$ β-catenin). Notably, a group of SOX2/β-catenin bound regions was most highly occupied in SOX2-OFF (cluster 2). The detection of SOX2 in SOX2-OFF conditions is probably due to the perdurance of low levels of SOX2 protein after removal of DOX (Extended Data Fig. 2j). These data therefore provide evidence of a profound and coordinated genome-wide alteration in the binding site occupancy of both SOX2 and β-catenin that is dependent on the level of SOX2 expression.

**SOX2 levels configure TCF/LEF occupancy.** TCF/LEF factors are differentially expressed between high- and low-SOX2-expressing cell types (Fig. 2c). We performed ChIP–seq with TCF7L1, TCF7L2 and LEF1 in ESCs, CEpiLCs, SOX2-OFF and SOX2-ON and found differentially occupied TCF/LEF1 overlapped with differentially occupied SOX2 (Extended Data Fig. 5a–c) and β-catenin sites (Extended Data Fig. 4d–f), suggesting that occupancy occurs at the same CREs. Indeed, TCF/LEF factors exhibited a similar pattern of cell-state-specific occupancy to SOX2/β-catenin (compare Fig. 3e,f with Extended Data Fig. 5g,h,i; Extended Data Fig. 4e,f with Extended Data Fig. 5j–l). These data indicate that the reduction in SOX2 levels during the transition from pluripotency to CEpiLC identity drives the coordinated reconfiguration of SOX2/TCF/β-catenin co-occupancy across the genome.

**TCF/β-catenin occupancy and transcriptional responses.** We asked whether the changes in SOX2/β-catenin binding could explain the distinct gene expression programmes of cells expressing different levels of SOX2. In line with the known positive effect of β-catenin on transcription, differentially expressed genes neighbouring differential SOX2 peaks were, on average, positively correlated with changes in SOX2/β-catenin occupancy for each of the cell-state clusters (Fig. 4).

Genes in cluster 1, which are specifically occupied by SOX2/β-catenin in CEpiLCs (Figs. 4a and 3e,f), are increased in expression in CEpiLCs (Fig. 4a,b). These were enriched for biological processes related to anterior–posterior patterning (Fig. 3f and Extended Data Fig. 6a). Genes in cluster 2 were occupied by SOX2/β-catenin most highly in SOX2-OFF cells, showed greatest expression in these cells (Figs. 4c and 3e,f) and included early primitive streak and mesendodermal genes (Fig. 3f). Cluster 3 genes showed greatest SOX2/β-catenin occupancy and expression in SOX2-ON cells (Figs. 4e and 3e,f) and comprised genes associated with pluripotency and anterior neural identity (Fig. 3f and Extended Data Fig. 6b). Genes in cluster 4, which are occupied by SOX2/β-catenin most highly in WT CEpiLCs and SOX2-OFF cells (Figs. 4g and 3e,f), were enriched for genes expressed in caudal epiblast, primitive streak and mesoderm (Fig. 3f) and showed highest expression in these conditions (Fig. 4h). Thus, differential SOX2/β-catenin occupancy driven by changes in SOX2 levels correlates with cell-state-specific gene expression patterns and points to a positive role for SOX2 in promoting WNT-dependent gene activation by β-catenin.

Differentially expressed genes associated with SOX2/β-catenin occupancy in SOX2-ON in FLC medium and naive progenitors (clusters 3 and 5), included a number of pluripotency factors as well as genes involved in nervous system development (Fig. 3f and Extended Data Fig. 6b). Neural genes were expressed at comparatively low levels (Extended Data Fig. 6c), consistent with the priming of these genes in ESCs[27]. Thus, the establishment of a naive-like

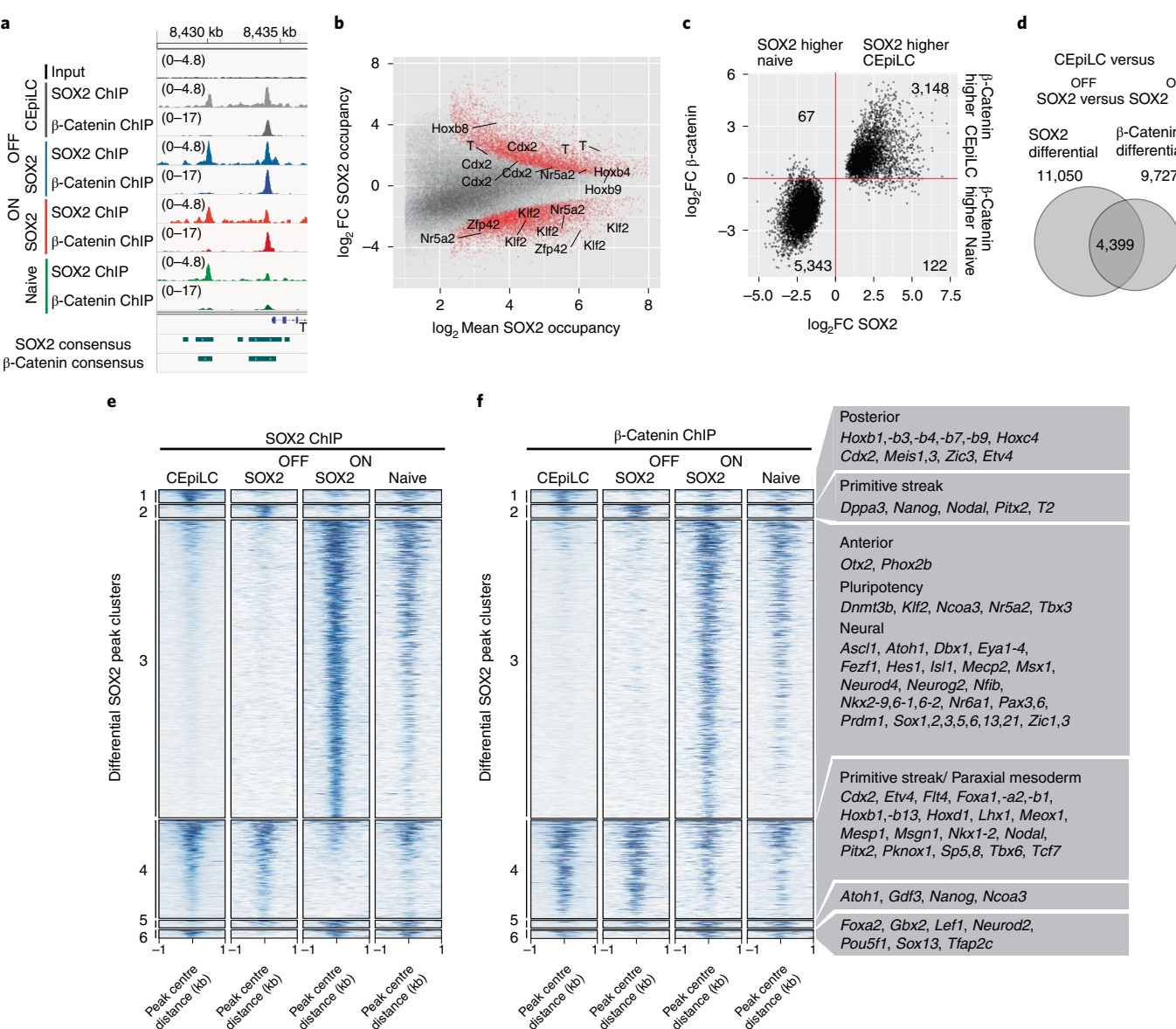

**Fig. 3 | SOX2 downregulation reconfigures β-catenin occupancy. a**, Representative SOX2 and β-catenin ChIP–seq signals used to define high-confidence consensus peaks (Methods). **b**, SOX2 occupancy increases at peaks associated with posterior genes and decreases at peaks associated with pluripotency genes during the differentiation of pluripotent ESCs to CEpiLCs. Differentially occupied peaks (red) are statistically different between conditions (FDR <0.05). FDR was determined by DESeq2 padj metric from $n = 3$ biological replicates. **c**, Peaks differentially occupied by SOX2 between ESCs and CEpiLCs (FDR <0.05) exhibit a correlated increase or reduction of β-catenin occupancy. FDR was determined by DESeq2 padj metric from $n = 3$ biological replicates. $\log_2$FC calculated by DESeq2. **d**, Peaks differentially occupied by β-catenin across any pairwise comparison between CEpiLC, SOX2-OFF and SOX2-ON (FDR <0.05) show a high degree of overlap (4,399/9,727) with differentially occupied SOX2 peaks. FDR was determined by DESeq2 padj metric from $n = 3$ biological replicates. **e,f**, Cell-type-specific SOX2 occupancy (**e**) and β-catenin occupancy (**f**) at SOX2 differential peaks. Representative nearest genes associated with SOX2 peaks from each cluster are shown. For details of clustering, see Methods.

SOX2/TCF/LEF configuration appears to underlie the re-expression of pluripotent factors in SOX2-ON cells stimulated with WNT agonist, and repression of the post-implantation epiblast WNT response. This supports the idea that a reduction in SOX2 levels is necessary to reconfigure SOX2 and β-catenin to ensure a caudal epiblast gene regulatory programme in response to WNT activity.

**High SOX2 levels maintain chromatin accessibility.** SOX2 has been proposed to act as a pioneer factor[28–31], suggesting that SOX2 may direct cell-state-specific TCF/β-catenin binding by altering chromatin accessibility. To test this, we performed assay for transposase-accessible chromatin using sequencing (ATAC–seq).

This revealed distinct relationships between chromatin accessibility and SOX2 occupancy in different cell states. Cluster 3 CREs (pluripotency and neural), occupied by SOX2 in SOX2-ON and pluripotent progenitors, were only accessible in cell states with high SOX2 (Fig. 5a,b and Extended Data Fig. 7a). By contrast, CREs in cluster 1, which were occupied by SOX2 specifically in CEpiLC, were accessible in all cell states (Fig. 5a and Extended Data Fig. 7a). Cluster 2 CREs (SOX2-OFF-specific and early streak) and cluster 4 CREs (primitive streak and paraxial mesoderm) exhibited a more complex pattern of accessibility, with comparable average accessibility in SOX2-OFF, SOX2-ON and pluripotent progenitors, but less accessibility in CEpiLCs (Fig. 5a and Extended Data Fig. 7a).

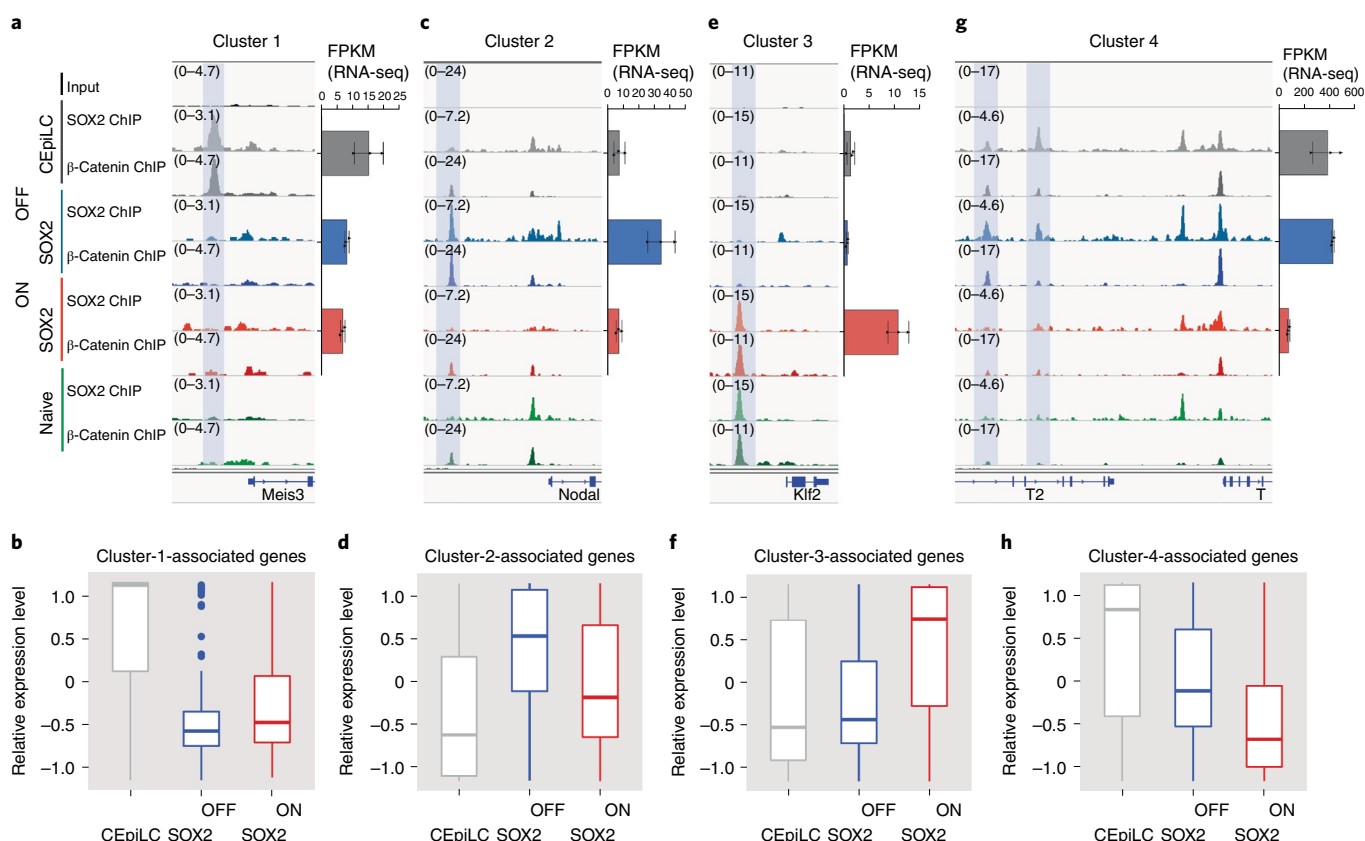

**Fig. 4 | SOX2/β-catenin co-occupancy correlates with cell-type-specific gene expression. a**, SOX2/β-catenin co-occupancy at a CRE shaded blue upstream of the transcriptional start site of Meis3 correlates with its cell-type-specific expression in CEpiLCs. **b**, Differentially expressed genes associated with cluster 1 CREs are most highly expressed in CEpiLCs. **c**, SOX2/β-catenin co-occupancy at an upstream of the transcriptional start site of Nodal correlates with its cell-type-specific expression in SOX2-OFF cells. **d**, Differentially expressed genes associated with cluster 2 CREs are most highly expressed in SOX2-OFF. **e**, SOX2/β-catenin co-occupancy at a CRE upstream of the pluripotency marker *Klf2* correlates with its cell-type-specific expression in SOX2-ON cells. **f**, Differentially expressed genes associated with cluster 3 CREs are most highly expressed in SOX2-ON. **g**, Reduced SOX2/β-catenin co-occupancy at multiple CREs upstream of the transcriptional start site of *T/Bra* in SOX2-ON and naive ESCs correlates with the absence of T/Bra expression in those cell states. **h**, Differentially expressed genes associated with cluster 4 CREs are expressed at lowest levels in SOX2-ON. Differential expression criterion for genes analysed in **b**, **d**, **f** and **h** is FDR <0.05, as determined by DESeq2 padj metric from $n = 3$ biological replicates. Data are represented by Tukey box plots (centre line is median, box limits are upper and lower quartiles, whiskers are 1.5× interquartile range and points are outliers). Source numerical data are available in source data.

Thus, whereas changes in chromatin accessibility may explain the specific occupancy of SOX2/TCF/β-catenin complexes at cluster 3 CREs in SOX2-ON and pluripotent cells, which express high levels of SOX2, they do not explain the cell-state-specific occupancy at other clusters.

To exclude the possibility that accessibility at SOX2-occupied cluster 3 CREs might be an indirect consequence of the WNT-dependent naive pluripotent cell state, we analysed EpiLCs cultured in the absence of CHIR, which express high levels of SOX2 but do not express naive pluripotency genes (Fig. 2d). SOX2 occupancy was higher at CREs associated with pluripotency and neural differentiation genes, and lower at CREs associated with CEpiLC genes in EpiLCs compared with CEpiLCs (Fig. 5c). Average SOX2 occupancy and accessibility at cluster 3 CREs was also higher in EpiLCs (Fig. 5d,e and Extended Data Fig. 7b), supporting the idea that high SOX2 levels promote accessibility at these sites in the absence of WNT signalling.

We explored whether the increased ATAC–seq signal at cluster 3 CREs in SOX2-ON could be driven directly by SOX2 occupancy by analysing the nucleosome landscape at sites of SOX2 binding. NucleoATAC analysis[32] revealed that average nucleosome occupancy at the centre of cluster 3 SOX2 binding peaks was markedly

depleted in high-SOX2-expressing SOX2-ON cells compared with low-SOX2-expressing SOX-OFF cells and WT CEpiLCs (Fig. 5f,g). By contrast, the average nucleosome density at SOX2 peak centres in cluster 4 peaks was largely independent of SOX2 levels (Fig. 5f). We conclude that SOX2 binding directly drives nucleosome eviction, rather than being a secondary consequence of increased neighbouring accessibility.

We hypothesized that the distinct relationship between SOX2 levels and nucleosome occupancy at cluster 3 peaks may reflect the affinity of SOX2 binding sites within the underlying sequence. Motif analysis identified both more SOX2 sites and a greater proportion of peaks with high-affinity SOX2 motifs in cluster 3 (Fig. 5h and Extended Data Fig. 7c,d). Within this cluster, motif score correlated with SOX2 occupancy and nucleosome depletion in SOX2-ON cells (Extended Data Fig. 7e). What then explains the change in SOX2 binding in CEpiLCs?

**SOX2 occupies low-affinity sites with cell-specific factors.** We reasoned that SOX2 occupancy at constitutively accessible sites might require additional cell-state-specific co-factors[33,34]. Motif enrichment analysis (Fig. 6a) revealed that cluster 1 peaks (CEpiLC-specific SOX2 binding) were enriched for CDX/HOX

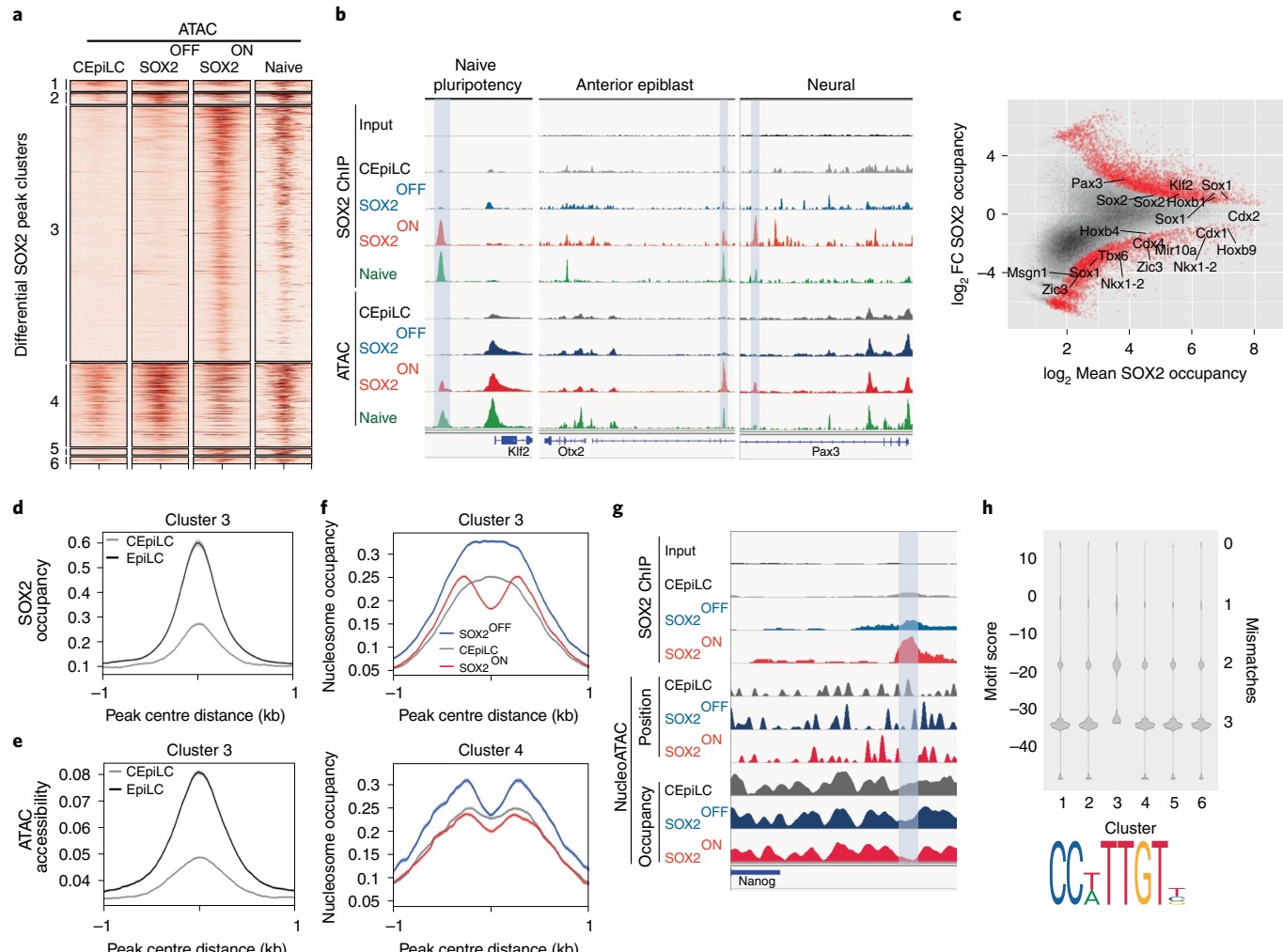

**Fig. 5 | SOX2 promotes chromatin accessibility at high-affinity sites. a**, Cluster 3 CREs classified in Fig. 3e exhibit ATAC–seq accessibility specifically in high-SOX2-expressing cells. Naive ATAC–seq data re-analysed from ref. [63]. **b**, Correlated SOX2 occupancy and ATAC–seq accessibility at representative cluster 3 CREs (shaded blue) associated with genes expressed in high-SOX2-expressing cell types. **c**, SOX2 occupancy in CEpiLCs is higher at peaks associated with trunk patterning and mesoderm genes and lower at peaks associated with pluripotency and neural genes than in EpiLCs. Differentially occupied peaks (red) are statistically different between conditions (FDR <0.05). FDR was determined by DESeq2 padj metric from $n = 3$ biological replicates. **d,e**, Cluster 3 peaks exhibit greater average SOX2 occupancy (**d**) and ATAC–seq accessibility (**e**) in EpiLCs than in CEpiLCs (re-analysed from ref. [35]). **f**, Average nucleosome occupancy at cluster 3 CREs is reduced at SOX2 peak centres in SOX2-ON, but relatively unchanged between cell types at cluster 4 peaks. **g**, Negative-correlation between SOX2 occupancy, nucleosome position and nucleosome occupancy at a representative CRE (shaded blue). **h**, SOX2 binding motifs at cluster 3 peaks most closely resemble the consensus sequence shown, as determined by FIMO calculated motif score (Methods). Source numerical data are available in source data.

motifs, factors expressed specifically in CEpiLCs (Fig. 2a,f). Cluster 4 sites, which are occupied by SOX2/TCF/β-catenin in both CEpiLCs and SOX-OFF primitive streak progenitors, were enriched for motifs for T/BRA, which is repressed in SOX2-ON. In addition, cluster 2 sites were enriched for the Nodal signalling mediator FOXH1, indicating that elevated Nodal signalling may contribute to the regulation of the distinct early primitive streak WNT response in SOX-OFF cells.

Consistent with these motif enrichment results, analysis of ChIP–seq data from CEpiLCs indicated that T/BRA was enriched, along with SOX2 and β-catenin, at both cluster 2 and cluster 4 sites (Fig. 6b,c and Extended Data Fig. 8a). Similarly, CDX2 ChIP–seq from CEpiLCs confirmed an increased CDX2 co-occupancy with SOX2 and β-catenin at cluster 1 sites (Fig. 6d,e and Extended Data Fig. 8a). Moreover, a larger proportion of low-affinity SOX2 motifs were found within close proximity (<100 bp) to CDX binding motifs

within cluster 1 peaks than in clusters 2–4 (Extended Data Fig. 8b). Similarly, a larger proportion of low-affinity SOX2 motifs within cluster 2 and 4 peaks were located closer to T/BRA binding motifs than in other clusters (Extended Data Fig. 8c). This suggested that CDX and T/BRA may act as co-factors to promote cell-type-specific recruitment of SOX2 or β-catenin.

To test directly whether cell-type-specific TFs such as CDX and T/BRA mediate SOX2/TCF/β-catenin occupancy, we performed ChIP–seq for SOX2, β-catenin and LEF1 in CEpiLC cells derived from ESCs either mutant for *T/Bra* (T/BraKO) or triple mutant for *Cdx1,2,4* (CdxKO) (ref. [35]). Strikingly, both SOX2 and β-catenin exhibited changes in binding in the absence of either CDX factors or T/BRA, and tended to be reduced at peaks adjacent to transcriptional targets of CDX (Fig. 6f,g,h) and T/BRA (Extended Data Fig. 8d–f). CdxKO cells showed reduced SOX2, β-catenin and LEF1 occupancy across cluster 1 CEpiLC-specific CREs co-occupied by SOX2 and

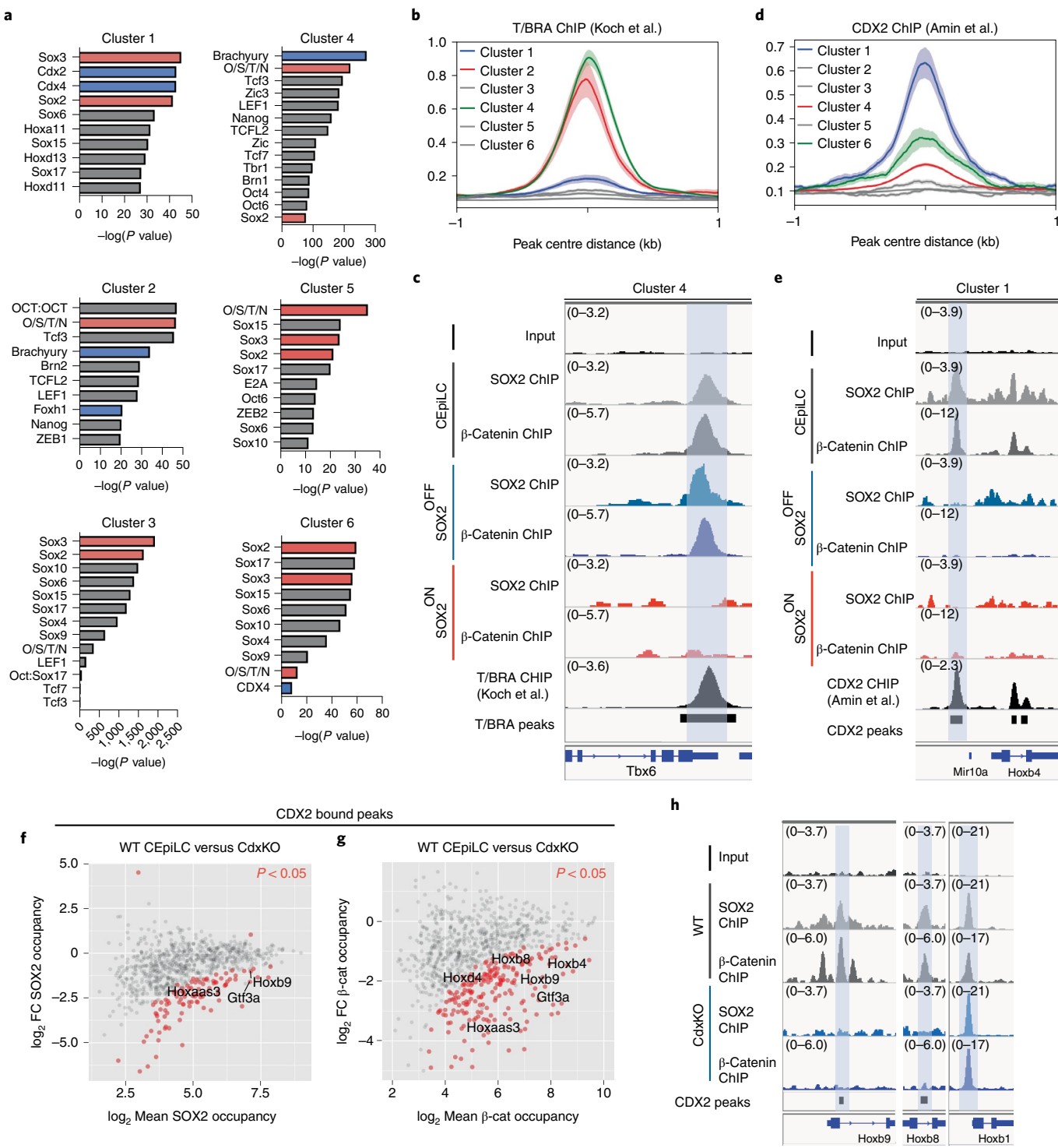

**Fig. 6 | SOX2 associates with cell-type-specific factors at low-affinity sites. a**, Cluster 1–6 CREs are enriched for distinct sets of TF binding motifs as determined by Homer (Methods). **b**, Cluster 2 and 4 CREs are enriched for T/BRA occupancy. T/BRA ChIP–seq re-analysed from ref. [12]. Metaplots show mean ± s.e.m. **c**, Correlated SOX2, β-catenin and T/BRA occupancy at a representative cluster 4 CRE (shaded blue) at the *Tbx6* locus. **d**, Cluster 1 CREs are enriched for CDX2 occupancy. CDX2 ChIP–seq re-analysed from ref. [64]. Metaplots show mean ± s.e.m. **e**, Correlated SOX2, β-catenin and CDX2 occupancy at a representative cluster 1 CRE (shaded blue) at the *HoxB* locus. O/S/T/N, OCT4/SOX2/TCF7L1/NANOG. **f,g**, Differential occupancy of SOX2 (**f**) and β-catenin (**g**) at CDX2 co-occupied peaks in CDX 1,2,4$^{-/-}$ mutant progenitors compared with CEpiLCs. Differentially occupied peaks (red) are statistically different between conditions (FDR <0.05). FDR was determined by DESeq2 padj metric from *n* = 3 biological replicates. Labelled peaks are adjacent to genes differentially expressed in CDX 1,2,4$^{-/-}$ mutant progenitors. FDR <0.05 determined by DESeq2 padj metric from *n* = 3 biological replicates. **h**, SOX2 and β-catenin occupancy is specifically reduced at CREs adjacent to trunk-expressed *HoxB8* and *HoxB9* in CDX 1,2,4$^{-/-}$ mutant progenitors.

CDX2 (Extended Data Fig. 8g–i), and reduced β-catenin at cluster 4 sites normally occupied by CDX factors (Extended Data Fig. 8j–l). These data support a direct role for CDX factors in configuring the CEpiLC WNT response. Similar results were observed for T/BraKO in clusters 2 (Extended Data Figs. 8m,n,o) and 4 (Extended Data Fig. 8p,q,r). As these changes in SOX2/β-catenin/LEF1 occupancy were not accompanied by discernable changes in nucleosome occupancy in either CdxKO or T/BraKO cells (Extended Data Fig. 8s), these data suggest that CDX and T/BRA regulate cell-type-specific WNT target-gene expression by directing the recruitment of SOX2 and TCF/β-catenin to constitutively accessible CREs.

**SOX2 levels control CDX2 enhancer activity.** WNT-induced CDX2 expression is constrained to a specific range of SOX2 levels (Fig. 2a and Extended Data Fig. 9a). A previously identified regulatory element within the CDX2 intron[36,37] displayed a cell-type-specific pattern of SOX2/β-catenin co-occupancy that correlated with SOX2 levels (Fig. 7a). We generated fluorescent reporter lines harbouring the intronic sequence (Fig. 7b). Both CDX2 expression and reporter activity were higher in CEpiLCs cultured in FLC medium than in pluripotent ESCs, which express high levels of SOX2, and activin-induced early primitive streak cells (Fig. 7c–e and Extended Data Fig. 9b) that express little if any SOX2 or Sox3 (Extended Data Fig. 9c,d). To test whether SOX2 regulates the CDX2 intronic enhancer, we scrambled all SOX2 binding sites in the reporter (Sox2del) (Fig. 7b). Activity of the Sox2del reporter was substantially reduced in CEpiLCs (Fig. 7f and Extended Data Fig. 9e,f) consistent with the idea that SOX2 occupancy promotes the induction of CDX2 by WNT signalling, and that its repression in ESCs is indirect.

SOX2 and CDX2 are repressed by Nodal signalling in early primitive streak progenitors[38,39]. Nodal expression is elevated in SOX2-OFF primitive streak progenitors (Fig. 3d). Inhibition of Nodal signalling in SOX2-OFF cells concurrently with WNT pathway activation led to the inhibition of both the general primitive streak marker T/Bra and of early primitive streak markers Eomes, Mixl1 and Nanog (Extended Data Fig. 9g), the upregulation of Sox3 (Fig. 7g) and a rescue of Cdx2 expression (Fig. 7h). We conclude that the presence of moderate levels of SOX2 in CEpiLCs promotes posterior identity by both positively regulating CDX2 expression and restraining the induction of early primitive streak identity by Nodal.

## Discussion

Here we show that the level of SOX2 expression determines its genome-wide occupancy and this underpins distinct WNT-driven transcriptional programmes at successive stages of pluripotent stem cell differentiation. We found that β-catenin frequently co-occupies genomic sites with SOX2. Perturbations to SOX2 levels led to coordinated changes in the genomic location of SOX2 and β-catenin binding. During the transition from pluripotency to caudal epiblast identity, a reduction in global SOX2 levels resulted in a reduction of SOX2 occupancy at a set of CREs accompanied by a corresponding reduction in β-catenin occupancy. Many of these CREs were associated with genes expressed in pluripotent epiblast or neural ectoderm, cell types that require high levels of SOX2 expression to maintain their identity[14,24,27,40]. Surprisingly, the reduction in global SOX2 levels also resulted in an increase in SOX2 and β-catenin co-occupancy at a set of CREs. These were associated with WNT-responsive genes expressed in caudal epiblast progenitors, many of which are responsible for posterior patterning and mesoderm differentiation. Artificially increasing or decreasing SOX2 expression redistributed SOX2/β-catenin, and prevented the transition to a CLE identity. Thus, SOX2 levels configure the WNT response of epiblast progenitors and shape the transcriptional changes accompanying the differentiation of pluripotent cells to CLE.

Different levels of TF expression have been found to control differential gene expression programmes in several systems[41–45]. In many cases, the mechanistic basis for this has been unclear. Here we provide evidence that, for SOX2, the level of expression has a marked effect on the selection of CREs to which it binds, providing an explanation for the different gene expression responses. At high levels of expression, SOX2 remains bound to a set of CREs associated with neural and pluripotency genes. For this set of CREs, SOX2 binding correlated with chromatin accessibility. This is consistent with the known role of SOX2 as a pioneer factor and its ability to bind and open inaccessible CREs[28–31].

The decrease in SOX2 levels resulted in a repositioning of SOX2 to CREs associated with genes involved in posterior patterning and mesoderm induction. Despite the lower levels of SOX2, these CREs contained lower-affinity SOX2 binding sites than the CREs bound by SOX2 in cell types with high SOX2 expression levels. Moreover, these CREs were accessible in pluripotent conditions as well as in CLE. Co-factor-mediated recruitment to low-affinity sites has been implicated in cell-type-specific CRE activity and gene expression[33,34,46,47]. For SOX2, we found evidence of the involvement of CDX2 and T/BRA in directing binding to low-affinity sites. These observations suggest that SOX2 adopts different modes of chromatin interaction and CRE selection depending on its level of expression (Fig. 7i). This resolves a paradox. Despite its pioneering activity and ability to bind and activate condensed chromatin, the distribution of SOX2 occupancy on chromatin differs between cell types. Our data provide further evidence that SOX2 acts as a pioneer factor in pluripotent cells when expressed at high levels, but collaborates with other TFs to select lower-affinity binding sites when expressed at lower levels. This is consistent with previous studies of how TFs gain access to their genomic targets[29,48–50] and provides an explanation for the distinct gene expression programmes regulated at different TF expression levels.

**Fig. 7 | Cdx2 induction requires low-level SOX2/SOX3 expression. a**, SOX2 and β-catenin co-occupy a CRE within intron 1 of Cdx2 (shaded blue). **b**, WT and modified sequence (Sox2del) from the Cdx2 intronic CRE was cloned into a fluorescent reporter construct. **c**, Measurement of relative mRNA expression by RT–qPCR shows that Cdx2 expression is reduced in pluripotent and activin-induced early streak progenitors. CHIR concentration was 5 μM in 2i⁺ conditions. **d,e**, Flow cytometry quantification shows that the activity of the Cdx2 CRE reporter construct is reduced in early streak progenitors (**d**) and pluripotent progenitors (**e**) relative to caudal epiblast progenitors (CEpiLC). **f**, Flow cytometry quantification shows that deletion of SOX2 binding sites (Sox2del) reduces the activity of the Cdx2 CRE reporter in CEpiLCs. For details of reporter quantification in **d–f**, see Methods. **g,h**, Measurement of relative mRNA expression by RT–qPCR shows that inhibition of Nodal signalling elevates Sox3 (**g**), and Cdx2 (**h**), expression in SOX2-OFF progenitors. **i**, At high levels, SOX2 displaces nucleosomes from regulatory elements with high-affinity SOX2 binding sites, recruiting the WNT effector TCF/β-catenin and maintaining pluripotent gene expression. At lower SOX2 levels, SOX2/TCF/β-catenin occupancy is reconfigured to caudal epiblast expressed genes. These contain low-affinity SOX2 sites and are co-occupied by T/BRA and CDX. At very low SOX2 levels, early primitive streak genes are induced. Each datapoint in **c–h** represents an individual biological replicate. Bars denote mean ± s.e.m. P values calculated for differences of mean expression by two-tailed Student's t-test are shown. In **c**, n=10 for CEpiLC, n=8 for early streak and n=3 for naive; in **d**, n=6 for CEpiLC and early streak progenitors; in **e**, n=4 for CEpiLC and naive progenitors; in **f**, n=9 for the WT CRE and n=11 for Sox2del; in **g**, n=10 for CEpiLCs, n=10 for SB-treated CEpiLCs, n=8 for SOX2-OFF and n=8 for SB-treated SOX2-OFF; in **h**, n=8 for CEpiLCs, n=8 for SB-treated CEpiLCs, n=6 for SOX2-OFF and n=6 for SB-treated SOX2-OFF. SB, SB-431542. Source numerical data are available in source data.

There was a positive correlation between SOX2 binding and the activation of WNT-responsive genes in CLE cells. Consistent with this, using a CRE from CDX2, we found that SOX2 occupancy is required for CDX2 activation by WNT signalling, providing direct evidence of an activator role for SOX2 in the regulation of β-catenin target genes. This suggests a self-reinforcing mechanism for cell-type specificity of WNT signalling. Downregulation of SOX2 in CEpiLCs leads to reconfiguration of the chromatin state and prevents re-expression of

the pluripotent transcriptional programme. This eases repression on CLE-specific WNT target genes such as T/BRA and CDX2 (Figs. 2 and 7)[51,52]. Consequently, SOX2 and TCF/β-catenin are recruited to CREs associated with CDX and T/BRA target genes, inducing gene expression programmes characteristic of posterior identity and primitive streak to paraxial mesoderm differentiation. Then, as SOX2 levels are further reduced during the differentiation of CLE progenitors to mesoderm progenitors, CDX and T/BRA expression decreases[17,53].

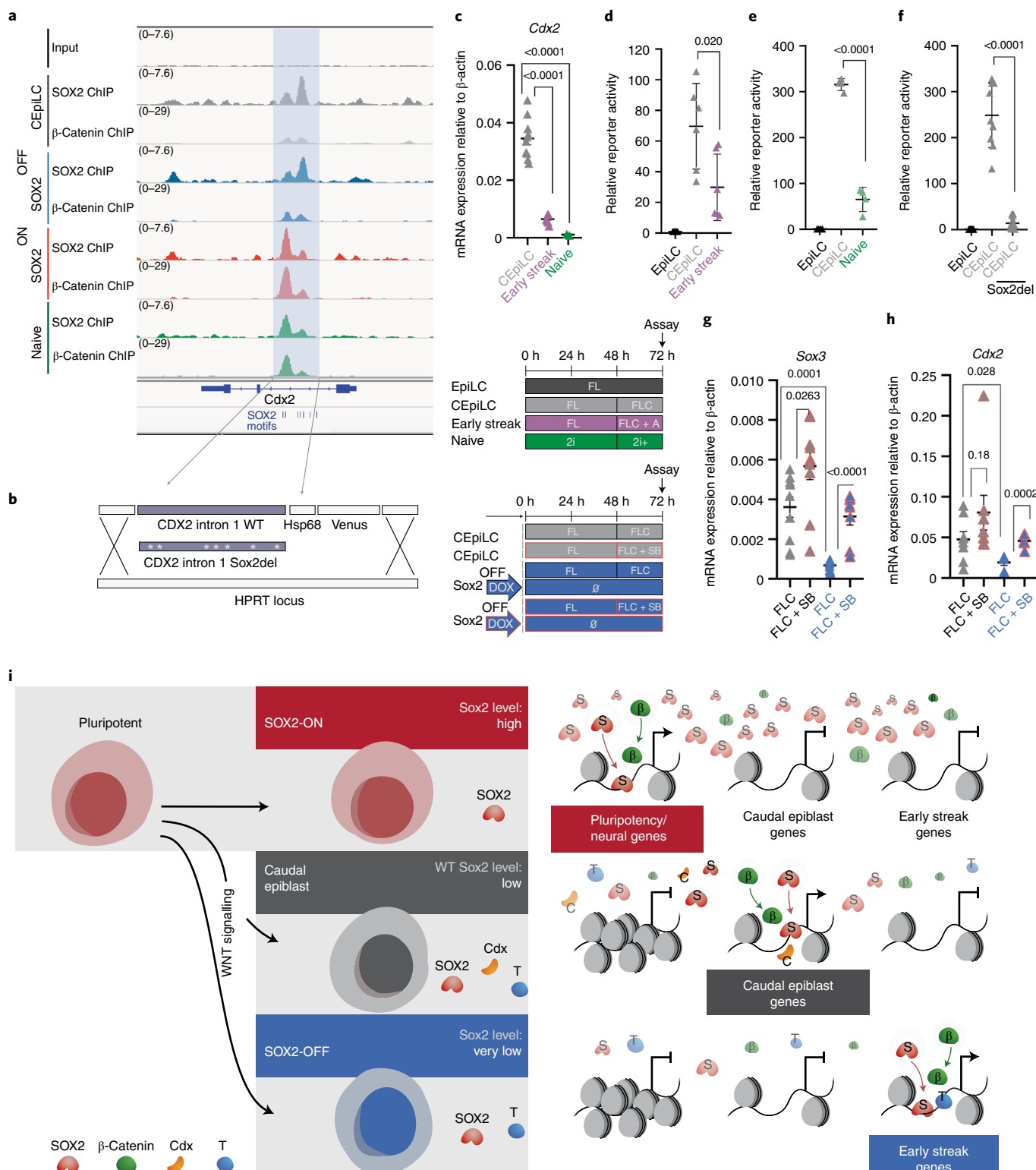

A consequence of the mechanism that establishes the primary body axis is that anterior and posterior structures derive from distinct epiblast progenitor pools[35]. Anterior tissues, including the forebrain and heart, are established early in embryonic development from pluripotent epiblast progenitors that co-express OCT4, NANOG and high levels of SOX2[16,18,38,54–59]. Caudal epiblast progenitors retain low levels of SOX2 expression, and this is required to assign trunk identity to both the mesoderm and spinal cord by establishing CDX/HOX expression in response to WNT signalling. As CREs associated with neural genes require high SOX2 to maintain accessibility, neural differentiation is restrained in CLE progenitors independently of inhibitory activity of OCT4/NANOG. By contrast, the initiation of mesoderm differentiation is controlled by regulation of T/BRA by WNT/Nodal signalling, independently of chromatin remodelling.

A division between the ontogeny of head and trunk tissue is also apparent in arthropods. Reminiscent of the CLE, homologues of SOX2 and CDX2 are co-expressed in a posterior progenitor pool that fuels WNT-dependent axis elongation. Moreover, the SOX orthologues have been shown to participate in the assignment of posterior identity within these progenitors[60–62]. A collaboration between WNT signalling and SOX2 in the regulation of CDX factors therefore appears to be an evolutionarily conserved mechanism that establishes the primary bilaterian axis and allocates cells to trunk tissues.

## Online content

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

## Methods

**Cell lines.** All cell lines were maintained and experiments performed at 37 °C with 5% $CO_2$. All ESC lines used were derived from the XY HM1 TetON line[66], which was used as the WT control. Sox2 TetON, *Cdx2* intron CRE reporter and *Cdx2* intron CRE reporter Sox2del lines were generated as described below. Cell lines were validated by DNA sequencing and flow cytometry, and routinely tested for mycoplasma.

**Sox2 TetON.** Sox2 TetON was generated by introducing a silent G > A mutation 54 bp into the open reading frame of *Sox2* complementary DNA (cDNA) using site-directed mutagenesis to ablate the protospacer adjacent motif site targeted by the guide Sox2_CRISPR_1. The Sox2_CRISPR_1-insensitive *Sox2* cDNA was then subcloned into pBI2 using Sal1/Not1 restriction digest and subsequently cloned into the HPRT locus targeting vector Hprt2 as described previously in ref. [66]. The HPRT_TetON-SOX2_54G > A construct was electroporated into HM1 TetON ESCs, and integrants were selected by culturing for 10 days in hypoxanthine-aminopterin-thymidine (HAT)-containing ESC medium. Construct integration was confirmed by genotyping, and transgene expression was confirmed by flow cytometry. SOX2_54G > A targeted cells were then adapted to 2i culture, the transgene was induced with Dox (1 µg ml⁻¹), and cells were electroporated with Sox2_crispr_1 using an Amaxa Nucleofector to ablate endogenous *Sox2* gene expression. Electroporated cells were seeded at clonal density on gelatin plates in 2i medium and the following day were selected by culturing in puromycin (1 µg ml⁻¹) for 36 h. Resistant clones were grown, picked, expanded and screened for SOX2 expression by flow cytometry to detect complete loss of SOX2 expression following withdrawal of Dox. Following ablation of endogenous SOX2, Sox2 TetON were stably maintained in pluripotency conditions by addition of 1 µg ml⁻¹ Dox to serum + leukaemia inhibitory factor (LIF) ESC culture medium. Oligonucleotide sequences are detailed in Supplementary Table 1.

**Sox2 TetON, Sox3⁻.** *Sox3* was ablated from Sox2 TetON by electroporating Sox3_CRISPR_1 and selecting edited clones using the approach described above for *Sox2* ablation. Functional ablation of the single *Sox3* allele was confirmed by genotyping to identify clones with frameshift mutations, and subsequent functional analysis. Oligonucleotide sequences are detailed in Supplementary Table 1.

**Cdx2 intron CRE reporter.** GeneBlock oligonucleotides coding for a 1.6 kb fragment of WT sequence containing the SOX2 peaks within *Cdx2* intron 1, or the same region in which seven predicted SOX2 motifs (JASPAR)[67] were scrambled (Sox2del), were cloned by Gibson assembly into a pENTR11 backbone upstream of an *hsp68* minimal promoter driving expression of a Venus-H2B transgene. Additionally, the Gateway cassette from FuTetO-GW (Addgene) was cloned by Gibson assembly into the Asc1/Pme1 site of Hprt2 (ref. [66]) to yield HPRT_GW. LR clonase (Invitrogen) was used to induce recombination between the pENTR reporter construct and HPRT_GW, yielding HPRT-locus targeting constructs, which were used to generate stable lines in HM1 TetON ESCs as described for Sox2 TetON. Oligonucleotide sequences are detailed in Supplementary Table 1.

**ESC culture and differentiation.** All mESCs were propagated on mitotically inactivated mouse embryonic fibroblasts (feeders) in DMEM knockout medium supplemented with 1,000 U ml⁻¹ LIF, 10% cell-culture-validated foetal bovine serum, penicillin–streptomycin and 2 mM ʟ-glutamine (Gibco). To obtain EpiLCs and CEpiLCs, ESCs were differentiated as previously described[9] with the addition of the porcupine inhibitor LGK974 in all culture media. Briefly, ESCs were dissociated with 0.05% trypsin, and plated on tissue-culture-treated plates for two sequential 20-min periods in ESC medium to separate them from their feeder layer cells, which adhere to the plastic. To start the differentiation, cells remaining in the supernatant were pelleted by centrifugation, counted and resuspended in N2B27 medium containing 10 ng ml⁻¹ bFGF + 5 µM, and 50,000 cells per 35 mm gelatin-coated CellBIND dish (Corning) were plated. N2B27 medium contained a 1:1 ratio of DMEM/F12:Neurobasal medium (Gibco) supplemented with 1× N2 (Gibco), 1× B27 (Gibco), 2 mM ʟ-glutamine, 40 mg ml⁻¹ BSA (Sigma), penicillin–streptomycin and 0.1 mM 2-mercaptoethanol. To generate EpiLCs, the cells were grown for 72 h in N2B27 + 10 ng ml⁻¹ bFGF + 5 µM LGK974 (FL medium). To generate CEpiLCs, cells were cultured with N2B27 + 10 ng ml⁻¹ bFGF + 5 µM LGK974 for 48 h, then N2B27 + 10 ng ml⁻¹ bFGF + 5 µM LGK974 + 5 µM CHIR99021 (FLC medium) for a further 24 h (day 3 in ref. [9]). CEpiLCs were differentiated to spinal cord neural progenitors by removal of bFGF and CHIR from culture medium at 72 h, and to paraxial mesoderm by removal of bFGF and maintenance of 5 µM CHIR from 72 h onwards. When investigating the activity of Nodal signalling, either 10 ng ml⁻¹ recombinant activin or 10 µM ALK-inhibitor SB-431542 was included in bFGF/CHIR-containing medium. Experiments conducted in 2i medium were initiated by separating serum/LIF-grown ESCs from feeders as described above and plating onto gelatin-coated CellBIND dishes in N2/B27-containing basal medium supplemented with 3 µM CHIR and 500 nM PD0325901. For all experiments described, cells were cultured for 48 h before changing medium. Medium changes were then made every 24 h. Details of key compounds are described in Supplementary Table 1.

**Immunofluorescence.** Cells were washed in PBS and fixed in 4% paraformaldehyde in PBS for 15 min at 4 °C, followed by two washes in PBS and one wash in PBST (0.1% Triton X-100 diluted in PBS). Primary antibodies were applied overnight at 4 °C diluted in filter-sterilized blocking solution (1% BSA diluted in PBST). Cells were washed three times in PBST and incubated with secondary antibodies at room temperature, for 1 h. Cells were washed three times in PBST, incubated with DAPI for 5 min in PBS and washed twice before mounting with Prolong Gold (Invitrogen). Cells were imaged on a Zeiss Imager. Z2 microscope using the ApoTome.2 structured illumination platform. Z stacks were acquired using Zeiss Zen software and represented as maximum intensity projections using ImageJ software. The same settings were applied to all images. Immunofluorescence was performed on a minimum of two biological replicates, from independent experiments. Secondary antibodies used were anti-mouse AlexaFluor 488 (Thermo Fisher), anti-rabbit AlexaFluor 488 (Thermo Fisher), anti-rabbit AlexaFluor 647 (Thermo Fisher) and anti-goat AlexaFluor 647 (Thermo Fisher). Details of primary antibodies are described in Supplementary Table 1.

**Intracellular flow cytometry.** Cells were washed in PBS and dissociated with minimal accutase (Gibco). Once detached, cells were collected into 1.5 ml Eppendorf tubes by dissociating in N2B27 and pelleted. Cells were resuspended in PBS, pelleted and resuspended in 4% paraformaldehyde in PBS. Following 15 min incubation at 4 °C, cells were centrifuged at 700 relative centrifugal force, resuspended in PBS and stored at 4 °C for future analysis. On the day of flow cytometry, cells were counted and equal cell numbers were transferred for staining in V-bottom 96-well plates. Samples were pelleted and resuspended in 5 µl FACS block (PBS + 0.2% Triton + 3% BSA). After 10 min incubation at room temperature, antibodies were added to the sample and incubated overnight at 4 °C. Cells were pelleted at 700*g* for 5 min and resuspended in 50 µl FACS block. One additional wash was performed before acquisition on a Fortessa flow cytometer (BD) using FACSDiva software. Analysis was performed using the R package flowCore[68] and data were graphed using ggplot2 (ref. [69]). A representative figure illustrating the gating strategy is provided in Extended Data Fig. 10. Details of antibodies are described in Supplementary Table 1.

**Quantification of flow cytometry data.** To determine the relative response of the *Cdx2* intron CRE reporter to WNT pathway activation in CEpiLCs, early streak and naïve progenitors, the median fluorescence intensity was determined and normalized against the value obtained for unstimulated EpiLCs. The same approach was taken when investigating the consequence of SOX2 binding site deletion in the Sox2del line.

**RNA extraction.** RNA used for quantitative PCR (qPCR) or RNA sequencing (RNA-seq) was extracted from cells using a QIAGEN RNeasy kit in RLT buffer, following the manufacturer's instructions. Extracts were digested with DNase I to eliminate genomic DNA.

**cDNA synthesis and qPCR analysis.** First-strand cDNA synthesis was performed using Superscript III (Invitrogen) using random hexamers and was amplified using PowerUp SYBR-Green Mastermix (Applied Biosystems). qPCR was performed using the Applied Biosystems QuantStudio Real Time PCR system and analysed with Applied Biosystems QuantStudio 12 K Flex software. PCR primers were designed using online GenScript qPCR primer design tool. Two technical replicates were obtained for each sample and averaged before normalization and statistical analysis. Relative expression values for each gene were calculated by normalization against β-actin, using the delta–delta CT method. qPCR analysis was performed on samples obtained from a minimum of three independent experiments for every primer pair analysed. Data were graphed and statistical tests were performed using GraphPad Prism software. Primer sequences are detailed in Supplementary Table 1.

**RNA-seq.** Libraries were prepared using the KAPA mRNA HyperPrep kit (Roche) and sequenced as 76 bp single-end, strand-specific reads on the Illumina HiSeq 4000 platform (Francis Crick Institute).

**RNA-seq analysis.** Adapter trimming was performed with cutadapt (version 1.16)[70] with parameters '–minimum-length=25 –quality-cutoff=20 -a AGATCGGAAGAGC', and for paired-end data '-A AGATCGGAAGAGC' was appended to the command. The RSEM package (version 1.3.0)[71] in conjunction with the STAR alignment algorithm (version 2.5.2a)[72] was used for the mapping and subsequent gene-level counting of the sequenced reads with respect to mm10 RefSeq genes downloaded from the UCSC Table Browser[73] on 11 December 2017. The parameters passed to the 'rsem-calculate-expression' command were '–star –star-gzipped-read-file –star-output-genome-bam –forward-prob 0', and for paired-end data '–paired-end' was appended to the command. Differential expression analysis was performed with the DESeq2 package (version 1.16.1)[74] within the R programming environment (version 3.4.1). An adjusted *P* value ≤0.05 was used as the significance threshold for the identification of differentially expressed genes.

**RNA-seq clustering.** The R 'kmeans' function was used to cluster standardized (*z*-transformed) FPKM values across biological conditions before plotting with R

'heatmap2' function. The lowest value of $k$ able to partition gross trends in the data was chosen.

**RNA-seq associating differential gene expression with differential SOX2 occupancy.** Homer 'annotatePeaks.pl' was used to associate consensus SOX2 ChIP peaks with nearest gene promoters. SOX2-associated genes were then filtered on the basis of their differential expression in pairwise comparisons between either CEpiLC, SOX2-OFF and SOX2-ON; WT CEpiLC and *Cdx1,2,4⁻/⁻* (CdxKO) CEpiLC; or WT CEpiLC and *T/Bra⁻/⁻* (T/BraKO) CEpiLC using DESeq2. Mean FPKM values from triplicate samples were z-transformed across the three experimental conditions to standardize fold change in expression and plotted using ggplot2.

**RNA-seq GO enrichment.** The online functional annotation tool of the DAVID bioinformatics resource https://david.ncifcrf.gov/summary.jsp was used with default parameters to identify statistically enriched biological process annotations within sets of gene IDs associated with differentially expressed transcripts, and to calculate associated Benjamini–Hochberg adjusted *P* values.

**RNA-seq comparison of in vitro to in vivo epiblast differentiation.** Principal component analysis was performed on mRNA-seq data from duplicate 2i, and triplicate ICM, E4.5 epiblast and E5.5 epiblast samples from ref. [65] using using the R function prcomp. PC1 aligned with developmental time, whereas PC2 separated in vitro (2i) and in vivo (ICM, E4.5 and E5.5) derived samples. The top 300 genes contributing most positively and negatively to PC1 were selected to represent the gene expression dynamics observed to occur during epiblast differentiation in vivo, and the dynamics of their expression during in vitro differentiation of ESCs to EpiLCs was represented by plotting standardized (z-transformed) FPKM values using heatmap2.

**ChIP–seq.** Adherent cells were washed three times with PBS, fixed with gentle agitation for 45 min at room temperature with fresh 2 mM di(*N*-succinimidyl) glutarate (Sigma) in PBS, washed an additional three times with PBS, then fixed for 10 min at room temperature with 1% molecular-biology-grade paraformaldehyde in PBS. Fixation was quenched by addition of 250 mM glycerine for 5 min, followed by additional washing with PBS. Plates were cooled, and cells were scraped into tubes in a low volume of PBS 0.02% Triton X-100 and pelleted by centrifugation at 100*g* for 5 min at 4 °C before snap freezing in liquid nitrogen and storing at −80 °C. Approximately $5 \times 10^6$ cells were transferred to a Diagenode TPX tube and resuspended in ice-cold shearing buffer containing 0.3% SDS and protease inhibitors (Sigma). Chromatin was sheared using a Bioruptor plus: 25 cycles of 30 s on/30 s off on high setting, and lysates were then diluted to 0.15% SDS and cleared by centrifugation at 14,000 r.p.m. for 10 min at 4 °C. Then, 1/20 of the chromatin from ~$1 \times 10^7$ cells was set aside and frozen for subsequent use as input control, and the remainder was incubated overnight at 4 °C under rotation with 100 µl of protein G dynabeads (Invitrogen) pre-loaded for 4 h at room temperature with 5 µg of ChIP antibodies diluted in shearing buffer containing 0.15% SDS. Beads were magnetically immobilized, unbound supernatant was discarded and beads were sequentially washed under rotation twice with Wash Buffer 1, once with Wash Buffer 2, once with Wash Buffer 3 and twice with Wash Buffer 4 for 5 min each, magnetically capturing beads between each wash. Chromatin was eluted from beads by incubating twice at 65 °C for 10 min in 100 µl elution buffer on a shaking heat block, capturing beads between each elution step and then pooling each eluted fraction. Input samples were made up to 200 µl with elution buffer, 6.4 µl of 5 M NaCl was added to each input or immunoprecipitated sample, and all samples were de-crosslinked overnight at 65 °C. Samples were incubated for 2hrs at 37 °C with 0.2 µg ml⁻¹ PureLink RNAse A (Invitrogen), then supplemented with 5 mM EDTA and incubated for an additional 2 h at 45 °C with 0.2 µg ml⁻¹ proteinase K (Thermo Scientific) before purifying DNA with Qiagen PCR clean-up columns. DNA fragmentation of IP and input samples was confirmed by Agilent TapeStation before library preparation using NEB Ultra II DNA. Biological triplicates were obtained for all conditions from separate experiments. Libraries were sequenced as single-end, 76 bp reads on the Illumina High-Seq 4000 platform (Francis Crick Institute). The composition of buffers and details of antibodies are described in Supplementary Table 1.

**ChIP-seq analysis.** The nf-core/ChIP-seq pipeline (version 1.1.0; https://doi.org/10.5281/zenodo.3529400)[75] written in the Nextflow domain specific language (version 19.10.0)[76] was used to perform the primary analysis of the samples in conjunction with Singularity (version 2.6.0)[77]. The command used was ' nextflow run nf-core/ChIP-seq –input design.csv –genome mm10 –gtf refseq_genes.gtf –single_end –narrow_peak –min_reps_consensus 2 -profile crick -r 1.1.0'. To summarize, the pipeline performs adapter trimming (Trim Galore! – https://www.bioinformatics.babraham.ac.uk/projects/trim_galore/), read alignment (BWA)[78] and filtering (SAMtools)[79]; (BEDTools)[80]; (BamTools)[81]; (pysam - https://github.com/pysam-developers/pysam); (picard-tools; http://broadinstitute.github.io/picard), normalized coverage track generation (BEDTools)[80]; (bedGraphToBigWig)[82], peak calling (MACS) (default *q*-value threshold <0.05)[83] and annotation relative to gene features (HOMER)[84], consensus peak set creation (BEDTools)[80], differential binding

analysis (featureCounts)[85]; (DESeq2)[74] and extensive quality control and version reporting (MultiQC)[86]; (FastQC; https://www.bioinformatics.babraham.ac.uk/projects/fastqc/); (preseq); deepTools[87]; (phantompeakqualtools)[88]. Inclusion of a peak in the consensus peak set required that it be called by MACS in a minimum of two of three biological replicates from any of the four experimental conditions (CEpiLC, SOX2-OFF, SOX2-ON and naïve ESCs). In all analyses, except for Fig. 3c and Extended Data Fig. 4c, the consensus peak set was derived from SOX2 peaks. For Fig. 3c and Extended Data Fig. 4c, the consensus peak set comprised peaks from SOX2/β-catenin/TCF7L1/LEF1. All data were processed relative to the mouse UCSC mm10 genome (UCSC)[73] downloaded from AWS iGenomes (https://github.com/ewels/AWS-iGenomes). Peak annotation was performed relative to the same GTF gene annotation file used for the RNA-seq analysis. Tracks illustrating representative peaks were visualized using the IGV genome browser[89].

**ChIP–seq peak clustering.** SOX2 peaks were manually assigned to six clusters on the basis of differential occupancy between WT, SOX2-OFF and SOX2-ON samples. Peaks in clusters 1, 2 and 3 had the highest mean read counts across biological triplicate samples in either WT, SOX2-OFF or SOX2-ON respectively, and were statistically different (false discovery rate (FDR) <0.05) as determined by DESeq2 compared with all other experimental conditions. Cluster 4, 5 and 6 peaks were statistically different to only one of the other experimental conditions. Browser Extensible Data (BED) files of genomic intervals defined by SOX2 peaks within these clusters were used to plot metaplots and heat maps from the BigWig files generated from the nf-core/ChIP-seq and nf-core/ATAC-seq pipelines using deepTools, for motif enrichment analysis and motif scanning.

**ChIP–seq motif enrichment.** Motifs enriched within each SOX2 peak cluster were identified using Homer[84] findMotifsGenome using default parameters. Region size was 200 bp (±100 bp adjacent to peak centre).

**ChIP–seq motif scoring with FIMO.** Regions ±100 bp adjacent to SOX2 ChIP–seq peak centres were used as inputs for the motif scanning tool Find Individual Motif Occurrences (FIMO) http://meme-suite.org/tools/fimo (ref. [90]). The SOX2 motif MA0143.3 (JASPAR)[67] was used as a target. *P*-value threshold was set to $P < 0.1$ so as to include low-scoring SOX2 motifs present within peak sets. Cluster 3 peaks were ranked on the basis of the total score of all motifs within each region with a score greater than −20, which represents up to two mismatches compared with the consensus. All ±100 bp regions within cluster 1–6 peaks contained at least one motif with a score greater than −20.

**ChIP–seq peak intersection.** BEDtools[80] intersectBed was used to identify genomic intervals overlapping by >10% in BED files listing coordinates of consensus and differentially occupied peak sets for each immunoprecipitated factor.

**ATAC–seq.** ATAC–seq sample preparation was performed as described in ref. [35]. Briefly, adherent cells were treated with StemPro Accutase (ThermoFisher) to obtain a single cell suspension, counted and resuspended to obtain 50,000 cells per sample in ice-cold PBS. Cells were pelleted and resuspended in lysis buffer (10 mM Tris–HCl pH 7.4, 10 mM NaCl, 3 mM MgCl₂ and 0.1% IGEPAL). Following a 10 min centrifugation at 4 °C, nucleic extracts were resuspended in transposition buffer for 30 min at 37 °C and purified using a QIAGEN MinElute PCR Purification kit following the manufacturer's instructions. Transposed DNA was eluted in a 10 µl volume and amplified by PCR with Nextera primers to generate single-indexed libraries. Libraries were sequenced as paired-end, 101 bp reads on the Illumina High-Seq 4000 platform (Francis Crick Institute).

**ATAC–seq analysis.** The nf-core/atacseq pipeline (version 1.0.0; https://doi.org/10.5281/zenodo.2634133)[75] written in the Nextflow domain specific language (version 19.10.0)[76] was used to perform the primary analysis of the samples in conjunction with Singularity (version 2.6.0)[77]. The command used was ' nextflow run nf-core/ATAC-seq –design design.csv –genome mm10 –gtf refseq_genes.gtf -profile crick -r 1.0.0'. The nf-core/ATAC-seq pipeline uses similar processing steps as described for the nf-core/ChIP-seq pipeline in the previous section but with additional steps specific to ATAC–seq analysis, including removal of mitochondrial reads.

**Nucleosome analysis.** The NucleoATAC package (version 0.3.4)[32] was run in default mode. Analysis was performed on all genomic intervals called as peaks from ATAC–seq data as described above. Metaplots of the occ.bedgraph files for each experimental condition were plotted using deepTools to score the average nucleosome occupancy within each peak cluster. Tracks of the occ.bedgraph and nucleoatac_signal.smooth.bedgraph files were visualized using the IGV genome browser[89] to illustrate the occupancy and position of nucleosomes at genomic intervals of interest.

**Statistics and reproducibility.** No statistical method was used to pre-determine sample size. No data were excluded from the analyses. The experiments were not randomized. The investigators were not blinded to allocation during experiments and outcome assessment. Software used for statistical analysis is detailed in Supplementary Table 2.

For all statistical analyses, data were obtained from a minimum of three independent experiments. Details of replicate numbers, quantification and statistics for each experiment are specified in the figure legends.

**Availability of unique biological material.** All embryonic stem cell lines described for the first time in this study are available from James Briscoe upon request.

**Reporting Summary.** Further information on research design is available in the Nature Research Reporting Summary linked to this article.

## Data availability
Deep-sequencing (ChIP–seq, ATAC–seq and RNA-seq) data generated during this study have been deposited in the Gene Expression Omnibus (GEO) under the accession code GSE162774. Previously published ChIP–seq, ATAC–seq and RNA-seq data that were re-analysed during this study are available under accession codes GSE64059, GSE84899, GSE93524, E-MTAB-2268, E-MTAB-2958 and E-MTAB-6337. Details of individual samples re-analysed are described in Supplementary Table 3. Source data for Figs. 1,2,4,5 and 7 and Extended Data Figs. 1,2,3 and 9 are provided in source data. All other data supporting the findings of this study are provided in supplementary information or are available from the corresponding author on reasonable request. Source data are provided with this paper.

## Code availability
All data were processed using published nf-core pipelines as detailed in Methods.

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

## Acknowledgements

We are grateful to T. Frith, K. Ivanovitch and M. Melchionda for experimental support and critical feedback during manuscript preparation. We also thank A. Sagner and other members of the lab for their generosity sharing insight, expertise and reagents, and the Crick Science Technology Platforms, in particular the Advanced Sequencing Facility, Flow Cytometry Facility, and the Bioinformatics and Biostatistics group. This work was supported by the Francis Crick Institute, which receives its core funding from Cancer Research UK, the UK Medical Research Council and Wellcome Trust (all under FC001051); and by the European Research Council under European Union (EU) Horizon 2020 research and innovation program grant 742138. This research was funded in whole, or in part, by the Wellcome Trust (FC001051). For the purpose of Open Access, the authors have applied a CC BY public copyright licence to any Author Accepted Manuscript version arising from this submission.

## Author contributions

R.B. and J.B. conceived the project, interpreted the data and wrote the manuscript with input from all authors. R.B. designed and performed experiments and data analysis. H.P. performed bioinformatic analysis. T.W. generated reagents. M.G. shared reagents, protocols and data unpublished at the time of initiating this study. V.M. provided advice and assistance with ATAC experiments. M.J.D. assisted with ATAC experiments and bioinformatic analyses and edited the manuscript.

## Competing interests
The authors declare no competing interests.

## Additional information
**Extended data** is available for this paper at https://doi.org/10.1038/s41556-022-00910-2.

**Correspondence and requests for materials** should be addressed to James Briscoe.

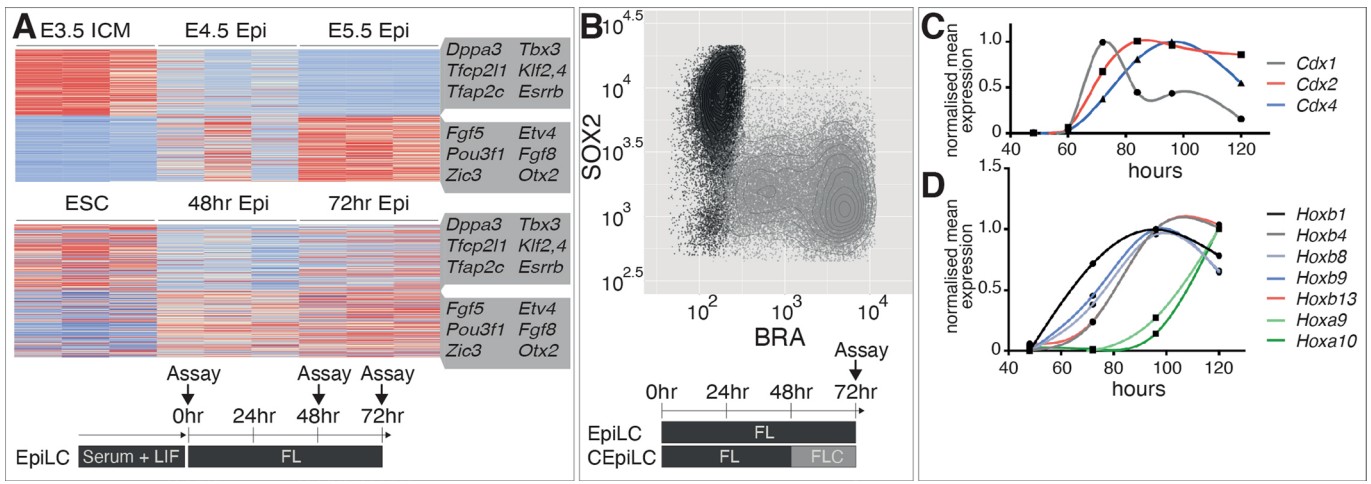

**Extended Data Fig. 1 | In vitro differentiated CEpiLCs recapitulate in vivo epiblast cell gene expression programmes. (a)** Epiblast-like cells differentiated in FL medium recapitulate in vivo gene-expression dynamics[65]. Illustrative genes for each cluster are highlighted. Gene expression profiles from 3 biological replicates are shown. See methods for details of genes plotted. **(b)** Flow cytometry analysis shows that culture in FLC medium reduces SOX2 levels and induces T/BRA. **(c)** Measurement of relative mRNA expression by RT-qPCR shows that FLC medium induces caudal epiblast markers *Cdx1,2,4*, and **(d)** posterior *Hox* genes. Normalised mean expression is shown relative to peak expression. Line is a loess fit to the data. The number of biological replicates averaged to fit each point are shown in Source numerical data available in source data.

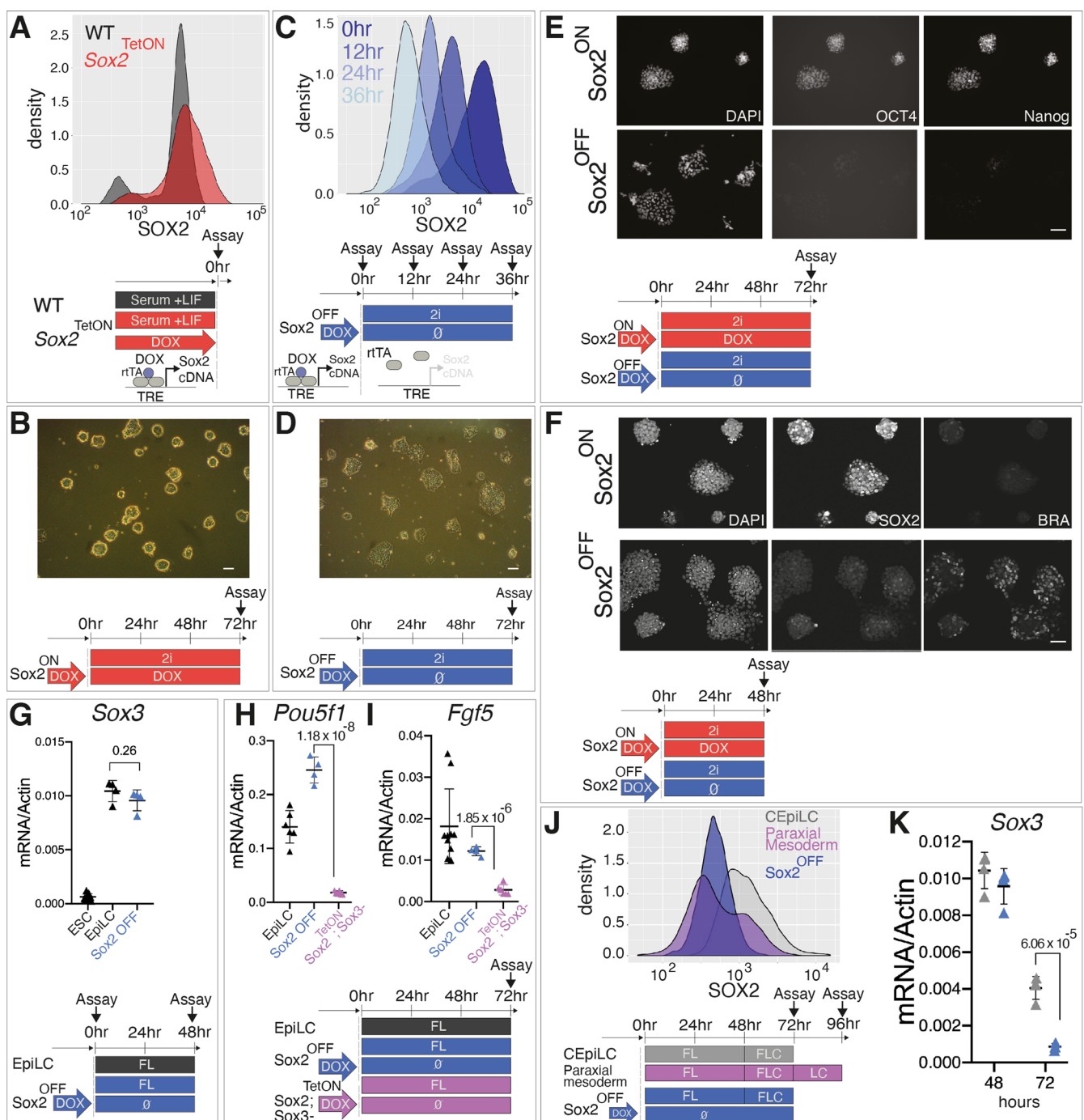

**Extended Data Fig. 2 | SOX2 maintains pluripotency of SOX2-TetON ES cells. (a)** Flow cytometry analysis shows that SOX2-TetON cultured in the presence of doxycycline (SOX2-ON) express SOX2 at comparable levels to pluripotent stem cells. **(b)** Brightfield migrograph showing that SOX2-ON maintain an undifferentiated morphology in '2i' medium. **(c)** Flow cytometry analysis shows that SOX2 levels are progressively reduced following removal of Dox from SOX2-TetON (SOX2-OFF). **(d)** Brightfield migrograph showing that SOX2-OFF cultured in '2i' medium lose their undifferentiated morphology. **(e)** Immunofluorescence image showing that SOX2-OFF cultured in '2i' lose expression of pluripotency markers OCT4 and NANOG and **(f)** induce the primitive streak marker T/BRA in the absence of SOX2 expression. Images shown in figures B, D, E, F are representative of 3 independent experiments. Scale bars represent 75uM. **(g)** Measurement of relative mRNA expression by RT-qPCR shows that SOX2-OFF induce expression of the EpiLC marker *Sox3* at similar levels to WT EpiLCs. Measurement of relative mRNA expression by RT-qPCR shows that ablation of *Sox3* in SOX2-OFF (SOX2-OFF, SOX3-) results in loss of expression of the pluripotency marker *Pou5f1* **(h)** and EpiLC marker *Fgf5* **(i)**. **(j)** FACS analysis shows that SOX2 levels in SOX2-OFF are intermediate to SOX2 low CEpiLCs and SOX2 -ve paraxial mesoderm progenitors. **(k)** Measurement of relative mRNA expression by RT-qPCR shows that *Sox3* levels are reduced in CHIR-stimulated SOX2-OFF compared to WT CEpiLCs. Each data point in G, H, I, and K represents an individual biological replicate. Bars denote mean ± s.e.m. *P* values calculated for differences of mean expression by two-tailed Student's *t*-test are shown. In G n=12 for ESC, n=4 for EpiLC, and n=4 for SOX2-OFF samples. In H n=6 for EpiLC, n=4 for SOX2-OFF, and n=6 for SOX2-OFF, SOX3- samples. In I n=10 for EpiLC, n=4 for SOX2-OFF, and n=6 for SOX2-OFF, SOX3- samples. In K n=4 for CEpiLC, and n=4 for SOX2-OFF samples. Source numerical data are available in source data.

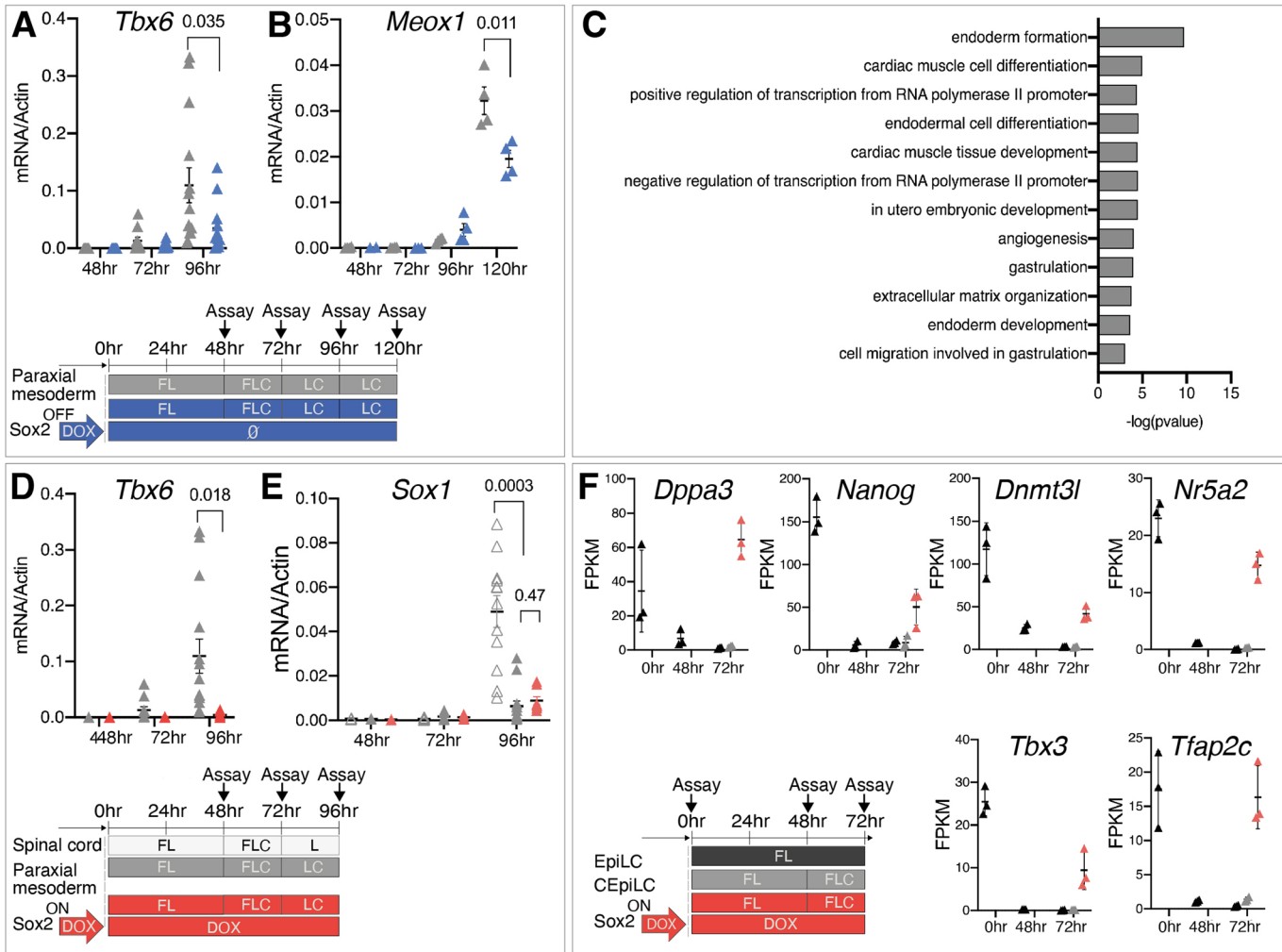

**Extended Data Fig. 3 | SOX2-OFF and SOX2-ON adopt distinct identities in response to WNT signaling.** Measurement of relative mRNA expression by RT-PCR shows that expression of **(a)** the paraxial mesoderm marker (*Tbx6*), and **(b)** the somitic mesoderm marker (*Meox1*), is reduced in SOX2-OFF compared to CEpiLCs. **(c)** GO analysis reveals that genes specifically upregulated in SOX2-OFF compared to EpiLC and CEpiLC (from clustering in Fig. 2c) are enriched for biological processes indicative of an early/anterior streak identity. **(d)** Measurement of relative mRNA expression by RT-qPCR shows that induction of the paraxial mesoderm marker (*Tbx6*) is repressed in SOX2-ON cultured in FLC medium. **(e)** Measurement of relative mRNA expression by RT-qPCR shows that *Sox1* is not induced in SOX2-ON cultured in FLC medium. **(f)** SOX2-ON re-express markers associated with pluripotency when cultured in FLC medium. Triplicate FPKM values for each gene are shown. In A, B, D, E each data point represents an individual biological replicate. Bars denote mean ± s.e.m. *P* values calculated for differences of mean expression by two-tailed Student's *t*-test are shown. In A n = 14 for 96hr samples. In B n = 4 for paraxial mesoderm and SOX2-OFF. In D n = 14 for CEpiLC and n = 8 for SOX2-ON at 96hr. In E n = 12 for Spinal cord, n = 14 for paraxial mesoderm, and n = 8 for SOX2-ON at 96hr. Source numerical data are available in source data.

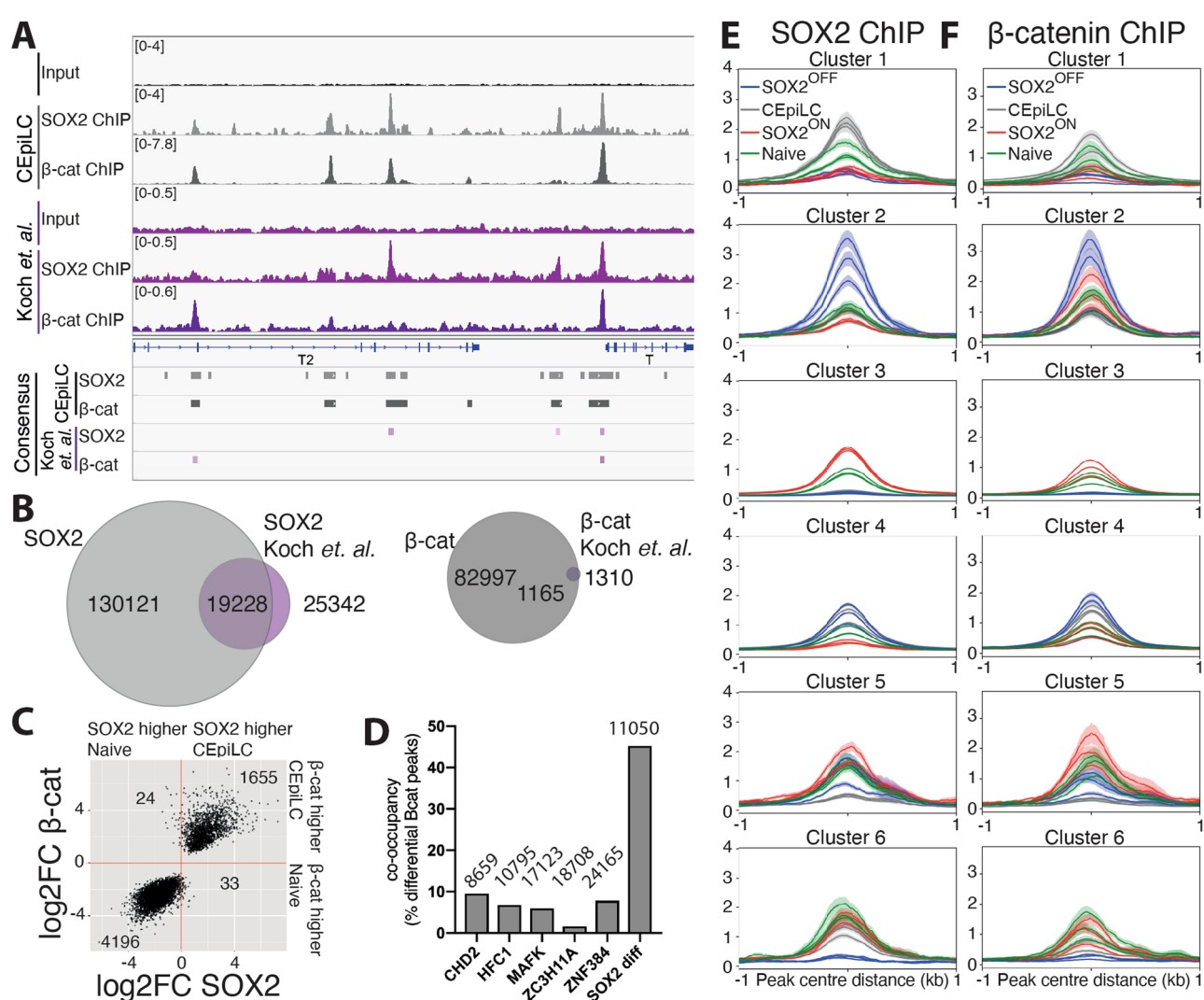

**Extended Data Fig. 4 | SOX2 and β-catenin co-occupy differential peaks. (a)** Comparison of SOX2 and β-catenin ChIP-seq peaks identified in data from this study with[12]. **(b)** Our consensus SOX2 and β-catenin peak sets include the majority of those identifiable from the data from[12], plus a large number of additional peaks. **(c)** Peaks differentially occupied by β-catenin (FDR <0.05) between ES cells and CEpiLCs exhibit a correlated change of SOX2 occupancy. FDR was determined by DESeq2 padj metric from n=3 biological replicates. log2FC calculated by DESeq2. **(d)** Differentially occupied β-catenin peaks overlap with differentially occupied SOX2 peaks to a greater extent than with peaks occupied by other ENCODE chromatin-associated proteins (ENCODE). Numbers of peaks in each set are show. **(e)** Metaplots of triplicate SOX2 and **(f)** β-catenin ChIP-seq signals at cell-type specific SOX2 bound CREs classified in Fig. 3e. Metaplots show mean ± s.e.m.

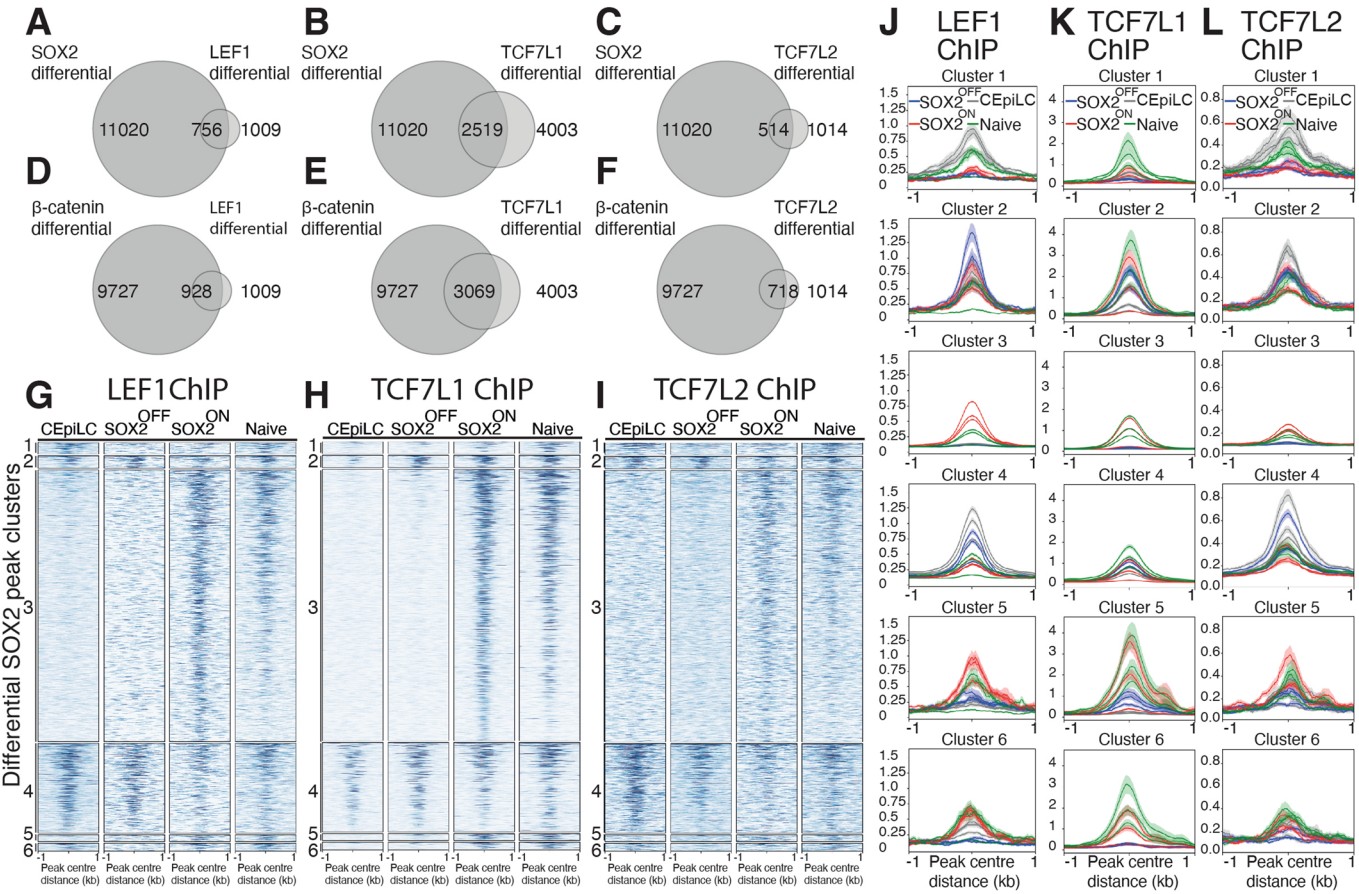

**Extended Data Fig. 5 | TCF/LEF are redistributed with SOX2 and β-catenin.** The majority of (**a**) LEF1, (**b**) TCF7L1, and (**c**) TCF7L2 differential peaks (FDR <0.05) overlap SOX2 differential peaks. FDR was determined by DESeq2 padj metric from n=3 biological replicates. The majority of (**d**) LEF1, (**e**) TCF7L1, and (**f**) TCF7L2 differential peaks (FDR <0.05) overlap β-catenin differential peaks. FDR was determined by DESeq2 padj metric from n=3 biological replicates. (**g**) LEF1, (**h**) TCF7L1, and (**i**) TCF7L2 exhibit a similar cell-type specific pattern of occupancy as β-catenin at SOX2 differential peaks classified in Fig. 3e (compare to Fig. 3e,f). Metaplots of triplicate (**j**) LEF1, (**k**) TCF7L1 and (**l**) TCF7L2 ChIP-seq signals at cell-type specific SOX2 bound peaks classified in Fig. 3e. Metaplots show mean ± s.e.m.

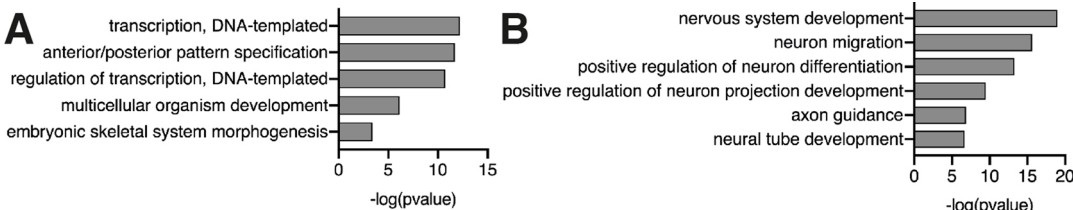

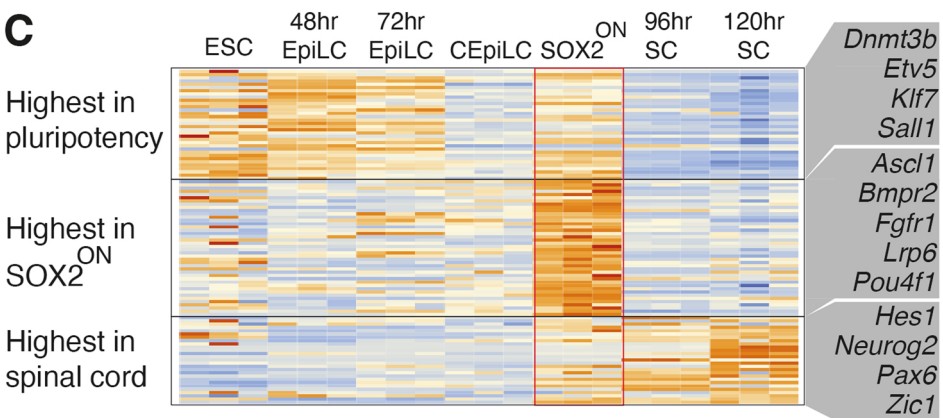

**Extended Data Fig. 6 | SOX2-ON express spinal-cord markers at low levels.** (**a**) Differentially expressed genes associated with CEpiLC specific cluster 1 CREs are enriched for biological processes underlying the patterning of the anterior-posterior axis. (**b**) Differentially expressed genes associated with SOX2-ON specific cluster 3 CREs are enriched for biological processes related to nervous system development. Differential expression criteria for genes analysed in A and B = FDR < 0.05 as determined by DESeq2 padj metric from n=3 biological replicates. (**c**) Cluster 3 associated genes with GO terms related to nervous system development from (**b**) that are expressed at high levels in spinal-cord (SC) neural progenitors exhibit comparatively low expression in SOX2-ON, ES cells, EpiLCs and CEpiLCs. Illustrative genes for each cluster are highlighted. Spinal cord progenitor data reanalysed from[9].

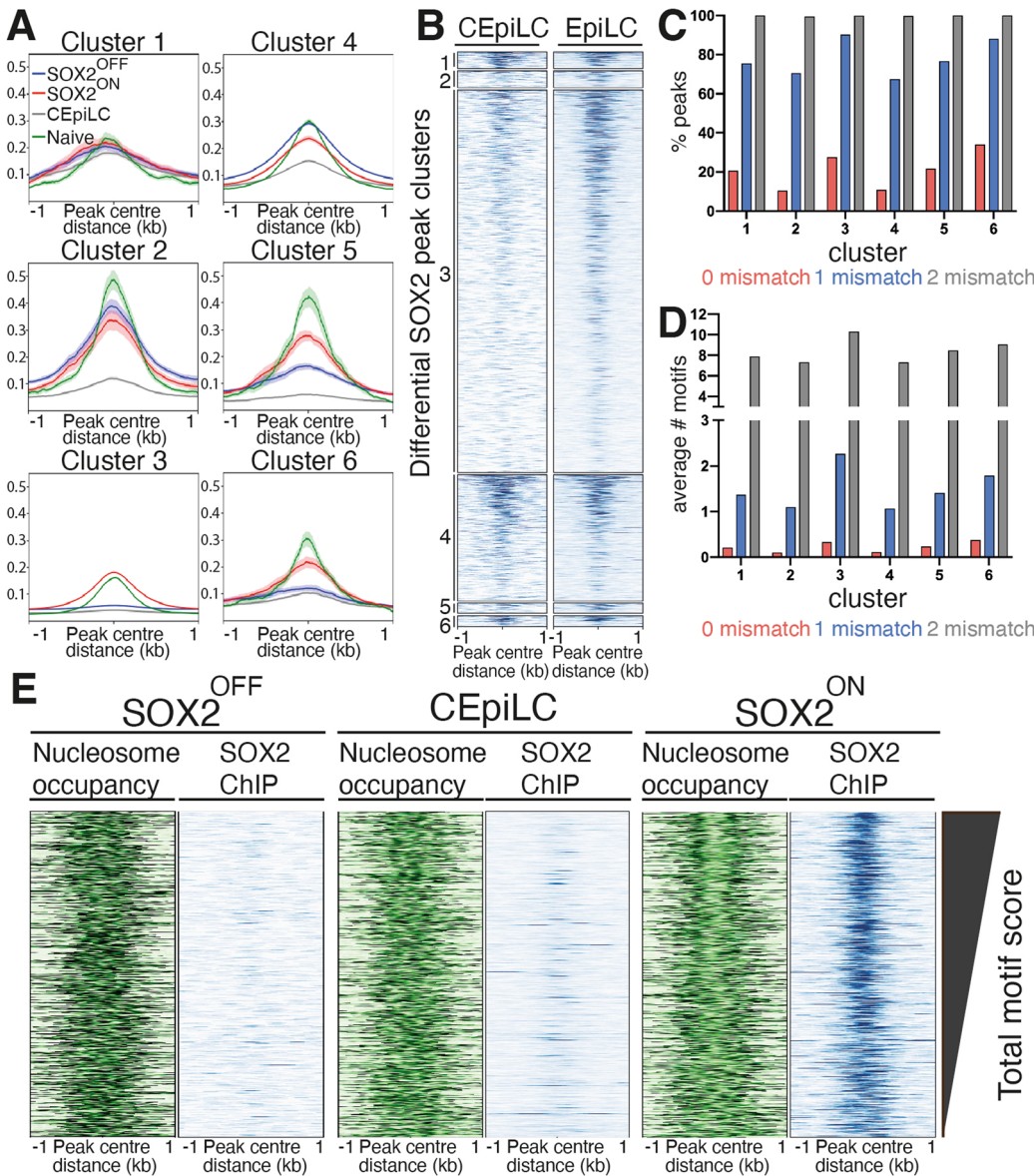

**Extended Data Fig. 7 | SOX2 promotes nucleosome eviction from peaks containing high scoring motifs.** (**a**) Metaplots of ATAC-seq data at cell-type specific SOX2 bound CREs classified in Fig. 3e. Naïve ATAC-seq data reanalysed from[63]. Metaplots show mean ± s.e.m. (**b**) SOX2 occupancy in CEpiLCs and EpiLCs at cell-type specific CREs classified in Fig. 3e. (**c**) Percentage of peaks from each cluster with at least 1 SOX2 motif with the indicated number of mismatches. (**d**) Average number of motifs per peak in each cluster with less than or equal to the indicated number of mismatches. (**e**) Nucleosome occupancy and SOX2 ChIP occupancy profiles of individual cluster 3 peaks ranked by their total FIMO motif score. Higher scoring peaks show a higher intensiy of SOX2 ChIP signal and a greater degree of nucleosome depletion at SOX2 peak centres in SOX2-ON.

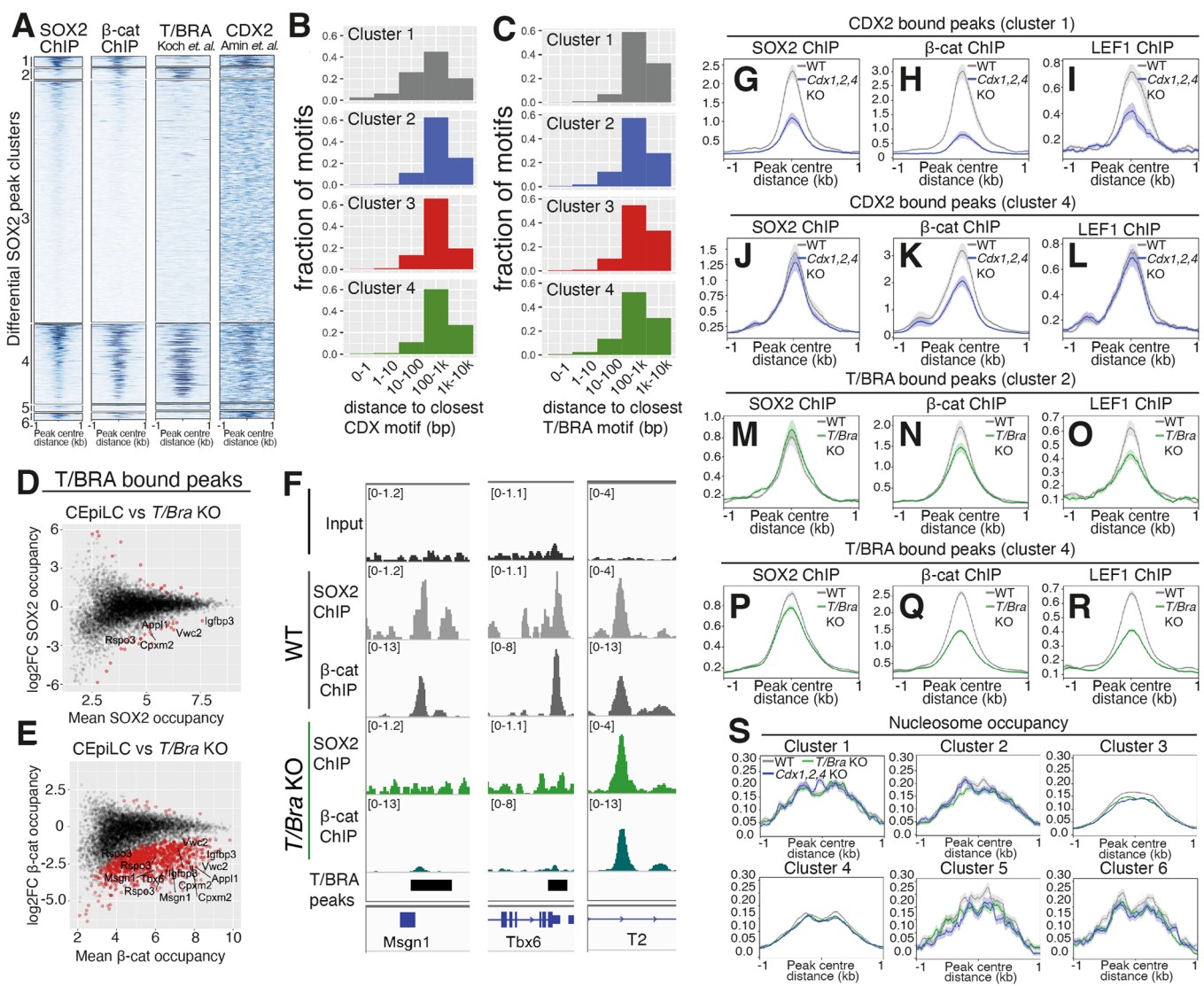

**Extended Data Fig. 8 | SOX2 and β-catenin co-occupy cell-type specific CREs with T/BRA and CDX2.** (**a**) SOX2 and β-catenin occupancy is correlated with T/BRA and CDX2 in CEpiLCs at cell-type specific CREs classified in Fig. 3e. Metaplots show mean ± s.e.m. (**b**) A sub-set of low-affinity SOX2 motifs within Cluster 1 SOX2 bound CREs are located in close proximity to CDX motifs. (**c**) A sub-set of low-affinity SOX2 motifs within Cluster 2 and Cluster 4 SOX2 bound CREs are located in close proximity to T/BRA motifs. Differential occupancy of (**d**) SOX2 and (**e**) β-catenin at T/BRA co-occupied peaks in T/BraKO progenitors compared to CEpiLCs. Differentially occupied peaks coloured red are statistically different between conditions (FDR <0.05). FDR was determined by DESeq2 padj metric from n=3 biological replicates. Labelled peaks are adjacent to genes differentially expressed in T/BraKO progenitors. (**f**) SOX2 and β-catenin occupancy is specifically reduced at CREs adjacent to paraxial mesoderm determinants in T/BraKO progenitors. Average SOX2 (**g**), β-catenin (**h**), and LEF1 (**i**) occupancy is reduced at cluster 1 CDX2 co-occupied peaks. Average SOX2 (**j**), β-catenin (**k**) and LEF1 (**l**) occupancy at cluster 4 CDX2 occupied peaks. Average (**m**) SOX2, (**n**) β-catenin, and (**o**) LEF1 at T/BRA co-occupied cluster 2 peaks in CEpiLCs and T/BraKO progenitors. Average (**p**) SOX2, (**q**) β-catenin, and (**r**) LEF1 at T/BRA co-occupied cluster 4 peaks in CEpiLCs and T/BraKO progenitors. (**s**) Average nucleosome occupancy is unchaged compared to CEpiLCs across all clusters of SOX differential peaks in CdxKO and T/BraKO progenitors (reanalysed from[35]).

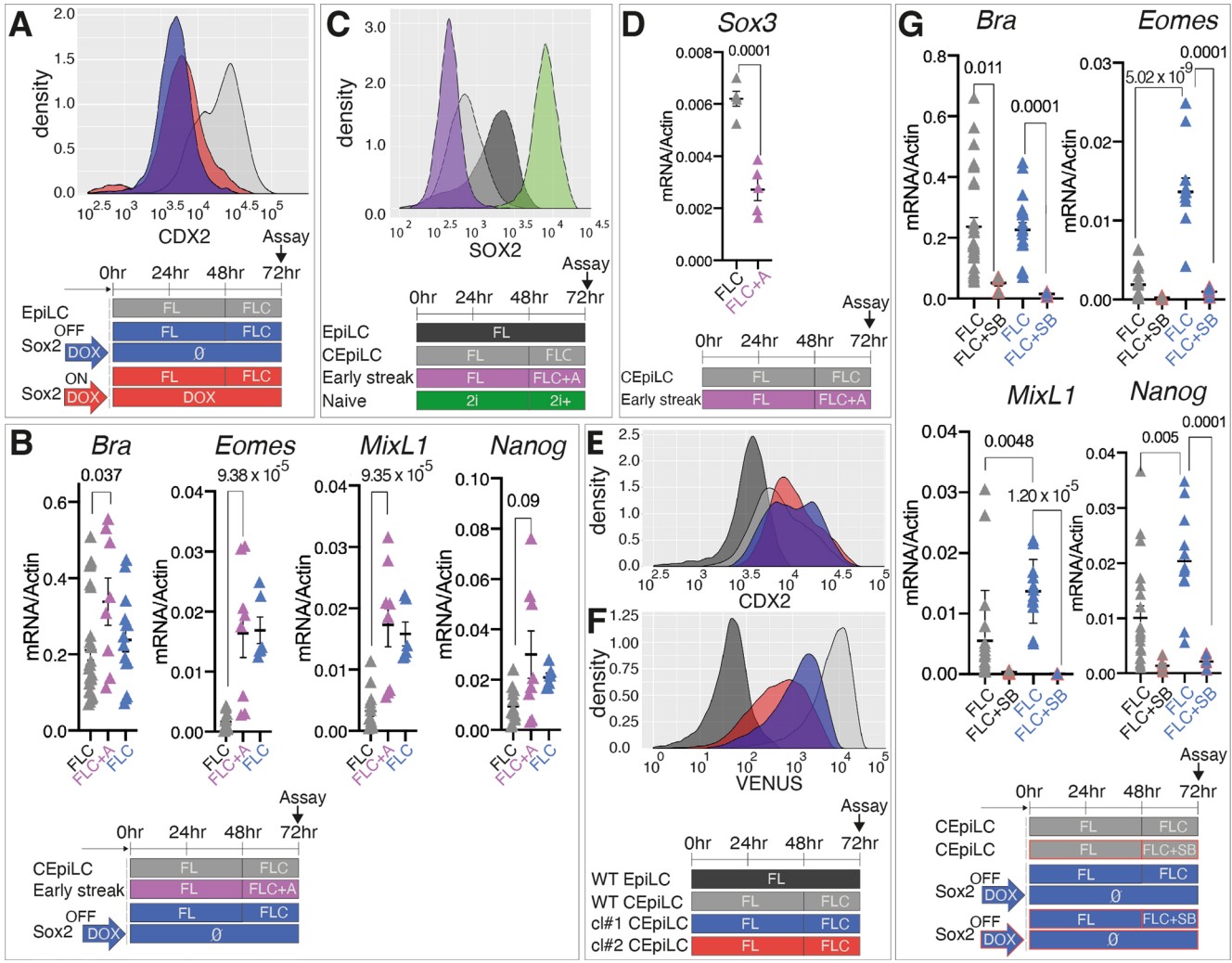

**Extended Data Fig. 9 | Repression of SOX2, *Sox3* and CDX2 expression in early-streak progenitors.** (**a**) Flow cytometry analysis shows that CDX2 expression is uniformly absent from SOX2-OFF and SOX2-ON cultured in FLC medium. (**b**) Measurement of relative mRNA expression by RT-qPCR shows that early primitive streak markers are induced to comparable levels by the addition of activin (10ng/ml) to FLC medium and in SOX2OFF relative to CEpiLCs. (**c**) Flow cytometry analysis shows SOX2 levels in EpiLCs, CEpiLCs, pluripotent and early streak progenitors. CHIR concentration was 5μM in 2i+ conditions. (**d**) Measurement of relative mRNA expression by RT-qPCR shows that *Sox3* levels are reduced in early-streak progenitors induced by the addition of activin to FLC medium. (**e**) Flow cytometry analysis shows the distribution of CDX2 expression is similar, whereas (**f**) Venus fluorescence is reduced in two Sox2del subclones analysed in Fig. 7f compared to the parental *Cdx2* CRE reporter line. (**g**) Measurement of relative mRNA expression by RT-PCR shows that primitive streak marker expression is reduced in CEpiLCs and SOX2-OFF cultured in FLC medium plus SB-431542. In B, D, and G each data point represents an individual biological replicate. Bars denote mean ± s.e.m. *P* values calculated for differences of mean expression by two-tailed Student's *t*-test are shown. In B for *Bra*, n = 26 for CEpiLCs and n = 8 for early streak progenitors; for *Eomes* n = 14 for CEpiLCs and n = 8 for early streak progenitors; for *MixL*1 n = 14 for CEpiLCs and n = 8 for early streak progenitors; for *Nanog* n = 14 for CEpiLCs and n = 8 for early streak progenitors. In D n = 5 for CEpiLCs and early streak progenitors. In G for *Bra*, n = 29 for CEpiLCs, n = 6 for SB treated CEpiLCs, n= 20 for SOX2-OFF, n = 6 for SB treated SOX2-OFF; for *Eomes* n = 20 for CEpiLCs, n = 6 for SB treated CEpiLCs, n= 12 for SOX2-OFF, n = 6 for SB treated SOX2-OFF; for *MixL1* n = 20 for CEpiLCs, n = 6 for SB treated CEpiLCs, n= 12 for SOX2-OFF, n = 6 for SB treated SOX2-OFF; for *Nanog* n = 20 for CEpiLCs, n = 6 for SB treated CEpiLCs, n= 12 for SOX2-OFF, n= 6 for SB treated SOX2-OFF. Source numerical data are available in source data.

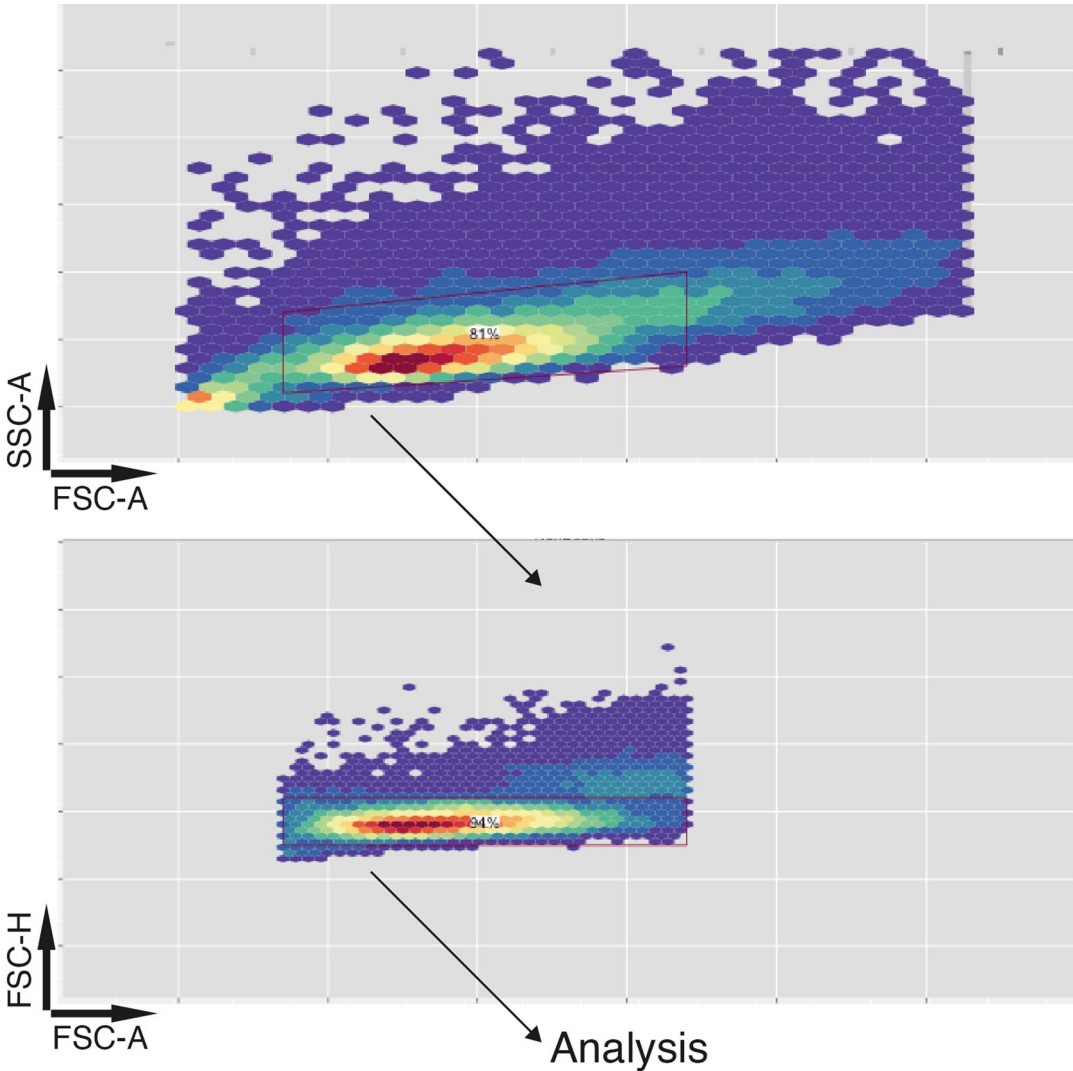

**Extended Data Fig. 10 | Representative FACS gating strategy used throughout the study.** Strategy applied to gate single cells plotted in Fig. 1f,h; Extended Data Figures 1B, 2A, 2C, 2J, 9A, 9C, 9E, 9F; and for analysis of average reporter activity in Fig. 7d–f.

# nature research

# Reporting Summary

Nature Research wishes to improve the reproducibility of the work that we publish. This form provides structure for consistency and transparency in reporting. For further information on Nature Research policies, see our Editorial Policies and the Editorial Policy Checklist.

## Statistics

For all statistical analyses, confirm that the following items are present in the figure legend, table legend, main text, or Methods section.

| n/a | Confirmed | |
|---|---|---|
| ☐ | ☒ | The exact sample size (*n*) for each experimental group/condition, given as a discrete number and unit of measurement |
| ☒ | ☐ | A statement on whether measurements were taken from distinct samples or whether the same sample was measured repeatedly |
| ☐ | ☒ | The statistical test(s) used AND whether they are one- or two-sided<br>*Only common tests should be described solely by name; describe more complex techniques in the Methods section.* |
| ☒ | ☐ | A description of all covariates tested |
| ☒ | ☐ | A description of any assumptions or corrections, such as tests of normality and adjustment for multiple comparisons |
| ☐ | ☒ | A full description of the statistical parameters including central tendency (e.g. means) or other basic estimates (e.g. regression coefficient) AND variation (e.g. standard deviation) or associated estimates of uncertainty (e.g. confidence intervals) |
| ☐ | ☒ | For null hypothesis testing, the test statistic (e.g. *F*, *t*, *r*) with confidence intervals, effect sizes, degrees of freedom and *P* value noted<br>*Give P values as exact values whenever suitable.* |
| ☒ | ☐ | For Bayesian analysis, information on the choice of priors and Markov chain Monte Carlo settings |
| ☒ | ☐ | For hierarchical and complex designs, identification of the appropriate level for tests and full reporting of outcomes |
| ☒ | ☐ | Estimates of effect sizes (e.g. Cohen's *d*, Pearson's *r*), indicating how they were calculated |

*Our web collection on statistics for biologists contains articles on many of the points above.*

## Software and code

Policy information about availability of computer code

| | |
|---|---|
| Data collection | Flow cytometry data was acquired using BD FACSDiva software. RT-PCR data was acquired using Applied Biosystems QuantStudio software. Immunofluorescence images were acquired using Zeiss Zen software. |
| Data analysis | A full list of all publicly available software used to analyse data is detailed in supplementary table 2 and below:<br><br>Applied Biosystems QuantStudio 12K Flex N/A<br>BAMTools (v2.5.1) Barnett et al., 2011<br>BD FACS DIVA (Version 8.0.1) N/A<br>bedGraphToBigWig (v1.0) Kent et al., 2010<br>BEDTools (v2.27.1) Quinlan and Hall, 2010<br>BWA (v0.7.17-r1188) Li and Durbin, 2009<br>cutadapt (version 1.16) Martin, 2011<br>DAVID (version 6.8) https://david.ncifcrf.gov/<br>deepTools (v3.2.1) Ramírez et al., 2016<br>DESeq2 (v1.20.0) Love et al., 2014<br>FastQC (v0.11.8) https://www.bioinformatics.babraham.ac.uk/projects/fastqc/<br>featureCounts (v1.6.4) Liao et al., 2014<br>Fiji (v2.0.0) Schindelin et. al. 2012<br>FIMO (v5.1.1) http://meme-suite.org/tools/fimo<br>flowcore (v1.44.2) Ellis et al., 2018<br>ggplot2 (3.3.0) Wickham, 2016<br>gplots (v3.1.0) Gregory et. al. 2015<br>HOMER (v4.10) Heinz et al., 2010<br>IGV genome browser (v2.4.13) Robinson et al., 2011 |

MACS (v2.1.2) Zhang et al., 2008
MultiQC (v1.7) Ewels et al., 2016
Nextflow (version 19.10.0) Di Tommaso et al., 2017
NucleoATAC (v0.3.4) Schep et al., 2015
phantompeakqualtools (v1.14) Landt et al., 2012
picard-tools (v2.19.0) http://broadinstitute.github.io/picard
Preseq (v2.0.3) http://smithlabresearch.org/software/preseq/
Prism (Version 8) N/A
Pysam (v0.15.2) https://github.com/pysam-developers/pysam
R (version 3.4.1) R Core Team, 2018
RSEM (version 1.3.0)  Li and Dewey, 2011
SAMtools (v1.9) Li et al., 2009
Singularity (version 2.6.0) Kurtzer et al., 2017
STAR (version 2.5.2a) Dobin et al., 2013
Trim Galore! (v0.5.0) https://www.bioinformatics.babraham.ac.uk/projects/trim_galore/

For RNAseq analysis adapter trimming was performed with cutadapt (version 1.16) with parameters "--minimum-length=25 --quality-cutoff=20 -a AGATCGGAAGAGC", and for paired-end data "-A AGATCGGAAGAGC" was appended to the command. The RSEM package (version 1.3.0)  in conjunction with the STAR alignment algorithm (version 2.5.2a) was used for the mapping and subsequent gene-level counting of the sequenced reads with respect to mm10 RefSeq genes downloaded from the UCSC Table Browser on 11th December 2017. The parameters passed to the "rsem-calculate-expression" command were "--star --star-gzipped-read-file --star-output-genome-bam --forward-prob 0", and for paired-end data "--paired-end" was appended to the command.

For ChIP-seq analysis the nf-core/ChIP-seq pipeline (version 1.1.0; https://doi.org/10.5281/zenodo.3529400) written in the Nextflow domain specific language (version 19.10.0) was used to perform the primary analysis of the samples in conjunction with Singularity (version 2.6.0). The command used was " nextflow run nf-core/ChIP-seq --input design.csv --genome mm10 --gtf refseq_genes.gtf --single_end --narrow_peak --min_reps_consensus 2 -profile crick -r 1.1.0". To summarise, the pipeline performs adapter trimming (Trim Galore!), read alignment (BWA) and filtering (SAMtools) ; (BEDTools); (BamTools); (pysam); (picard-tools), normalised coverage track generation (BEDTools); (bedGraphToBigWig), peak calling (MACS) (default q-value threshold < 0.05) and annotation relative to gene features (HOMER), consensus peak set creation (BEDTools), and read counting (featureCounts). Inclusion of a peak in the consensus peak set required that it be called by MACS in a minimum of 2/3 biological replicates from any of the four experimental conditions (CEpiLC, SOX2-OFF, SOX2-ON, Naïve ES). In all analyses, except for Fig 3C and Extended Data Fig 4C, the consensus peak set was derived from SOX2 peaks. For Fig 3C and Extended Data Fig 4C the consensus peak set comprised peaks from SOX2/B-catenin/TCF7L1/LEF1. All data was processed relative to the mouse UCSC mm10 genome (UCSC) downloaded from AWS iGenomes (https://github.com/ewels/AWS-iGenomes). Peak annotation was performed relative to the same GTF gene annotation file used for the RNA-seq analysis.

ATAC-seq analysis was performed using the nf-core/atacseq pipeline (version 1.0.0; https://doi.org/10.5281/zenodo.2634133), which uses similar processing steps as described for the nf-core/ChIP-seq pipeline with the addition of the removal of mitochondrial reads.

For nucleosome analysis the NucleoATAC package (version 0.3.4) (Schep et al., 2015) was run in default mode. Analysis was performed on all genomic intervals called as peaks from ATAC-seq data.

bigWig and bedGraph tracks of ChIPseq and NucleoATAC data were visualised using the IGV genome browser to illustrate representative peaks.

Differential analysis of ChIPseq and RNAseq read counts was performed with the DESeq2 package within the R programming environment (version 3.4.1). Log2FC and adjusted p-value of thresholds used for filtering results are indicated in legends.

SOX2 peaks were manually assigned to 6 clusters based upon differential occupancy between WT, 'SOX2 OFF', and 'SOX2 ON' samples. Peaks in clusters 1 ,2 and 3 had the highest mean read counts across biological triplicate samples in either WT, 'SOX2 OFF', or 'SOX2 ON' respectively, and were statistically different (FDR <0.05) as determined by DESeq2 in comparison to all other experimental conditions. Cluster 4, 5, and 6 peaks were statistically different to only one of the other experimental conditions. Bed files of genomic intervals defined by SOX2 peaks within these clusters were used to plot metaplots and heatmaps from the BigWig files generated from the nf-core/ChIP-seq and nf-core/ATAC-seq pipelines using deepTools, for motif enrichment analysis, and for motif scanning.

For GO analysis the online functional annotation tool of the DAVID bioinformatics resource https://david.ncifcrf.gov/summary.jsp was used with default parameters to identify statistically enriched biological process annotations within sets of gene IDs associated with differentially expressed transcripts, and to calculate associated Benjamini-hochberg adjusted p-values.

For motif scanning regions of +/-100bp adjacent to SOX2 ChIP-seq peak centres were used as inputs for the motif scanning tool http://meme-suite.org/tools/fimo. The Sox2 motif MA0143.3 (JASPAR) was used as a target. p-value threshold was set to p <0.1 so as to include low scoring SOX2 motifs present within peak sets.

Motifs enriched within each SOX2 peak cluster were identified using Homer findMotifsGenome using default parameters. Region size = 200b (+/-100bp adjacent to peak centre).

## Data

Policy information about availability of data

All manuscripts must include a data availability statement. This statement should provide the following information, where applicable:
- Accession codes, unique identifiers, or web links for publicly available datasets
- A list of figures that have associated raw data
- A description of any restrictions on data availability

Deep-sequencing (ChIP-seq, ATAC-seq, and RNA–seq) data generated during this study have been deposited in the Gene Expression Omnibus (GEO) under the accession code GSE162774. Previously published ChIP-seq, ATAC-seq, and RNA–seq data that were re-analysed during this study are available under accession codes GSE64059, GSE84899, GSE93524, E-MTAB-2268, E-MTAB-2958, E-MTAB-6337. All other data are available from the authors on reasonable request.

# Field-specific reporting

Please select the one below that is the best fit for your research. If you are not sure, read the appropriate sections before making your selection.

☒ Life sciences ☐ Behavioural & social sciences ☐ Ecological, evolutionary & environmental sciences

For a reference copy of the document with all sections, see nature.com/documents/nr-reporting-summary-flat.pdf

# Life sciences study design

All studies must disclose on these points even when the disclosure is negative.

| | |
|---|---|
| Sample size | Sample size of RNAseq and ChIPseq experiments was based on previous experience gained during the studies of Metzis et. al. 2018, Gouti et. al. 2014 & 2017, and other similar studies analysed by the Crick Institute Bioinformatics and Biostatistics group. 3 biological replicates were deemed the minimum necessary to provide sensitivity for detection of differential features by DESeq2 analysis. For qRT-PCR assays all samples collected over the course of exploratory and validatory stages of the study were aggregated for statistical analysis. |
| Data exclusions | Melt-curve analysis and extreme cT values were used as a basis for exclusion outliers of qPCR technical replicates prior to statistical analysis of data. |
| Replication | All experimental findings reported were reproducible. Experimental data presented in this study was obtained from a minimum of three independent experiments, each containing one or more biological replicates. This allowed evaluation of both within- and between experiment variability. Multiple RT-PCR and flow cytometry experiments were performed to confirm the robustness of experimental observations before proceeding to perform comprehensive analysis using next-generation sequencing assays. |
| Randomization | No randomisation was used to allocate cell culture samples to groups when comparisons were made between cell lines of different genotypes. When cell culture samples of the same genotype were compared for their response to a compound or cytokine, multiple samples were plated and assigned randomly to treatment or control groups. |
| Blinding | Investigators were not blinded during data collection or analysis. Experiments in this study required frequent intervention by the researcher to maintain cell cultures that precluded effective blinding. |

# Reporting for specific materials, systems and methods

We require information from authors about some types of materials, experimental systems and methods used in many studies. Here, indicate whether each material, system or method listed is relevant to your study. If you are not sure if a list item applies to your research, read the appropriate section before selecting a response.

## Materials & experimental systems

| n/a | Involved in the study |
|---|---|
| ☐ | ☒ Antibodies |
| ☐ | ☒ Eukaryotic cell lines |
| ☒ | ☐ Palaeontology and archaeology |
| ☒ | ☐ Animals and other organisms |
| ☒ | ☐ Human research participants |
| ☒ | ☐ Clinical data |
| ☒ | ☐ Dual use research of concern |

## Methods

| n/a | Involved in the study |
|---|---|
| ☐ | ☒ ChIP-seq |
| ☐ | ☒ Flow cytometry |
| ☒ | ☐ MRI-based neuroimaging |

## Antibodies

| | |
|---|---|
| Antibodies used | All antibodies used in this study are detailed in supplemental table 1 and below: |

Mouse OCT3/4 Santa cruz Cat#sc-5279 RRID:AB_628051 1:500
Rabbit NANOG Abcam Cat#ab80892 RRID:AB_2150114 1:500
Goat T/BRA R&D Cat#AF2085 RRID:AB_2200235 1:500
Rabbit SOX2 Millipore Cat#AB5603 RRID:AB_2286686 1:500
Mouse Sox2 V450 BD Cat#561610 RRID:AB_10712763 1:100
Mouse CDX2 647 BD Cat#560395 RRID:AB_1645405 1:20
Mouse CDX2 PE BD Cat#563428 RRID:AB_2738198 1:50
Goat Bra 488 R&D Cat#IC2085G 1:100
Goat SOX2 R&D Cat#AF2018 RRID:AB_355110 Lot#KOY0316101 5ug/IP
Rabbit B-Catenin Invitrogen Cat#71-2700 RRID:AB_2533982 Lot#SH2565754 5ug/IP
Goat TCF3 Santa Cruz  Cat#sc-8635 RRID:AB_2199133 Lot#L1415 5ug/IP
Goat TCF4 Santa Cruz  Cat#sc-8631 RRID:AB_2199826 Lot#K0615 5ug/IP
Goat LEF1 Millipore Cat#CS200635 Lot#2816642 3ug/IP
Donkey anti-Rabbit IgG AlexaFluor 647, ThermoFisher, Cat#A31573, 1:1000
Donkey anti-Goat IgG AlexaFluor 647, ThermoFisher, Cat#A21447, 1:1000
Donkey anti-Rabbit IgG AlexaFluor 488, ThermoFisher, Cat#A21206, 1:1000
Donkey anti-Mouse IgG AlexaFluor 488, ThermoFisher, Cat#A21202, 1:1000

| Validation | Antibodies used in this study are established reagents and have been validated in previous studies. References and links to manufacturers websites are provided below: |
|---|---|

Mouse OCT3/4 PMID: 35143761 https://www.scbt.com/p/oct-3-4-antibody-c-10
Rabbit NANOG PMID: 33472064 https://www.abcam.com/nanog-antibody-ab80892
Goat T/BRA PMID: 25157815 https://www.rndsystems.com/products/human-mouse-brachyury-antibody_af2085
Rabbit SOX2 PMID: 25115237 https://www.merckmillipore.com/GB/en/product/Anti-Sox2-Antibody,MM_NF-AB5603
Mouse Sox2 V450 PMID: 34536382 https://www.bdbiosciences.com/en-gb/products/reagents/flow-cytometry-reagents/research-reagents/single-color-antibodies-ruo/v450-mouse-anti-sox2.561610
Mouse CDX2 647 PMID: 34536382 https://www.bdbiosciences.com/en-us/products/reagents/flow-cytometry-reagents/research-reagents/single-color-antibodies-ruo/alexa-fluor-647-mouse-anti-cdx-2.560395
Mouse CDX2 PE PMID: 34536382 https://www.bdbiosciences.com/en-eu/products/reagents/flow-cytometry-reagents/research-reagents/single-color-antibodies-ruo/pe-mouse-anti-human-cdx-2.563428
Mouse Bra 488 PMID: 34536382 https://www.rndsystems.com/products/human-mouse-brachyury-alexa-fluor-488-conjugated-antibody_ic2085g
Goat SOX2 PMID: 28826820 https://www.rndsystems.com/products/human-mouse-rat-sox2-antibody_af2018
B-Catenin PMID: 28826820 https://www.thermofisher.com/antibody/product/beta-Catenin-Antibody-clone-CAT-15-Polyclonal/71-2700
TCF3 PMID: 18347094 https://www.scbt.com/p/tcf-3-antibody-m-20
TCF4 PMID: 24413017 https://www.scbt.com/p/tcf-4-antibody-n-20
LEF1 PMID: 24413017 https://www.merckmillipore.com/GB/en/product/ChIPAb-LEF1-ChIP-Validated-Antibody-and-Primer-Set,MM_NF-17-604

# Eukaryotic cell lines

Policy information about cell lines

| Cell line source(s) | The embryonic stem cell line, HM1 TetON, used in this study was first described in Serafimidis et. al. and were a gift from Anthony Gavalas to James Briscoe. |
|---|---|
| Authentication | Karyotyping was performed to verify genome integrity. |
| Mycoplasma contamination | All cells used in this study were routinely tested for Mycoplasma. No mycoplasma infected cells were used in this study. |
| Commonly misidentified lines (See ICLAC register) | No lines from this register were used in the study. |

# ChIP-seq

## Data deposition

☒ Confirm that both raw and final processed data have been deposited in a public database such as GEO.

☒ Confirm that you have deposited or provided access to graph files (e.g. BED files) for the called peaks.

| Data access links *May remain private before publication.* | https://www.ncbi.nlm.nih.gov/geo/query/acc.cgi?acc=GSE162774 |
|---|---|
| Files in database submission | WT_NAIVE_BCATENIN_IP_R1.fastq.gz WT_NAIVE_BCATENIN_IP_R2.fastq.gz WT_NAIVE_BCATENIN_IP_R3.fastq.gz WT_NAIVE_INPUT_R1.fastq.gz WT_NAIVE_LEF1_IP_R1.fastq.gz WT_NAIVE_LEF1_IP_R2.fastq.gz WT_NAIVE_LEF1_IP_R3.fastq.gz WT_NAIVE_SOX2_IP_R1.fastq.gz |

```
WT_NAIVE_SOX2_IP_R2.fastq.gz
WT_NAIVE_SOX2_IP_R3.fastq.gz
WT_NAIVE_TCF3_IP_R1.fastq.gz
WT_NAIVE_TCF3_IP_R2.fastq.gz
WT_NAIVE_TCF3_IP_R3.fastq.gz
WT_NAIVE_TCF4_IP_R1.fastq.gz
WT_NAIVE_TCF4_IP_R2.fastq.gz
WT_NAIVE_TCF4_IP_R3.fastq.gz
D3_SOX2_CHIR_FGF_BCATENIN_IP_R1.fastq.gz
D3_SOX2_CHIR_FGF_BCATENIN_IP_R2.fastq.gz
D3_SOX2_CHIR_FGF_BCATENIN_IP_R3.fastq.gz
D3_SOX2_CHIR_FGF_DOX_BCATENIN_IP_R1.fastq.gz
D3_SOX2_CHIR_FGF_DOX_BCATENIN_IP_R2.fastq.gz
D3_SOX2_CHIR_FGF_DOX_BCATENIN_IP_R3.fastq.gz
D3_SOX2_CHIR_FGF_DOX_INPUT_R1.fastq.gz
D3_SOX2_CHIR_FGF_DOX_LEF1_IP_R1.fastq.gz
D3_SOX2_CHIR_FGF_DOX_LEF1_IP_R2.fastq.gz
D3_SOX2_CHIR_FGF_DOX_LEF1_IP_R3.fastq.gz
D3_SOX2_CHIR_FGF_DOX_SOX2_IP_R1.fastq.gz
D3_SOX2_CHIR_FGF_DOX_SOX2_IP_R2.fastq.gz
D3_SOX2_CHIR_FGF_DOX_SOX2_IP_R3.fastq.gz
D3_SOX2_CHIR_FGF_DOX_TCF3_IP_R1.fastq.gz
D3_SOX2_CHIR_FGF_DOX_TCF3_IP_R2.fastq.gz
D3_SOX2_CHIR_FGF_DOX_TCF3_IP_R3.fastq.gz
D3_SOX2_CHIR_FGF_DOX_TCF4_IP_R1.fastq.gz
D3_SOX2_CHIR_FGF_DOX_TCF4_IP_R2.fastq.gz
D3_SOX2_CHIR_FGF_DOX_TCF4_IP_R3.fastq.gz
D3_SOX2_CHIR_FGF_INPUT_R1.fastq.gz
D3_SOX2_CHIR_FGF_LEF1_IP_R1.fastq.gz
D3_SOX2_CHIR_FGF_LEF1_IP_R2.fastq.gz
D3_SOX2_CHIR_FGF_LEF1_IP_R3.fastq.gz
D3_SOX2_CHIR_FGF_SOX2_IP_R1.fastq.gz
D3_SOX2_CHIR_FGF_SOX2_IP_R2.fastq.gz
D3_SOX2_CHIR_FGF_SOX2_IP_R3.fastq.gz
D3_SOX2_CHIR_FGF_TCF3_IP_R1.fastq.gz
D3_SOX2_CHIR_FGF_TCF3_IP_R2.fastq.gz
D3_SOX2_CHIR_FGF_TCF3_IP_R3.fastq.gz
D3_SOX2_CHIR_FGF_TCF4_IP_R1.fastq.gz
D3_SOX2_CHIR_FGF_TCF4_IP_R2.fastq.gz
D3_SOX2_CHIR_FGF_TCF4_IP_R3.fastq.gz
D3_WT_CHIR_FGF_BCATENIN_IP_R1.fastq.gz
D3_WT_CHIR_FGF_BCATENIN_IP_R2.fastq.gz
D3_WT_CHIR_FGF_BCATENIN_IP_R3.fastq.gz
D3_WT_CHIR_FGF_INPUT_R1.fastq.gz
D3_WT_CHIR_FGF_LEF1_IP_R1.fastq.gz
D3_WT_CHIR_FGF_LEF1_IP_R2.fastq.gz
D3_WT_CHIR_FGF_LEF1_IP_R3.fastq.gz
D3_WT_CHIR_FGF_SOX2_IP_R1.fastq.gz
D3_WT_CHIR_FGF_SOX2_IP_R2.fastq.gz
D3_WT_CHIR_FGF_SOX2_IP_R3.fastq.gz
D3_WT_CHIR_FGF_TCF3_IP_R1.fastq.gz
D3_WT_CHIR_FGF_TCF3_IP_R2.fastq.gz
D3_WT_CHIR_FGF_TCF3_IP_R3.fastq.gz
D3_WT_CHIR_FGF_TCF4_IP_R1.fastq.gz
D3_WT_CHIR_FGF_TCF4_IP_R2.fastq.gz
D3_WT_CHIR_FGF_TCF4_IP_R3.fastq.gz
D3_BRA_CHIR_FGF_BCATENIN_IP_R1.fastq.gz
D3_BRA_CHIR_FGF_BCATENIN_IP_R2.fastq.gz
D3_BRA_CHIR_FGF_BCATENIN_IP_R3.fastq.gz
D3_BRA_CHIR_FGF_INPUT_R1.fastq.gz
D3_BRA_CHIR_FGF_INPUT_R2.fastq.gz
D3_BRA_CHIR_FGF_INPUT_R3.fastq.gz
D3_BRA_CHIR_FGF_LEF1_IP_R1.fastq.gz
D3_BRA_CHIR_FGF_LEF1_IP_R2.fastq.gz
D3_BRA_CHIR_FGF_LEF1_IP_R3.fastq.gz
D3_BRA_CHIR_FGF_SOX2_IP_R1.fastq.gz
D3_BRA_CHIR_FGF_SOX2_IP_R2.fastq.gz
D3_BRA_CHIR_FGF_SOX2_IP_R3.fastq.gz
D3_CDX_CHIR_FGF_BCATENIN_IP_R1.fastq.gz
D3_CDX_CHIR_FGF_BCATENIN_IP_R2.fastq.gz
D3_CDX_CHIR_FGF_BCATENIN_IP_R3.fastq.gz
D3_CDX_CHIR_FGF_INPUT_R1.fastq.gz
D3_CDX_CHIR_FGF_INPUT_R2.fastq.gz
D3_CDX_CHIR_FGF_INPUT_R3.fastq.gz
D3_CDX_CHIR_FGF_LEF1_IP_R1.fastq.gz
D3_CDX_CHIR_FGF_LEF1_IP_R2.fastq.gz
```

```
D3_CDX_CHIR_FGF_LEF1_IP_R3.fastq.gz
D3_CDX_CHIR_FGF_SOX2_IP_R1.fastq.gz
D3_CDX_CHIR_FGF_SOX2_IP_R2.fastq.gz
D3_CDX_CHIR_FGF_SOX2_IP_R3.fastq.gz
D3_WT_CHIR_FGF_BCATENIN_IP_R1.fastq.gz
D3_WT_CHIR_FGF_BCATENIN_IP_R2.fastq.gz
D3_WT_CHIR_FGF_BCATENIN_IP_R3.fastq.gz
D3_WT_CHIR_FGF_INPUT_R1.fastq.gz
D3_WT_CHIR_FGF_INPUT_R2.fastq.gz
D3_WT_CHIR_FGF_INPUT_R3.fastq.gz
D3_WT_CHIR_FGF_LEF1_IP_R1.fastq.gz
D3_WT_CHIR_FGF_LEF1_IP_R2.fastq.gz
D3_WT_CHIR_FGF_LEF1_IP_R3.fastq.gz
D3_WT_CHIR_FGF_SOX2_IP_R1.fastq.gz
D3_WT_CHIR_FGF_SOX2_IP_R2.fastq.gz
D3_WT_CHIR_FGF_SOX2_IP_R3.fastq.gz
D3_WT_FGF_INPUT_R1.fastq.gz
D3_WT_FGF_INPUT_R2.fastq.gz
D3_WT_FGF_INPUT_R3.fastq.gz
D3_WT_FGF_SOX2_IP_R1.fastq.gz
D3_WT_FGF_SOX2_IP_R2.fastq.gz
D3_WT_FGF_SOX2_IP_R3.fastq.gz
WT_NAIVE_BCATENIN_IP_R1.narrowPeak.gz
WT_NAIVE_BCATENIN_IP_R2.narrowPeak.gz
WT_NAIVE_BCATENIN_IP_R3.narrowPeak.gz
WT_NAIVE_LEF1_IP_R1.narrowPeak.gz
WT_NAIVE_LEF1_IP_R2.narrowPeak.gz
WT_NAIVE_LEF1_IP_R3.narrowPeak.gz
WT_NAIVE_SOX2_IP_R1.narrowPeak.gz
WT_NAIVE_SOX2_IP_R2.narrowPeak.gz
WT_NAIVE_SOX2_IP_R3.narrowPeak.gz
WT_NAIVE_TCF3_IP_R1.narrowPeak.gz
WT_NAIVE_TCF3_IP_R2.narrowPeak.gz
WT_NAIVE_TCF3_IP_R3.narrowPeak.gz
WT_NAIVE_TCF4_IP_R1.narrowPeak.gz
WT_NAIVE_TCF4_IP_R2.narrowPeak.gz
WT_NAIVE_TCF4_IP_R3.narrowPeak.gz
D3_SOX2_CHIR_FGF_BCATENIN_IP_R1.narrowPeak.gz
D3_SOX2_CHIR_FGF_BCATENIN_IP_R2.narrowPeak.gz
D3_SOX2_CHIR_FGF_BCATENIN_IP_R3.narrowPeak.gz
D3_SOX2_CHIR_FGF_DOX_BCATENIN_IP_R1.narrowPeak.gz
D3_SOX2_CHIR_FGF_DOX_BCATENIN_IP_R2.narrowPeak.gz
D3_SOX2_CHIR_FGF_DOX_BCATENIN_IP_R3.narrowPeak.gz
D3_SOX2_CHIR_FGF_DOX_LEF1_IP_R1.narrowPeak.gz
D3_SOX2_CHIR_FGF_DOX_LEF1_IP_R2.narrowPeak.gz
D3_SOX2_CHIR_FGF_DOX_LEF1_IP_R3.narrowPeak.gz
D3_SOX2_CHIR_FGF_DOX_SOX2_IP_R1.narrowPeak.gz
D3_SOX2_CHIR_FGF_DOX_SOX2_IP_R2.narrowPeak.gz
D3_SOX2_CHIR_FGF_DOX_SOX2_IP_R3.narrowPeak.gz
D3_SOX2_CHIR_FGF_DOX_TCF3_IP_R1.narrowPeak.gz
D3_SOX2_CHIR_FGF_DOX_TCF3_IP_R2.narrowPeak.gz
D3_SOX2_CHIR_FGF_DOX_TCF3_IP_R3.narrowPeak.gz
D3_SOX2_CHIR_FGF_DOX_TCF4_IP_R1.narrowPeak.gz
D3_SOX2_CHIR_FGF_DOX_TCF4_IP_R2.narrowPeak.gz
D3_SOX2_CHIR_FGF_DOX_TCF4_IP_R3.narrowPeak.gz
D3_SOX2_CHIR_FGF_LEF1_IP_R1.narrowPeak.gz
D3_SOX2_CHIR_FGF_LEF1_IP_R2.narrowPeak.gz
D3_SOX2_CHIR_FGF_LEF1_IP_R3.narrowPeak.gz
D3_SOX2_CHIR_FGF_SOX2_IP_R1.narrowPeak.gz
D3_SOX2_CHIR_FGF_SOX2_IP_R2.narrowPeak.gz
D3_SOX2_CHIR_FGF_SOX2_IP_R3.narrowPeak.gz
D3_SOX2_CHIR_FGF_TCF3_IP_R1.narrowPeak.gz
D3_SOX2_CHIR_FGF_TCF3_IP_R2.narrowPeak.gz
D3_SOX2_CHIR_FGF_TCF3_IP_R3.narrowPeak.gz
D3_SOX2_CHIR_FGF_TCF4_IP_R1.narrowPeak.gz
D3_SOX2_CHIR_FGF_TCF4_IP_R2.narrowPeak.gz
D3_SOX2_CHIR_FGF_TCF4_IP_R3.narrowPeak.gz
D3_WT_CHIR_FGF_BCATENIN_IP_R1.narrowPeak.gz
D3_WT_CHIR_FGF_BCATENIN_IP_R2.narrowPeak.gz
D3_WT_CHIR_FGF_BCATENIN_IP_R3.narrowPeak.gz
D3_WT_CHIR_FGF_LEF1_IP_R1.narrowPeak.gz
D3_WT_CHIR_FGF_LEF1_IP_R2.narrowPeak.gz
D3_WT_CHIR_FGF_LEF1_IP_R3.narrowPeak.gz
D3_WT_CHIR_FGF_SOX2_IP_R1.narrowPeak.gz
D3_WT_CHIR_FGF_SOX2_IP_R2.narrowPeak.gz
D3_WT_CHIR_FGF_SOX2_IP_R3.narrowPeak.gz
```

```
D3_WT_CHIR_FGF_TCF3_IP_R1.narrowPeak.gz
D3_WT_CHIR_FGF_TCF3_IP_R2.narrowPeak.gz
D3_WT_CHIR_FGF_TCF3_IP_R3.narrowPeak.gz
D3_WT_CHIR_FGF_TCF4_IP_R1.narrowPeak.gz
D3_WT_CHIR_FGF_TCF4_IP_R2.narrowPeak.gz
D3_WT_CHIR_FGF_TCF4_IP_R3.narrowPeak.gz
WT_NAIVE_BCATENIN_IP_R1.bigWig
WT_NAIVE_BCATENIN_IP_R2.bigWig
WT_NAIVE_BCATENIN_IP_R3.bigWig
WT_NAIVE_INPUT_R1.bigWig
WT_NAIVE_LEF1_IP_R1.bigWig
WT_NAIVE_LEF1_IP_R2.bigWig
WT_NAIVE_LEF1_IP_R3.bigWig
WT_NAIVE_SOX2_IP_R1.bigWig
WT_NAIVE_SOX2_IP_R2.bigWig
WT_NAIVE_SOX2_IP_R3.bigWig
WT_NAIVE_TCF3_IP_R1.bigWig
WT_NAIVE_TCF3_IP_R2.bigWig
WT_NAIVE_TCF3_IP_R3.bigWig
WT_NAIVE_TCF4_IP_R1.bigWig
WT_NAIVE_TCF4_IP_R2.bigWig
WT_NAIVE_TCF4_IP_R3.bigWig
D3_SOX2_CHIR_FGF_BCATENIN_IP_R1.bigWig
D3_SOX2_CHIR_FGF_BCATENIN_IP_R2.bigWig
D3_SOX2_CHIR_FGF_BCATENIN_IP_R3.bigWig
D3_SOX2_CHIR_FGF_DOX_BCATENIN_IP_R1.bigWig
D3_SOX2_CHIR_FGF_DOX_BCATENIN_IP_R2.bigWig
D3_SOX2_CHIR_FGF_DOX_BCATENIN_IP_R3.bigWig
D3_SOX2_CHIR_FGF_DOX_INPUT_R1.bigWig
D3_SOX2_CHIR_FGF_DOX_LEF1_IP_R1.bigWig
D3_SOX2_CHIR_FGF_DOX_LEF1_IP_R2.bigWig
D3_SOX2_CHIR_FGF_DOX_LEF1_IP_R3.bigWig
D3_SOX2_CHIR_FGF_DOX_SOX2_IP_R1.bigWig
D3_SOX2_CHIR_FGF_DOX_SOX2_IP_R2.bigWig
D3_SOX2_CHIR_FGF_DOX_SOX2_IP_R3.bigWig
D3_SOX2_CHIR_FGF_DOX_TCF3_IP_R1.bigWig
D3_SOX2_CHIR_FGF_DOX_TCF3_IP_R2.bigWig
D3_SOX2_CHIR_FGF_DOX_TCF3_IP_R3.bigWig
D3_SOX2_CHIR_FGF_DOX_TCF4_IP_R1.bigWig
D3_SOX2_CHIR_FGF_DOX_TCF4_IP_R2.bigWig
D3_SOX2_CHIR_FGF_DOX_TCF4_IP_R3.bigWig
D3_SOX2_CHIR_FGF_INPUT_R1.bigWig
D3_SOX2_CHIR_FGF_LEF1_IP_R1.bigWig
D3_SOX2_CHIR_FGF_LEF1_IP_R2.bigWig
D3_SOX2_CHIR_FGF_LEF1_IP_R3.bigWig
D3_SOX2_CHIR_FGF_SOX2_IP_R1.bigWig
D3_SOX2_CHIR_FGF_SOX2_IP_R2.bigWig
D3_SOX2_CHIR_FGF_SOX2_IP_R3.bigWig
D3_SOX2_CHIR_FGF_TCF3_IP_R1.bigWig
D3_SOX2_CHIR_FGF_TCF3_IP_R2.bigWig
D3_SOX2_CHIR_FGF_TCF3_IP_R3.bigWig
D3_SOX2_CHIR_FGF_TCF4_IP_R1.bigWig
D3_SOX2_CHIR_FGF_TCF4_IP_R2.bigWig
D3_SOX2_CHIR_FGF_TCF4_IP_R3.bigWig
D3_WT_CHIR_FGF_BCATENIN_IP_R1.bigWig
D3_WT_CHIR_FGF_BCATENIN_IP_R2.bigWig
D3_WT_CHIR_FGF_BCATENIN_IP_R3.bigWig
D3_WT_CHIR_FGF_INPUT_R1.bigWig
D3_WT_CHIR_FGF_LEF1_IP_R1.bigWig
D3_WT_CHIR_FGF_LEF1_IP_R2.bigWig
D3_WT_CHIR_FGF_LEF1_IP_R3.bigWig
D3_WT_CHIR_FGF_SOX2_IP_R1.bigWig
D3_WT_CHIR_FGF_SOX2_IP_R2.bigWig
D3_WT_CHIR_FGF_SOX2_IP_R3.bigWig
D3_WT_CHIR_FGF_TCF3_IP_R1.bigWig
D3_WT_CHIR_FGF_TCF3_IP_R2.bigWig
D3_WT_CHIR_FGF_TCF3_IP_R3.bigWig
D3_WT_CHIR_FGF_TCF4_IP_R1.bigWig
D3_WT_CHIR_FGF_TCF4_IP_R2.bigWig
D3_WT_CHIR_FGF_TCF4_IP_R3.bigWig
D3_BRA_CHIR_FGF_BCATENIN_IP_R1.narrowPeak.gz
D3_BRA_CHIR_FGF_BCATENIN_IP_R2.narrowPeak.gz
D3_BRA_CHIR_FGF_BCATENIN_IP_R3.narrowPeak.gz
D3_BRA_CHIR_FGF_LEF1_IP_R1.narrowPeak.gz
D3_BRA_CHIR_FGF_LEF1_IP_R2.narrowPeak.gz
D3_BRA_CHIR_FGF_LEF1_IP_R3.narrowPeak.gz
```

D3_BRA_CHIR_FGF_SOX2_IP_R1.narrowPeak.gz
D3_BRA_CHIR_FGF_SOX2_IP_R2.narrowPeak.gz
D3_BRA_CHIR_FGF_SOX2_IP_R3.narrowPeak.gz
D3_CDX_CHIR_FGF_BCATENIN_IP_R1.narrowPeak.gz
D3_CDX_CHIR_FGF_BCATENIN_IP_R2.narrowPeak.gz
D3_CDX_CHIR_FGF_BCATENIN_IP_R3.narrowPeak.gz
D3_CDX_CHIR_FGF_LEF1_IP_R1.narrowPeak.gz
D3_CDX_CHIR_FGF_LEF1_IP_R2.narrowPeak.gz
D3_CDX_CHIR_FGF_LEF1_IP_R3.narrowPeak.gz
D3_CDX_CHIR_FGF_SOX2_IP_R1.narrowPeak.gz
D3_CDX_CHIR_FGF_SOX2_IP_R2.narrowPeak.gz
D3_CDX_CHIR_FGF_SOX2_IP_R3.narrowPeak.gz
D3_WT_CHIR_FGF_BCATENIN_IP_R1.narrowPeak.gz
D3_WT_CHIR_FGF_BCATENIN_IP_R2.narrowPeak.gz
D3_WT_CHIR_FGF_BCATENIN_IP_R3.narrowPeak.gz
D3_WT_CHIR_FGF_LEF1_IP_R1.narrowPeak.gz
D3_WT_CHIR_FGF_LEF1_IP_R2.narrowPeak.gz
D3_WT_CHIR_FGF_LEF1_IP_R3.narrowPeak.gz
D3_WT_CHIR_FGF_SOX2_IP_R1.narrowPeak.gz
D3_WT_CHIR_FGF_SOX2_IP_R2.narrowPeak.gz
D3_WT_CHIR_FGF_SOX2_IP_R3.narrowPeak.gz
D3_WT_FGF_SOX2_IP_R1.narrowPeak.gz
D3_WT_FGF_SOX2_IP_R2.narrowPeak.gz
D3_WT_FGF_SOX2_IP_R3.narrowPeak.gz
D3_BRA_CHIR_FGF_BCATENIN_IP_R1.bigWig
D3_BRA_CHIR_FGF_BCATENIN_IP_R2.bigWig
D3_BRA_CHIR_FGF_BCATENIN_IP_R3.bigWig
D3_BRA_CHIR_FGF_INPUT_R1.bigWig
D3_BRA_CHIR_FGF_INPUT_R2.bigWig
D3_BRA_CHIR_FGF_INPUT_R3.bigWig
D3_BRA_CHIR_FGF_LEF1_IP_R1.bigWig
D3_BRA_CHIR_FGF_LEF1_IP_R2.bigWig
D3_BRA_CHIR_FGF_LEF1_IP_R3.bigWig
D3_BRA_CHIR_FGF_SOX2_IP_R1.bigWig
D3_BRA_CHIR_FGF_SOX2_IP_R2.bigWig
D3_BRA_CHIR_FGF_SOX2_IP_R3.bigWig
D3_CDX_CHIR_FGF_BCATENIN_IP_R1.bigWig
D3_CDX_CHIR_FGF_BCATENIN_IP_R2.bigWig
D3_CDX_CHIR_FGF_BCATENIN_IP_R3.bigWig
D3_CDX_CHIR_FGF_INPUT_R1.bigWig
D3_CDX_CHIR_FGF_INPUT_R2.bigWig
D3_CDX_CHIR_FGF_INPUT_R3.bigWig
D3_CDX_CHIR_FGF_LEF1_IP_R1.bigWig
D3_CDX_CHIR_FGF_LEF1_IP_R2.bigWig
D3_CDX_CHIR_FGF_LEF1_IP_R3.bigWig
D3_CDX_CHIR_FGF_SOX2_IP_R1.bigWig
D3_CDX_CHIR_FGF_SOX2_IP_R2.bigWig
D3_CDX_CHIR_FGF_SOX2_IP_R3.bigWig
D3_WT_CHIR_FGF_BCATENIN_IP_R1.bigWig
D3_WT_CHIR_FGF_BCATENIN_IP_R2.bigWig
D3_WT_CHIR_FGF_BCATENIN_IP_R3.bigWig
D3_WT_CHIR_FGF_INPUT_R1.bigWig
D3_WT_CHIR_FGF_INPUT_R2.bigWig
D3_WT_CHIR_FGF_INPUT_R3.bigWig
D3_WT_CHIR_FGF_LEF1_IP_R1.bigWig
D3_WT_CHIR_FGF_LEF1_IP_R2.bigWig
D3_WT_CHIR_FGF_LEF1_IP_R3.bigWig
D3_WT_CHIR_FGF_SOX2_IP_R1.bigWig
D3_WT_CHIR_FGF_SOX2_IP_R2.bigWig
D3_WT_CHIR_FGF_SOX2_IP_R3.bigWig
D3_WT_FGF_INPUT_R1.bigWig
D3_WT_FGF_INPUT_R2.bigWig
D3_WT_FGF_INPUT_R3.bigWig
D3_WT_FGF_SOX2_IP_R1.bigWig
D3_WT_FGF_SOX2_IP_R2.bigWig
D3_WT_FGF_SOX2_IP_R3.bigWig

Genome browser session
(e.g. UCSC)

n/a

## Methodology

Replicates

Three biological replicates for each experimental condition were collected from independent experiments. Samples from each independent set of experimental samples was immunoprecipitated together. Only peaks common to 2/3 replicates per group were carried forward for consensus peak generation and the subsequent differential analysis.

| Sequencing depth | Single-end, 76bp reads were sequenced on the Illumina HiSeq 4000 platform with an average yield of ~30 million reads per sample and ~91% alignment rate. |
|---|---|
| Antibodies | Goat SOX2 R&D Cat#AF2018 RRID:AB_355110 Lot#KOY0316101 5ug/IP<br>Rabbit B-Catenin Invitrogen Cat#71-2700 RRID:AB_2533982 Lot#SH2565754 5ug/IP<br>Goat TCF3 Santa Cruz  Cat#sc-8635 RRID:AB_2199133 Lot#L1415 5ug/IP<br>Goat TCF4 Santa Cruz  Cat#sc-8631 RRID:AB_2199826 Lot#K0615 5ug/IP<br>Goat LEF1 Millipore Cat#CS200635 Lot#2816642 3ug/IP |
| Peak calling parameters | BWA MEM version 0.7.17 was used to map the reads to the mouse mm10 genome assembly with the command: "bwa mem -M <GENOME_INDEX> <INPUT_FASTQ> \| samtools view -b -h -F 0x0100 -O BAM -o <OUTPUT_BAM> -"<br>MACS2 version 2.1.2 was used to call peaks with the command: "macs2 callpeak -t <IP_BAM> -c <INPUT_BAM> -f BAM -g 1.87e9 -n <OUTPUT_PREFIX> --keep-dup all" |
| Data quality | The nf-core/chipseq pipeline generated extensive QC metrics that are collated and rendered in a MultiQC report at the end of the pipeline. Samples were assessed for total number of reads, duplication rate, total number of peaks and FRiP score. As a stringency filter, only peaks common to 2/3 replicates per group were carried forward for consensus peak generation and the subsequent differential analysis. |
| Software | The nf-core/chipseq pipeline (version 1.1.0; Ewels et al., 2020; https://doi.org/10.5281/zenodo.3529400) written in the Nextflow domain specific language (version 19.10.0; Di Tommaso et al., 2017) was used to perform the primary analysis of the samples in conjunction with Singularity (version 2.6.0; Kurtzer et al., 2017). The command used was "nextflow run nf-core/chipseq --input design.csv --genome mm10 --gtf refseq_genes.gtf --single_end --narrow_peak --min_reps_consensus 2 -profile crick -r 1.1.0" |

# Flow Cytometry

## Plots

Confirm that:

☒ The axis labels state the marker and fluorochrome used (e.g. CD4-FITC).

☒ The axis scales are clearly visible. Include numbers along axes only for bottom left plot of group (a 'group' is an analysis of identical markers).

☒ All plots are contour plots with outliers or pseudocolor plots.

☒ A numerical value for number of cells or percentage (with statistics) is provided.

## Methodology

| Sample preparation | A detailed protocol for sample preparation is provided in the Methods section. |
|---|---|
| Instrument | Becton Dickinson (BD) LSR Fortessa |
| Software | Flow cytometry data was acquired using BD FACSDiva software, and analysed in the R programming environment using the flowCore package. |
| Cell population abundance | No cell-sorting was performed in this study. Flow cytometry was used for quantification only. |
| Gating strategy | Gates applied in every experiment: SSC-A/FSC-A gate was used to remove debris and dead cells; FSC-A/FSC-H was used to remove cell doublets |

☒ Tick this box to confirm that a figure exemplifying the gating strategy is provided in the Supplementary Information.

