## [Peer Review File · Nature Cell Biology]

Peer Review Information

Journal: Nature Cell Biology

Manuscript Title: Sox2 levels configure the WNT response of epiblast progenitors responsible for vertebrate body formation

Corresponding author name(s): James Briscoe

Reviewer Comments & Decisions:

Decision Letter, initial version:

Dear James,

Your manuscript, "Sox2 levels configure the WNT response of epiblast progenitors responsible for vertebrate body formation", has now been seen by 3 referees, who are experts in pioneer factors, nucleosome, chromatin (referee 1); ESCs, pluripotency, chromatin (referee 2); and pluripotency, signalling, chromatin (referee 3). As you will see from their comments (attached below) they find this work of potential interest, but have raised substantial concerns, which in our view would need to be addressed with considerable revisions before we can consider publication in Nature Cell Biology.

Nature Cell Biology editors discuss the referee reports in detail within the editorial team, including the chief editor, to identify key referee points that should be addressed with priority, and requests that are overruled as being beyond the scope of the current study. To guide the scope of the revisions, I have listed these points below. We are committed to providing a fair and constructive peer-review process, so please feel free to contact me if you would like to discuss any of the referee comments further.

In particular, it would be essential to address the following points:

(A) Mechanistic insights relating to SOX2 binding sites, expression levels and Tcf/beta-catenin binding should be further strengthened by providing additional experiments and causal links, as advised by referees 1 and 2.

Referee 1 notes:

** The idea that due to a biochemical equilibrium, the "pioneering" activity of a transcription factor would be a function of its expression level has been proposed before (e.g. by Workman and Kingston, 1992; Polach and Widom, 1995; Li and Widom 2004 and more recently Wei et al., Nat Biotech 2018,

and the Dodonova et al. paper cited). However, this work provides evidence consistent for this model in an important developmental context. The specific binding of SOX2 to other sites when it is expressed at a lower level is much harder to understand. If co-factors direct SOX2 to those sites, they should also be present when SOX2 is expressed at a higher level. Co-occupancy of transcription factors at these regions in the absence of evidence for direct interaction between them and SOX2 cannot explain this finding: the authors should analyze these sequences more in detail, to determine if there are clear composite binding sites or defined spacings between motifs that would explain the increased affinity (as openness of chromatin does not appear to explain this, so co-binding to nearby sites cannot be the mechanism). One other possibility is that high SOX2 levels direct the lower-expressed transcription factors to new sites, depleting them.

* The occupancy of different peaks when SOX2 is expressed at lower levels is very surprising. More evidence is needed. The authors should rule out alternative explanations such as cross-reactivity of antibodies, and use spike-in and ENCODE IDR replicate metric or the like to call the ChIP-seq peaks to rule out the possibility that the bioinformatic pipeline calls weaker peaks when the signal overall is weaker.

* The analysis of the ChIP-seq data, and the effect of open chromatin on localization of SOX2 and TCF/LEF is qualitative. Very high numbers of peaks are reported, suggesting that a low threshold is used. A more quantitative analysis is needed, comparing strong features with weak ones.

* Co-occupancy analysis of the ChIP-seq peaks is not convincing. Most transcription factors ChIP-seq peaks cluster at a very high rate, irrespective of the biological roles of the factors. The authors should use control factors that do not have a biological role in development to assess whether the co-occupancy of TCF/LEF and SOX2 is higher than expected. The rate of co-occupancy could also be compared to pairs from the recent ENCODE study of >200 chromatin-associated proteins.

* Some statements are made that are consistent with the findings, but biochemically impossible without additional unknown effects. For example, p.10, the authors claim that low expression of SOX2 leads to occupancy of lower affinity sites, providing a "causal explanation". No causal explanation is provided here, as the converse is true – it is higher expression that leads to occupancy of lower affinity sites. The reason why lower affinity sites are occupied when SOX2 is expressed at a lower level is not mechanistically explained in this work. I do not think it is necessary for publication, but such clear errors in interpretation should be removed throughout."

Referee 2 notes:

"2/ The authors make strong statements from their ON/OFF ChIP-seq results regarding the conclusion that b-Catenin binding is dependent on Sox2. While I agree that the data strongly suggests this, I am not sure the experimental system allows to rigorously disentangle direct and indirect effects. The analyses are indeed made after 3 days of Dox treatment, which has already a strong impact on the identities of the differentiating cells. So how can we really disentangle a direct impact of Sox2 on b-Catenin recruitment from an indirect one due to other TFs differentially expressed in ON/OFF conditions? To address this in a more accurate way, the authors should use a fast degradation system of Sox2 to ascertain how fast b-Catenin is lost upon Sox2 depletion.

23/ The distinction between high affinity Sox sites, where Sox2 may act as a pioneer factor, and those with low affinity, where Sox2 likely behaves as a passenger of other drivers, is important. However, it requires better support, both by analysing more deeply their current data as well as (ideally) generating new data. On the one hand, it would be important to exploit their ATAC by splitting the fragments in small (<120bp) and nucleosome length (140-250bp) and making average profiles of cutsites or midpoints of the fragments, after centering the regions on either the Sox2 motif or the motif of the candidate driver (T, Cdx). This will more directly show the nucleosome/accessibility role of each factor at each cluster of regions. It would also be important to stratify the clusters by the Sox2 motif score of each peak, to ask the question: at Sox2/Cdx/T regions with good Sox2 motifs, does Sox2 also act as a passenger, with no impact on accessibility upon its loss, or rather as a potential pioneer? On the other hand, adding experimental data to test their model would be a huge improvement. Using a fast depletion strategy, as suggested above, would allow them to monitor the loss of accessibility and b-Cat and more directly support their idea that the accessibility created by Sox2 enables b-Cat recruitment. Also, it would seem required to prove their model right by studying the effects of ectopic T/Cdx expression in Sox2-ON cells: does a fraction of Sox2 and b-Cat redistribute their binding at T/Cdx targets with low affinity Sox2 motifs?"

(B) The characterisation of the SOX2-ON/OFF expression system should be further improved.

Referee 2 notes:

"1/ The expression of Sox2 in Sox2-OFF cells requires a more detailed and quantitative analysis. I understand its silencing is not full, and this allows the authors to compare high/low Sox2 rather than WT/KO. The impression I have from the data is that Sox2 is decreased 3-fold, which seems a small variation for a Tet-ON system. Moreover, the level detected in Sox2-ON cells also appears higher than in WT. So, it seems to me the authors are studying a hypermorph and a hypomorph. Unless I am incorrect, I do not think they take into consideration the hypermorphic character of Sox2-ON, which raises concerns for an accurate extrapolation of their results to real endogenous levels of Sox2. Finally, the Dox system leads to a strong heterogeneity in Sox2 expression, which can also make difficult to analyse the results in a simple high/low framework. To really address the impact of Sox2 levels in b-Cat binding, the authors should use a fluorescent-tagged protein approach to carefully sort cells displaying homogeneous levels of Sox2 matching those observed in WT cells."

Referee 3 notes:

"A significant concern of mine is that some important comparisons between groups involve multiple variables, which are not controlled for. The most concerning is central to the manuscript's findings in that the effects of Sox2 ON/OFF expression are insufficiently characterized during the first two days of EpiLC (Fig 1C-E). As a result, it is not clear if effects observed for Sox2 ON/OFF in CEpiLC are caused by the effects of Sox2 during the first two days or EpiLC or during the Gsk3-inhibition in the last day. The manuscript essentially concludes that there is an effect on Wnt-response, but this appears to be an assumption to me. For example, effects of Sox2 ON for cluster 3 could be because those CREs remained active through EpiLC process, not because Wnt/beta-catenin reactivates them. The

3manuscript could be improved with analyses of Sox2 ON/OFF EpiLC that would allow a direct comparison to Sox2 ON/OFF CEpiLC.”

(C) The text should be rewritten to better highlight the main advance and questions of general interest, aiming to make the manuscript accessible to a broader community.

Referee 1 notes:

“* In general, the manuscript needs a rewrite that highlights the more general points regarding levels of transcription factors, and their roles in development. The detail aimed at specialists could be moved to supplement and/or shortened considerably.”

Referee 3 notes:

“The question being examined has broad interest to communities of stem cell biologists, developmental biologists, and folks interested in how transcription factors work.”

(D) All other referee concerns pertaining to strengthening existing data, providing controls, methodological details, clarifications and textual changes should also be addressed.

(E) Finally please pay close attention to our guidelines on statistical and methodological reporting (listed below) as failure to do so may delay the reconsideration of the revised manuscript. In particular please provide:

We would be happy to consider a revised manuscript that would satisfactorily address these points, unless a similar paper is published elsewhere, or is accepted for publication in Nature Cell Biology in the meantime.

- ensure that it conforms to our format instructions and publication policies (see below and www.nature.com/nature/authors/).

4- provide a point-by-point rebuttal to the full referee reports verbatim, as provided at the end of this letter.

- provide the completed Editorial Policy Checklist (found here <https://www.nature.com/authors/policies/Policy.pdf>), and Reporting Summary (found here <https://www.nature.com/authors/policies/ReportingSummary.pdf>). This is essential for reconsideration of the manuscript and these documents will be available to editors and referees in the event of peer review. For more information see <http://www.nature.com/authors/policies/availability.html> or contact me.

[REDACTED]

We would like to receive a revised submission within six months. We would be happy to consider a revision even after this timeframe, however if the resubmission deadline is missed and the paper is eventually published, the submission date will be the date when the revised manuscript was received.

We hope that you will find our referees' comments, and editorial guidance helpful. Please do not hesitate to contact me if there is anything you would like to discuss.

With best wishes,

Christine.

Christine Weber, PhD
Senior Editor
Nature Cell Biology
E-mail: christine.weber@nature.com
Phone: +44 (0)207 843 4924

Reviewers' Comments:

Reviewer #1:

Remarks to the Author:

Blassberg and others study the role of the transcription factor SOX2 in coupling the Wnt-signalling

5pathway to different outcomes during development. They find that high levels of SOX2 in embryonic stem cells lead to opening of chromatin and occupancy of a set of target sites that are not bound in cells expressing low levels of SOX2. Surprisingly, they also find that intermediate/low levels of SOX2 leads to binding of sites that are not occupied when SOX2 is expressed at a high level.

This manuscript addresses two of the most important questions in the role of transcription factors in cell lineage determination: how are signalling pathways re-used for different purposes during development, and (to me more novel and interesting) what is the general role of transcription factor expression levels in lineage determination. The latter question has been extensively studied for the few factors that are involved in gradient morphogenesis, but the authors results here suggest a more general mechanism by which factor levels may encode developmental information, and how this quantitative signal would converted to discrete outputs by cells.

The manuscript describes a large amount of work, and the authors present compelling evidence that different levels of SOX2 reconfigure Wnt responses. However, the mechanism by which this happens is not clear (see below). The writing of the manuscript would also benefit from clearer formulation of the questions, and less detailed description of relationship of the findings to minor controversies in the stem cell field. Currently, the work is written more to specialists, even when it directly addresses major questions of general interest.

In summary, this is an important work that addresses key problems in developmental biology. I recommend publication of this work after the points below are addressed.

Major points:

- * The idea that due to a biochemical equilibrium, the "pioneering" activity of a transcription factor would be a function of its expression level has been proposed before (e.g. by Workman and Kingston, 1992; Polach and Widom, 1995; Li and Widom 2004 and more recently Wei et al., Nat Biotech 2018, and the Dodonova et al. paper cited). However, this work provides evidence consistent for this model in an important developmental context. The specific binding of SOX2 to other sites when it is expressed at a lower level is much harder to understand. If co-factors direct SOX2 to those sites, they should also be present when SOX2 is expressed at a higher level. Co-occupancy of transcription factors at these regions in the absence of evidence for direct interaction between them and SOX2 cannot explain this finding: the authors should analyze these sequences more in detail, to determine if there are clear composite binding sites or defined spacings between motifs that would explain the increased affinity (as openness of chromatin does not appear to explain this, so co-binding to nearby sites cannot be the mechanism). One other possibility is that high SOX2 levels direct the lower-expressed transcription factors to new sites, depleting them.
- * The occupancy of different peaks when SOX2 is expressed at lower levels is very surprising. More evidence is needed. The authors should rule out alternative explanations such as cross-reactivity of antibodies, and use spike-in and ENCODE IDR replicate metric or the like to call the ChIP-seq peaks to rule out the possibility that the bioinformatic pipeline calls weaker peaks when the signal overall is weaker.

6* The analysis of the ChIP-seq data, and the effect of open chromatin on localization of SOX2 and TCF/LEF is qualitative. Very high numbers of peaks are reported, suggesting that a low threshold is used. A more quantitative analysis is needed, comparing strong features with weak ones.

* Co-occupancy analysis of the ChIP-seq peaks is not convincing. Most transcription factors ChIP-seq peaks cluster at a very high rate, irrespective of the biological roles of the factors. The authors should use control factors that do not have a biological role in development to assess whether the co-occupancy of TCF/LEF and SOX2 is higher than expected. The rate of co-occupancy could also be compared to pairs from the recent ENCODE study of >200 chromatin-associated proteins.

* Some statements are made that are consistent with the findings, but biochemically impossible without additional unknown effects. For example, p.10, the authors claim that low expression of SOX2 leads to occupancy of lower affinity sites, providing a "causal explanation". No causal explanation is provided here, as the converse is true – it is higher expression that leads to occupancy of lower affinity sites. The reason why lower affinity sites are occupied when SOX2 is expressed at a lower level is not mechanistically explained in this work. I do not think it is necessary for publication, but such clear errors in interpretation should be removed throughout.

* In general, the manuscript needs a rewrite that highlights the more general points regarding levels of transcription factors, and their roles in development. The detail aimed at specialists could be moved to supplement and/or shortened considerably.

Reviewer #2:

Remarks to the Author:

In this manuscript, the Briscoe lab addresses how Sox2 binds the chromatin in order to mediate distinct responses by a single signalling pathway (WNT) and its known effectors (Tcfs and b-Catenin). Using a combination of genome-wide approaches in ES-differentiation derivatives where Sox2 expression is experimentally controlled, the authors suggest that the level of Sox2 impacts its binding site selection, differentially channelling WNT effectors binding to different sets of genes. The proposed model is that at high levels, Sox2 binding preferentially occurs at high affinity sites, where Sox2 renders the chromatin accessible to enable Tcf/b-Cat recruitment; at low levels, Sox2 binds at low affinity sites where it likely acts as a passenger of other TFs, such as T or Cdx, but nevertheless recruits Tcf/b-Cat. While the results are certainly interesting and the general question under study is very important for developmental and chromatin biology, I feel more appropriate experimental systems should be used to propose such a potentially important concept with a higher degree of confidence and certainty. My main concerns/suggestions are listed below.

1/ The expression of Sox2 in Sox2-OFF cells requires a more detailed and quantitative analysis. I understand its silencing is not full, and this allows the authors to compare high/low Sox2 rather than WT/KO. The impression I have from the data is that Sox2 is decreased 3-fold, which seems a small variation for a Tet-ON system. Moreover, the level detected in Sox2-ON cells also appears higher than in WT. So, it seems to me the authors are studying a hypermorph and a hypomorph. Unless I am

7incorrect, I do not think they take into consideration the hypermorphic character of Sox2-ON, which raises concerns for an accurate extrapolation of their results to real endogenous levels of Sox2. Finally, the Dox system leads to a strong heterogeneity in Sox2 expression, which can also make difficult to analyse the results in a simple high/low framework. To really address the impact of Sox2 levels in b-Cat binding, the authors should use a fluorescent-tagged protein approach to carefully sort cells displaying homogeneous levels of Sox2 matching those observed in WT cells.

2/ The authors make strong statements from their ON/OFF ChIP-seq results regarding the conclusion that b-Catenin binding is dependent on Sox2. While I agree that the data strongly suggests this, I am not sure the experimental system allows to rigorously disentangle direct and indirect effects. The analyses are indeed made after 3 days of Dox treatment, which has already a strong impact on the identities of the differentiating cells. So how can we really disentangle a direct impact of Sox2 on b-Catenin recruitment from an indirect one due to other TFs differentially expressed in ON/OFF conditions? To address this in a more accurate way, the authors should use a fast degradation system of Sox2 to ascertain how fast b-Catenin is lost upon Sox2 depletion.

3/ The distinction between high affinity Sox sites, where Sox2 may act as a pioneer factor, and those with low affinity, where Sox2 likely behaves as a passenger of other drivers, is important. However, it requires better support, both by analysing more deeply their current data as well as (ideally) generating new data. On the one hand, it would be important to exploit their ATAC by splitting the fragments in small (<120bp) and nucleosome length (140-250bp) and making average profiles of cutsites or midpoints of the fragments, after centering the regions on either the Sox2 motif or the motif of the candidate driver (T, Cdx). This will more directly show the nucleosome/accessibility role of each factor at each cluster of regions. It would also be important to stratify the clusters by the Sox2 motif score of each peak, to ask the question: at Sox2/Cdx/T regions with good Sox2 motifs, does Sox2 also act as a passenger, with no impact on accessibility upon its loss, or rather as a potential pioneer? On the other hand, adding experimental data to test their model would be a huge improvement. Using a fast depletion strategy, as suggested above, would allow them to monitor the loss of accessibility and b-Cat and more directly support their idea that the accessibility created by Sox2 enables b-Cat recruitment. Also, it would seem required to prove their model right by studying the effects of ectopic T/Cdx expression in Sox2-ON cells: does a fraction of Sox2 and b-Cat redistribute their binding at T/Cdx targets with low affinity Sox2 motifs?

Reviewer #3:

Remarks to the Author:

This manuscript from Blassberg et al examines roles for Sox2 in embryonic stem cells during early stages of cell specification from pluripotency. They develop and use a system of dox-inducible Sox2 expressing cells to measure effects of Sox2 levels on a variety of activities (Sox2 binding sites, Tcf/beta-catenin binding, gene expression, ATAC-seq accessibility, etc) at steps along conversion from naïve to EpiLC to caudal EpiLC to either paraxial mesoderm or spinal cord progenitors. These conversion steps can result in cells responding to Wnt signaling (or GSK3 inhibition, here) in significantly different ways. The manuscript shows how progressing through conversions with different levels of Sox2 affects responses to Gsk3 inhibition, and uses Sox2/Tcf/beta-catenin occupancy and

8gene expression data to develop explanations for the different responses.

The manuscript is very nicely written, and data and information are presented well (with few minor exceptions). The question being examined has broad interest to communities of stem cell biologists, developmental biologists, and folks interested in how transcription factors work. Mechanisms explaining how Wnt signaling can cause such different effects in similar cell types are particularly interesting. The manuscript details a significant body of work, communicating high quality genome-wide datasets using the controlled Sox2 ON/OFF cells. So, there are definitive strengths.

A significant concern of mine is that some important comparisons between groups involve multiple variables, which are not controlled for. The most concerning is central to the manuscript's findings in that the effects of Sox2 ON/OFF expression are insufficiently characterized during the first two days of EpiLC (Fig 1C-E). As a result, it is not clear if effects observed for Sox2 ON/OFF in CEpiLC are caused by the effects of Sox2 during the first two days or EpiLC or during the Gsk3-inhibition in the last day. The manuscript essentially concludes that there is an effect on Wnt-response, but this appears to be an assumption to me. For example, effects of Sox2 ON for cluster 3 could be because those CREs remained active through EpiLC process, not because Wnt/beta-catenin reactivates them. The manuscript could be improved with analyses of Sox2 ON/OFF EpiLC that would allow a direct comparison to Sox2 ON/OFF CEpiLC.

Additional comments:

-Line 199 – “Exiting the pluripotent state” should be changed to “exiting naïve pluripotency”

-Line 245 – Does high Sox2 actually “increase chromatin accessibility” or is it more appropriate to state that it “maintains chromatin accessibility” at the CRE.

-Paragraph starting Line 390 – The roles of Tcf/Lef factors in pluripotent cells and other cell types display both redundant and counter-active against one another. There are plenty of example of this in frogs, mice and humans. Tcf7l1 appears to function primarily (possibly exclusively) as a repressor in pluripotent cells.

-Figure 2 is missing a panel D.

READABILITY OF MANUSCRIPTS – Nature Cell Biology is read by cell biologists from diverse backgrounds, many of whom are not native English speakers. Authors should aim to communicate

9their findings clearly, explaining technical jargon that might be unfamiliar to non-specialists, and avoiding non-standard abbreviations. Titles and abstracts should concisely communicate the main findings of the study, and the background, rationale, results and conclusions should be clearly explained in the manuscript in a manner accessible to a broad cell biology audience. Nature Cell Biology uses British spelling.

REFERENCES – are limited to a total of 70 for Articles, Resources, Technical Reports; and 40 for Letters. This includes references in the main text and Methods combined. References must be numbered sequentially as they appear in the main text, tables and figure legends and Methods and must follow the precise style of Nature Cell Biology references. References only cited in the Methods

10should be numbered consecutively following the last reference cited in the main text. References only associated with Supplementary Information (e.g. in supplementary legends) do not count toward the total reference limit and do not need to be cited in numerical continuity with references in the main text. Only published papers can be cited, and each publication cited should be included in the numbered reference list, which should include the manuscript titles. Footnotes are not permitted.

Methods should be written concisely, but should contain all elements necessary to allow interpretation and replication of the results. As a guideline, Methods sections typically do not exceed 3,000 words. The Methods should be divided into subsections listing reagents and techniques. When citing previous methods, accurate references should be provided and any alterations should be noted. Information must be provided about: antibody dilutions, company names, catalogue numbers and clone numbers for monoclonal antibodies; sequences of RNAi and cDNA probes/primers or company names and catalogue numbers if reagents are commercial; cell line names, sources and information on cell line identity and authentication. Animal studies and experiments involving human subjects must be reported in detail, identifying the committees approving the protocols. For studies involving human subjects/samples, a statement must be included confirming that informed consent was obtained. Statistical analyses and information on the reproducibility of experimental results should be provided in a section titled "Statistics and Reproducibility".

All Nature Cell Biology manuscripts submitted on or after March 21 2016 must include a Data availability statement at the end of the Methods section. For Springer Nature policies on data availability see <http://www.nature.com/authors/policies/availability.html>; for more information on this particular policy see <http://www.nature.com/authors/policies/data/data-availability-statements-data-citations.pdf>. The Data availability statement should include:

- Accession codes for primary datasets (generated during the study under consideration and designated as "primary accessions") and secondary datasets (published datasets reanalysed during the study under consideration, designated as "referenced accessions"). For primary accessions data should be made public to coincide with publication of the manuscript. A list of data types for which submission to community-endorsed public repositories is mandated (including sequence, structure, microarray, deep sequencing data) can be found here <http://www.nature.com/authors/policies/availability.html#data>.
- Unique identifiers (accession codes, DOIs or other unique persistent identifier) and hyperlinks for datasets deposited in an approved repository, but for which data deposition is not mandated (see here for details <http://www.nature.com/sdata/data-policies/repositories>).
- At a minimum, please include a statement confirming that all relevant data are available from the authors, and/or are included with the manuscript (e.g. as source data or supplementary information), listing which data are included (e.g. by figure panels and data types) and mentioning any restrictions on availability.

- If a dataset has a Digital Object Identifier (DOI) as its unique identifier, we strongly encourage including this in the Reference list and citing the dataset in the Methods.

We recommend that you upload the step-by-step protocols used in this manuscript to the Protocol Exchange. More details can found at www.nature.com/protocolexchange/about.

All imaging data should be accompanied by scale bars, which should be defined in the legend. Cropped images of gels/blots are acceptable, but need to be accompanied by size markers, and to retain visible background signal within the linear range (i.e. should not be saturated). The boundaries of panels with low background have to be demarked with black lines. Splicing of panels should only be considered if unavoidable, and must be clearly marked on the figure, and noted in the legend with a statement on whether the samples were obtained and processed simultaneously. Quantitative comparisons between samples on different gels/blots are discouraged; if this is unavoidable, it should only be performed for samples derived from the same experiment with gels/blots were processed in parallel, which needs to be stated in the legend.

- We accept PowerPoint (.PPT) files if they are fully editable. However, please refrain from adding

12PowerPoint graphical effects to objects, as this results in them outputting poor quality raster art. Text used for PowerPoint figures should be Helvetica (preferred) or Arial.

Supplementary items should relate to a main text figure, wherever possible, and should be mentioned sequentially in the main manuscript, designated as Supplementary Figure, Table, Video, or Note, and numbered continuously (e.g. Supplementary Figure 1, Supplementary Figure 2, Supplementary Table

131, Supplementary Table 2 etc.).

The total number of Supplementary Figures (not including the “unprocessed scans” Supplementary Figure) should not exceed the number of main display items (figures and/or tables (see our Guide to Authors and March 2012 editorial <http://www.nature.com/ncb/authors/submit/index.html#suppinfo>; <http://www.nature.com/ncb/journal/v14/n3/index.html#ed>). No restrictions apply to Supplementary Tables or Videos, but we advise authors to be selective in including supplemental data.

GUIDELINES FOR EXPERIMENTAL AND STATISTICAL REPORTING

REPORTING REQUIREMENTS – To improve the quality of methods and statistics reporting in our papers we have recently revised the reporting checklist we introduced in 2013. We are now asking all life sciences authors to complete two items: an Editorial Policy Checklist (found here <https://www.nature.com/authors/policies/Policy.pdf>) that verifies compliance with all required editorial policies and a reporting summary (found here <https://www.nature.com/authors/policies/ReportingSummary.pdf>) that collects information on experimental design and reagents. These documents are available to referees to aid the evaluation of the manuscript. Please note that these forms are dynamic ‘smart pdfs’ and must therefore be downloaded and completed in Adobe Reader. We will then flatten them for ease of use by the reviewers. If you would like to reference the guidance text as you complete the template, please access these flattened versions at <http://www.nature.com/authors/policies/availability.html>.

STATISTICS – Wherever statistics have been derived the legend needs to provide the n number (i.e. the sample size used to derive statistics) as a precise value (not a range), and define what this value represents. Error bars need to be defined in the legends (e.g. SD, SEM) together with a measure of centre (e.g. mean, median). Box plots need to be defined in terms of minima, maxima, centre, and percentiles. Ranges are more appropriate than standard errors for small data sets. Wherever statistical significance has been derived, precise p values need to be provided and the statistical test used needs to be stated in the legend. Statistics such as error bars must not be derived from $n < 3$. For sample sizes of $n < 5$ please plot the individual data points rather than providing bar graphs. Deriving

14statistics from technical replicate samples, rather than biological replicates is strongly discouraged. Wherever statistical significance has been derived, precise p values need to be provided and the statistical test stated in the legend.

Author Rebuttal to Initial comments

(A) Mechanistic insights relating to SOX2 binding sites, expression levels and Tcf/beta-catenin binding should be further strengthened by providing additional experiments and causal links, as advised by referees 1 and 2.

Referee 1 notes:

"* The idea that due to a biochemical equilibrium, the "pioneering" activity of a transcription factor would be a function of its expression level has been proposed before (e.g. by Workman and Kingston, 1992; Polach and Widom, 1995; Li and Widom 2004 and more recently Wei et al., Nat Biotech 2018, and the Dodonova et al. paper cited).

We now cite these references in the Discussion.

However, this work provides evidence consistent for this model in an important developmental context. The specific binding of SOX2 to other sites when it is expressed at a lower level is much harder to understand. If co-factors direct SOX2 to those sites, they should also be present when SOX2 is expressed at a higher level. Co-

15occupancy of transcription factors at these regions in the absence of evidence for direct interaction between them and SOX2 cannot explain this finding: the authors should analyze these sequences more in detail, to determine if there are clear composite binding sites or defined spacings between motifs that would explain the increased affinity (as openness of chromatin does not appear to explain this, so co-binding to nearby sites cannot be the mechanism). One other possibility is that high SOX2 levels direct the lower-expressed transcription factors to new sites, depleting them.

We show that SOX2 specifically occupies Cluster 1 sites in CEpiLCs (in which Sox2 is expressed at moderate levels) and is depleted from these sites in cell types that either express higher or lower levels of SOX2. We agree with the reviewer that this result is surprising and we have investigated the mechanistic basis for it in some detail. We find that Cluster 1 sites are enriched for CDX motifs. Moreover, we show that CDX factors are bound to these sites in CEpiLCs.

In the previous version of the manuscript, these data led us to propose that CDX factors act as a cofactor for SOX2 binding in CEpiLCs. To test this, we have now performed new ChIPseq experiments in an ES cell line in which we have mutated all three CDX factors (CDX1, 2, 4). These data indicate that both SOX2 and Bcat occupancy are reduced at CDX bound Cluster 1 sites in the absence of CDX factors (Figs 6J,K). This provides independent evidence that CDX proteins act as co-factors to recruit SOX2 and Bcat to CEpiLC-specific CREs.

CDX factors are expressed in CEpiLCs, but not other cell types. Moreover, we demonstrate that CDX expression is reduced in CEpiLCs if SOX2 levels are either raised or depleted. This provides an explanation for why Cluster 1 sites are specifically occupied by SOX2 and Bcat in CEpiLC

In addition, we have identified a WNT-responsive CDX2 regulatory element containing SOX2 binding sites. These SOX2 binding sites are critical for its activity, providing an explanation for the loss of CDX2 expression in the absence of SOX2. We also show that the activity of this element is reduced in high SOX2 expressing pluripotent progenitors cultured in the presence of WNT pathway agonist compared to CEpiLCs. This is consistent with the absence of CDX expression in ES cells and pre-implantation epiblast. We therefore conclude that factors present in the pluripotent state repress the induction of CDX expression by WNT signalling and explain the absence of SOX2 binding to Cluster 1 sites in these cells.

T/BRA binding sites are enriched in Cluster 2 and Cluster 4. These sites are occupied by SOX2 and Bcat in CEpiLCs, which express T/BRA. We have now included new ChIPseq experiments carried out using a T/BRA mutant ES cell line. This indicated that in T/BRA mutant CEpiLCs, LEF/BCAT occupancy is reduced at sites in Cluster 2 and 4 (FigS6 F-K)

Although the data do not demonstrate whether or not SOX2 interacts directly with either CDX and/or T/BRA, taken together, the new ChIPseq datasets provide evidence to support the proposal that CDX and T/BRA are important specificity determinants in CEpiLCs. Determining the molecular basis for this will be subject of further studies.

* The occupancy of different peaks when SOX2 is expressed at lower levels is very surprising. More evidence is needed. The authors should rule out alternative explanations such as cross-reactivity of antibodies, and use spike-in and ENCODE IDR replicate metric or the like to call the ChIP-seq peaks to rule out the possibility that the bioinformatic pipeline calls weaker peaks when the signal overall is weaker.

We now provide evidence that the absence of CDX factors (in differentiated mutant ES cells) is sufficient to reduce SOX2 occupancy at CEpiLC-specific sites. In combination with the demonstration that the regulation of CDX expression is dependent on SOX2 levels, we believe that our model explains the surprising increase in SOX2 occupancy at selected sites in low SOX2 expressing CEpiLCs compared to high SOX2 expressing pluripotent ES cells. Importantly, WT CEpiLCs and CDX null CEpiLCs express equivalent levels of SOX2, and SOX2 occupancy is altered specifically at CDX bound Cluster 1 CREs. These results argue against the idea that the CEpiLC-specific elevated SOX2 occupancy at Cluster 1 sites is an artefact of either antibody cross-reactivity or the bioinformatic pipeline.

* The analysis of the ChIP-seq data, and the effect of open chromatin on localization of SOX2 and TCF/LEF is qualitative. Very high numbers of peaks are reported, suggesting that a low threshold is used. A more quantitative analysis is needed, comparing strong features with weak ones.

We agree that a large number of peaks are called from our data. We think this reflects the high signal-to-noise ratio in our data. We have compared our data with previous published SOX2 and Bcat ChIP-seq data from CEpiLCs¹. Compared to our dataset, we found substantially fewer peaks in the Koch et al. dataset with our analysis pipeline (we used the default MACS q-value parameter of 0.05 in both cases). When calling peaks from our datasets we

required that a peak be called in at least two out of three replicates from a given experimental condition, whereas this additional stringency criteria was not applied when calling peaks from the previously published data (which included only a single replicate for each ChIP). As our SOX2 and BCat peak sets include the majority of the peaks called by Koch et al. we believe that the differences in number of peaks called is the result of the improved signal-to-noise we achieved with our ChIPseq experiments, which we performed using an additional DSG fixation step in our protocol. We have added further details of our peak calling and consensus peak set creation workflow in the text of the Methods section:

p23- 'peak calling (MACS) (default q-value threshold < 0.05)'

p23- 'Inclusion of a peak in the consensus peak set required that it be called by MACS in a minimum of 2/3 biological replicates from any of the four experimental conditions (CEpiLC, SOX2-OFF, SOX2-ON, Naïve ES)'

We note that the total number of peaks that we reported for SOX2, BCAT and TCF/LEF are the consensus peak sets for the 4 biological conditions we investigated in this study (CEpiLC, ES, SOX2-OFF, SOX2-ON). When determined for a single representative experimental condition (Naïve ES cells) the number of consensus peaks for SOX2 and Bcat were reduced to 74398 and 31611 respectively. This is of similar order as the number of peaks called from ENCODE data by the ENCODE peak-calling pipeline (17123-24165.)

The differentially occupied SOX2 and TCF peaks that we report were determined quantitatively by DESeq2 analysis and therefore satisfied a statistical test. The DESeq p-adj statistic accounts for weaker features by requiring these to show a greater mean fold-change between conditions in order to be reported as significant. As these peaks also satisfy 1) the MACS peak calling q-value, and 2) are present in 2/3 replicates, we consider these differential peaks to be highly evidenced features.

* Co-occupancy analysis of the ChIP-seq peaks is not convincing. Most transcription factors ChIP-seq peaks cluster at a very high rate, irrespective of the biological roles of the factors. The authors should use control factors that do not have a biological role in development to assess whether the co-occupancy of TCF/LEF and SOX2 is higher than expected. The rate of co-occupancy could also be compared to pairs from the recent ENCODE study of >200 chromatin-associated proteins.

We have clarified our argument in the results section that differential Bcat occupancy between high and low SOX2 expressing cells is a consequence of differential SOX2 occupancy by considering only the overlap between differentially occupied peaks.

p5- 'This identified 11421 β -catenin peaks that were differentially occupied in response to experimental manipulation of SOX2 levels, of which 5228 (46%) overlapped with peaks differentially occupied by SOX2 (Fig 3D)'

This overlap is substantially higher than between differential Bcat peaks and ENCODE peaks. As suggested by the referee, we analysed ENCODE peaks for ChIP data of chromatin associated proteins and TFs in ES cells. Overlap between Bcat peaks differentially occupied in response to experimental manipulation of SOX2 levels with ENCODE factors ranged from 1.5-7% for ENCODE TFs and 6-8% for the general chromatin associated factors HCFC1 and CHD2. By contrast, 46% of differentially occupied Bcat peaks overlapped with differentially occupied SOX2 peaks. We have added these data to the manuscript in figure S3D.

p5- 'By contrast, differentially occupied β -catenin peaks showed a markedly lower association with chromatin-associated factors identified by ENCODE (1.5% -8.2% overlap) (Fig S3D)'.

When taken in combination with the correlated changes in occupancy between high and low SOX2 expressing cells, we view these data as strong evidence for a major role for SOX2 in driving the redistribution of Bcat. We also note that SOX2 differential peak clusters 2-6 are enriched for TCF/LEF motifs (Fig 6A), providing orthogonal evidence for their association with the TCF/LEF/Bcat ChIPseq peaks we report.

During the process of re-analysis we performed pairwise comparisons between ENCODE factors assayed in ES cells and found that co-occupancy of these factors ranges between 2%-29%, notably less than the 46% co-occupancy observed at differential SOX2/Bcat peaks. While we consider this analysis to further support our assertion that co-occupancy between SOX2 and Bcat is higher than would be expected by chance, we do not propose to add these data to the manuscript as it would distract from the main message of the study.

* Some statements are made that are consistent with the findings, but biochemically impossible without additional unknown effects. For example, p.10, the authors claim that low expression of SOX2 leads to occupancy of lower affinity sites, providing a "causal explanation". No causal explanation is provided here, as the converse is true – it is

higher expression that leads to occupancy of lower affinity sites. The reason why lower affinity sites are occupied when SOX2 is expressed at a lower level is not mechanistically explained in this work. I do not think it is necessary for publication, but such clear errors in interpretation should be removed throughout.”

We now provide further evidence, from ChIPseq experiments in CDX null ES cells, that CDX co-factors are necessary for SOX2 occupancy at low-affinity Cluster1 sites (Figs 6J,K). In the discussion on p.10 we state:

‘For SOX2, we found evidence of the involvement of CDX2 and T/BRA in directing binding to low affinity sites. Taken together, these observations suggest that SOX2 adopts different modes of chromatin interaction and CRE selection depending on its level of expression.’

We have amended the text on p12 to clarify our proposal that SOX2 occupancy at high affinity sites is mediated by SOX2 pioneering activity, whereas occupancy at low-affinity sites is mediated by cell-type specific cofactors such as CDX.

We have changed:

‘Our data provide evidence that SOX2 acts as a pioneer factor in pluripotent cells when expressed at high levels, but collaborates with other transcription factors to select lower affinity binding sites when expressed at lower levels. *This provides a causal explanation for how modulating TF expression levels can regulate distinct gene expression programmes.*’

to:

‘Our data provide evidence that SOX2 acts as a pioneer factor in pluripotent cells when expressed at high levels, but collaborates with other transcription factors *such as CDX* to select lower affinity binding sites when expressed at lower levels. *This is consistent with previous studies of how TFs gain access to their genomic targets and provides an explanation for the distinct gene expression programmes regulated at different TF expression levels.*’

Referee 2 notes:

"2/ The authors make strong statements from their ON/OFF ChIP-seq results regarding the conclusion that b-Catenin binding is dependent on Sox2. While I agree that the data strongly suggests this, I am not sure the experimental system allows to rigorously disentangle direct and indirect effects. The analyses are indeed made after 3 days of Dox treatment, which has already a strong impact on the identities of the differentiating cells. So how can we really disentangle a direct impact of Sox2 on b-Catenin recruitment from an indirect one due to other TFs differentially expressed in ON/OFF conditions? To address this in a more accurate way, the authors should use a fast degradation system of Sox2 to ascertain how fast b-Catenin is lost upon Sox2 depletion.

We agree that this would be an informative project. However, this would require developing and characterizing a new experimental model and repeating the majority of the study, we consider this to be beyond the reasonable scope of a revision.

It would also address a slightly different question: i.e. what is the effect of acutely depleting SOX2.

We have performed additional SOX2 ChIPseq analysis in EpiLCs, which express high levels of SOX2 in the absence of WNT signaling (see below), and find that chromatin accessibility at cluster 3 CREs is also correlated with elevated SOX2 occupancy at these sites, similar to high SOX2 expressing SOX2-ON and naïve ES cells. As high SOX2 levels promote chromatin accessibility at Cluster 3 sites across a range of cell types and WNT signaling conditions, we believe it reasonable to propose that high SOX2 is permissive for BCAT occupancy at these sites i.e p7: 'Thus, whereas changes in chromatin accessibility may explain the specific occupancy of SOX2/TCF/ β -catenin complexes at cluster 3 CREs...'

We have amended the text to clarify the statements regarding the dependence of BCAT on SOX2 binding in the following ways:

Results p.6, end of section 'SOX2 downregulation reconfigures β -CATENIN occupancy at cell-state specific CREs'

21We have changed:

'These data provide evidence of a profound genome-wide alteration in the binding site occupancy of β -catenin that is *dependent on SOX2*'.

to:

'These data therefore provide evidence of a profound and coordinated genome-wide alteration in the binding site occupancy of both SOX2 and β -catenin that is *dependent on the level of SOX2 expression*'.

On the second line of the discussion, p10, we have changed:

'We found that β -catenin frequently co-occupies genomic sites with SOX2 and that changes in the genomic location of SOX2 *led to* coordinated changes in chromatin binding of β -catenin'

to:

'We found that β -catenin frequently co-occupies genomic sites with SOX2. Perturbations to SOX2 levels led to coordinated changes in the genomic location of SOX2 and β -catenin binding.'

3/ The distinction between high affinity Sox sites, where Sox2 may act as a pioneer factor, and those with low affinity, where Sox2 likely behaves as a passenger of other drivers, is important. However, it requires better support, both by analysing more deeply their current data as well as (ideally) generating new data. On the one hand, it would be important to exploit their ATAC by splitting the fragments in small (<120bp) and nucleosome length (140-250bp) and making average profiles of cutsites or midpoints of the fragments, after centering the regions on either the Sox2 motif or the motif of the candidate driver (T, Cdx). This will more directly show the nucleosome/accessibility role of each factor at each cluster of regions. It would also be important to stratify the clusters by the Sox2 motif score of each peak, to ask the question: at Sox2/Cdx/T regions with good Sox2 motifs, does Sox2 also act as a passenger, with no impact on accessibility upon its loss, or rather as a potential pioneer? On the other hand, adding experimental data to test their model would be a huge improvement. Using a fast depletion strategy, as suggested above, would allow them to monitor the loss of accessibility and b-Cat and more directly support their idea that the accessibility created by Sox2 enables b-Cat recruitment. Also, it would seem required to prove their model right by studying the effects of ectopic T/Cdx

expression in Sox2-ON cells: does a fraction of Sox2 and b-Cat redistribute their binding at T/Cdx targets with low affinity Sox2 motifs?”

Instead of trying to infer mechanisms from further bioinformatic analysis we decided to test directly whether CDX and/or T/BRA act as cofactors in for SOX2 and/or BCat binding in CEpiLCs. To this end we performed ChIPseq assays for SOX2, BCat, LEF1 in mutant ES cells lacking either the three CDX factors or T/BRA. These new data showed that SOX2 occupancy at Cluster 1 sites is driven by the CEpiLC specific expression of CDX factors. While BCat occupancy is reduced at Cluster 2 and Cluster 4 sites in CEpiLC cells lacking T/BRA.

We have also reanalysed ATAC data previously generated in T/BRA and CDX mutant cells cultured under CEpiLC conditions. Consistent with the constitutive accessibility we observe at SOX2 peak clusters containing weak SOX2 motifs, ablation of either CDX or T/BRA has no effect on either chromatin accessibility or nucleosome density as computed by NucleoATAC.

As there is evidence that pluripotency factors such as Nanog repress the binding of factors such as SOX2 at certain sites², we consider the use of T/BRA and CDX mutant cells cultured under CEpiLC conditions to be the most appropriate experimental context in which to investigate their role in determining cell-type specific SOX2 occupancy. The additional SOX2, BCAT and LEF ChIPseq data from CDX and T/BRA mutant ES cells supports the idea that CDX is required for SOX2/BCAT/LEF1 occupancy at Cluster 1 sites. Moreover, T/BRA appears to influence the occupancy of LEF1 BCat at Cluster 2 and 4 sites. Together, the data support the idea that CDX and T/BRA act as cofactors for the binding of SOX2 and BCAT in cells expressing low levels of SOX2.

As suggested by the referee, we have also analysed the relationship between SOX2 and CDX motifs across the differentially occupied SOX2 peak sets and find that CDX binding motifs are enriched in proximity to a subset of low-affinity SOX2 motifs in CEpiLC-specific cluster 1 SOX2 peaks (Fig 6F). In combination with the genetic evidence of the specific dependence of SOX2 occupancy at cluster 1 CREs on CDX factor expression detailed above we propose that CDX factors promote the recruitment of SOX2 to constitutively accessible chromatin.

(B) The characterisation of the SOX2-ON/OFF expression system should be further improved.

Referee 2 notes:

23“1/ The expression of Sox2 in Sox2-OFF cells requires a more detailed and quantitative analysis. I understand its silencing is not full, and this allows the authors to compare high/low Sox2 rather than WT/KO. The impression I have from the data is that Sox2 is decreased 3-fold, which seems a small variation for a Tet-ON system. Moreover, the level detected in Sox2-ON cells also appears higher than in WT. So, it seems to me the authors are studying a hypermorph and a hypomorph. Unless I am incorrect, I do not think they take into consideration the hypermorphic character of Sox2-ON, which raises concerns for an accurate extrapolation of their results to real endogenous levels of Sox2. Finally, the Dox system leads to a strong heterogeneity in Sox2 expression, which can also make difficult to analyse the results in a simple high/low framework. To really address the impact of Sox2 levels in b-Cat binding, the authors should use a fluorescent-tagged protein approach to carefully sort cells displaying homogeneous levels of Sox2 matching those observed in WT cells.”

We have amended the misleading scales (which were incorrectly plotted in the initial submission) on flow cytometry plots in figures S1 E,G and throughout the other figures in the manuscript, and added additional data to better compare SOX2 levels between experimental conditions (Fig S1 N). Indeed, SOX2 levels are marginally higher in SOX2-OFF than in paraxial mesoderm progenitors which are considered to be SOX2 negative cells. We now also draw attention to the fact that the SOX2 orthologue SOX3 is expressed (Fig S1 K), and comment that this means we are dealing with a hypomorphic phenotype, as the reviewer points out. Sox3 has been shown previously to be sufficient to maintain EpiLC identity³ and we demonstrate with additional data (Fig S1 L,M) that it is sufficient to sustain the differentiation of SOX2-OFF from ES to EpiLC identity.

SOX2 levels in a proportion of SOX2-ON cells are somewhat higher than in ES cells, however the majority of SOX2-ON cells express SOX2 at similar levels to ES cells. Nonetheless, we do not make strong claims that SOX2-ON are a precise recapitulation of ES cells. Comparison between CEpiLC and ES cells, and between SOX2-ON and SOX2-OFF give the same results - a redistribution of SOX2 occupancy. This redistribution includes the counterintuitive increase in SOX2 occupancy at the same subset of sites in CEpiLC and SOX2-OFF cells, which both express low SOX2.

We agree that the high SOX2 expressing SOX2-ON cells are somewhat heterogeneous, however SOX2 expression in low SOX2 expressing SOX2-OFF is more homogeneous. We have better illustrated this point in Fig S1 N.

Referee 3 notes:

24“A significant concern of mine is that some important comparisons between groups involve multiple variables, which are not controlled for. The most concerning is central to the manuscript’s findings in that the effects of Sox2 ON/OFF expression are insufficiently characterized during the first two days of EpiLC (Fig 1C-E). As a result, it is not clear if effects observed for Sox2 ON/OFF in CEpiLC are caused by the effects of Sox2 during the first two days or EpiLC or during the Gsk3-inhibition in the last day. The manuscript essentially concludes that there is an effect on Wnt-response, but this appears to be an assumption to me. For example, effects of Sox2 ON for cluster 3 could be because those CREs remained active through EpiLC process, not because Wnt/beta-catenin reactivates them. The manuscript could be improved with analyses of Sox2 ON/OFF EpiLC that would allow a direct comparison to Sox2 ON/OFF CEpiLC.”

To strengthen the claim that chromatin accessibility at Cluster 3 sites requires high levels of SOX2 and is independent of GSK3b inhibition we now include additional SOX2 ChIPseq data from high SOX2 expressing EpiLCs (cultured in the absence of GSK3b) (Fig5 C,D; Fig S5 C). These data demonstrate that SOX2 occupancy at Cluster 3 sites is elevated in EpiLCs compared to CEpiLCs. Thus, SOX2 occupancy at Cluster 3 sites does not depend on GSK3b inhibition.

We agree that the data we present supports a model whereby SOX2 promotes accessibility at Cluster 3 sites in EpiLCs independently of WNT/b-catenin, and that this accessibility is lost in CEpiLCs due to the reduction of SOX2 levels in response to WNT signalling. In support of this we show that accessibility at Cluster 3 sites in EpiLCs is independent of Gsk3b inhibition and correlates with differential SOX2 occupancy at these sites between CEpiLCs and EpiLCs. We propose a model in which the expression of WNT responsive Cluster 3 genes differs between ES cells and CEpiLCs at least in part due to the different levels of SOX2 expressed by these cell types. We tested this hypothesis by maintaining SOX2 expression at high levels in CEpiLCs in which SOX2 levels are normally reduced in response to WNT signalling. By decoupling the reduction of SOX2 levels from the activation of WNT signalling in this way we showed that the ES cell pattern of SOX2/ Bcat co-occupancy and chromatin accessibility was maintained, demonstrating that the levels of SOX2, not WNT signalling, are responsible for the pattern of chromatin accessibility and SOX2/ Bcat binding.

(C) The text should be rewritten to better highlight the main advance and questions of general interest, aiming to make the manuscript accessible to a broader community.

Referee 1 notes:

25“* In general, the manuscript needs a rewrite that highlights the more general points regarding levels of transcription factors, and their roles in development. The detail aimed at specialists could be moved to supplement and/or shortened considerably.”

Referee 3 notes:

“The question being examined has broad interest to communities of stem cell biologists, developmental biologists, and folks interested in how transcription factors work.”

To address these comments from Referee 1 and Referee 3, we have extensively restructured and rewritten the Discussion and we have made editorial changes throughout the manuscript. We hope these make the work more accessible and highlight the general points.

Additional comments:

-Line 199 – “Exiting the pluripotent state” should be changed to “exiting naïve pluripotency”

Changed.

-Line 245 – Does high Sox2 actually “increase chromatin accessibility” or is it more appropriate to state that it “maintains chromatin accessibility” at the CRE.

Changed.

-Paragraph starting Line 390 – The roles of Tcf/Lef factors in pluripotent cells and other cell types display both redundant and counter-active against one another. There are plenty of example of this is frogs, mice and humans. Tcf711 appears to function primarily (possibly exclusively) as a repressor in pluripotent cells.

During the rewriting, we removed this paragraph.

-Figure 2 is missing a panel D.

Corrected

(D) All other referee concerns pertaining to strengthening existing data, providing controls, methodological details, clarifications and textual changes should also be addressed.

These have been addressed.

(E) Finally please pay close attention to our guidelines on statistical and methodological reporting (listed below) as failure to do so may delay the reconsideration of the revised manuscript. In particular please provide:

These are included.

We would be happy to consider a revised manuscript that would satisfactorily address these points, unless a similar paper is published elsewhere, or is accepted for publication in Nature Cell Biology in the meantime.

Additional references

1. Koch, F. *et al.* Antagonistic Activities of Sox2 and Brachyury Control the Fate Choice of Neuro-Mesodermal Progenitors. *Developmental Cell* **42**, 514-526.e7 (2017).

2. Heurtier, V. *et al.* The molecular logic of Nanog-induced self-renewal in mouse embryonic stem cells. *Nat Commun* **10**, 1109 (2019).
3. Corsinotti, A. *et al.* Distinct SoxB1 networks are required for naïve and primed pluripotency. *eLife* **6**, e27746 (2017).

Decision Letter, first revision:

Dear Dr Briscoe,

Your manuscript, "Sox2 levels configure the WNT response of epiblast progenitors responsible for vertebrate body formation", has now been seen by our original referees 1 and 2. We were unable to obtain comments by the original referee 3 despite our persistent efforts to chase the report. This has unfortunately extended the peer review process and we apologize for the delay. We have asked referee 2 to assess whether the revision has addressed the concerns by referee 3 and the related feedback is shown as extra comments in his/her report. As you will see from the referees' comments (attached below) referee 2 raised a few concerns that should be addressed before we can consider publication in Nature Cell Biology.

Nature Cell Biology editors discuss the referee reports in detail within the editorial team, including the chief editor, to identify key referee points that should be addressed with priority, and requests that are overruled as being beyond the scope of the current study. To guide the scope of the revisions, I have listed these points below. We are committed to providing a fair and constructive peer-review process, so please feel free to contact me if you would like to discuss any of the referee comments further.

In particular, it would be essential to:

a) acknowledge in the discussion that the experimental system is not sufficient to establish the direct link between Sox2 and the attributed effects, as noted by referee 2:

1/ the authors have tuned down claims of direct Sox2 activity with intelligent and subtle changes to the text. However, I think they should more directly explain in the discussion that the experimental system is not sufficient to establish a molecular, direct link, between Sox2 and the attributed effects. Pointing to the limitation of the study will be beneficial.

I have looked at the comments from Reviewer 3 and to the authors responses. My impression is that the reply is not directly addressing the key point of the dynamics of the events under study ; which is something I was also concerned with. The Reviewer was asking for a comparison of Sox2 ON/OFF in EpiLC but the authors only compared regular EpiLC to CEpiLC. True, they see in normal EpiLC (without

28Chiron) Sox2 binds robustly at cluster 3, but the real question, I guess, is whether in Sox2 OFF EpiLC the level of Sox2 binding is maintained or not ; which they do not address. Also, I am not sure I fully understand why they show so many regions binding Sox2 preferentially in CEpiLC compared to EpiLC (Fig5C) when in CEpiLC the binding of Sox2 is relatively minor (Fig3E). So, overall, I think this is a difficult decision to take. Both of our key points about the direct involvement of Sox2 are only indirectly and partially addressed, but the paper is overall of good quality. Perhaps tuning down the discussion and adding a paragraph to explain some limitations would be a solution.

b) add further bioinformatic analyses of the motifs/nucleosomes suggested by referee 2:

2/ I still believe that further bioinformatic analyses of the motifs/nucleosomes are required to strengthen the pioneering/passenger activity of Sox2:

2a/ NucleoATAC should be reanalysed by centering the regions on Sox2/Cdx/T motifs across clusters

2b/ better statistics of presence of "good" Sox2 motifs across clusters should be provided (ie how many regions do have at least one good motif) and regions lacking Sox2 motifs, particularly at cluster 3 commented and analysed on their own

2c/ using the motifs score rather than their p_value may be important, I would therefore ask the authors to reconsider performing an analysis of all the binding regions identified, ranking them by motif score for either Sox2/Cdx/T and assessing the correlation to binding/nucleosome properties in different cell types and upon genetic interventions as well as their cluster allocation. Observing the expected patterns would provide tremendous support to their claims.

c) All other referee concerns pertaining to clarifications and textual changes should also be addressed.

d) Finally please pay close attention to our guidelines on statistical and methodological reporting (listed below) as failure to do so may delay the reconsideration of the revised manuscript. In particular please provide:

We therefore invite you to take these points into account when revising the manuscript. In addition, when preparing the revision please:

- ensure that it conforms to our format instructions and publication policies (see below and www.nature.com/nature/authors/).

- provide a point-by-point rebuttal to the full referee reports verbatim, as provided at the end of this

29letter.

- provide the completed Editorial Policy Checklist (found here <https://www.nature.com/authors/policies/Policy.pdf>), and Reporting Summary (found here <https://www.nature.com/authors/policies/ReportingSummary.pdf>). This is essential for reconsideration of the manuscript and these documents will be available to editors and referees in the event of peer review. For more information see <http://www.nature.com/authors/policies/availability.html> or contact me.

Nature Cell Biology is committed to improving transparency in authorship. As part of our efforts in this direction, we are now requesting that all authors identified as 'corresponding author' on published papers create and link their Open Researcher and Contributor Identifier (ORCID) with their account on the Manuscript Tracking System (MTS), prior to acceptance. ORCID helps the scientific community achieve unambiguous attribution of all scholarly contributions. You can create and link your ORCID from the home page of the MTS by clicking on 'Modify my Springer Nature account'. For more information please visit www.springernature.com/orcid.

[REDACTED]

We would like to receive the revision within four weeks. If submitted within this time period, reconsideration of the revised manuscript will not be affected by related studies published elsewhere, or accepted for publication in Nature Cell Biology in the meantime. We would be happy to consider a revision even after this timeframe, but in that case we will consider the published literature at the time of resubmission when assessing the file.

We hope that you will find our referees' comments, and editorial guidance helpful. Please do not hesitate to contact me if there is anything you would like to discuss.

Best wishes,

Jie Wang

Jie Wang, PhD
Senior Editor
Nature Cell Biology

Tel: +44 (0) 207 843 4924

30email: jie.wang@nature.com

Reviewers' Comments:

Reviewer #1:

Remarks to the Author:

The authors have addressed all of my concerns satisfactorily, and the manuscript is now significantly improved.

I only have one minor suggestion that the authors would consider moving some details from the main figures to the supplement and would add a model figure to summarize the findings. This would help readers who are not familiar with the field to fully grasp the general significance of the work (both to gene regulation and to mechanisms of early development).

I recommend publication of this important work without further delay.

Reviewer #2:

Remarks to the Author:

The paper has been significantly improved by adding genetic evidence of the passenger role of Sox2 at sites with low affinity motifs. This is a key addition that allows, in my opinion, publication of the manuscript even though the direct effects of Sox2 are not unambiguously established. I would however suggest the authors to implement two modifications:

1/ the authors have tuned down claims of direct Sox2 activity with intelligent and subtle changes to the text. However, I think they should more directly explain in the discussion that the experimental system is not sufficient to establish a molecular, direct link, between Sox2 and the attributed effects. Pointing to the limitation of the study will be beneficial.

2/ I still believe that further bioinformatic analyses of the motifs/nucleosomes are required to strengthen the pioneering/passenger activity of Sox2:

2a/ NucleoATAC should be reanalysed by centering the regions on Sox2/Cdx/T motifs across clusters

2b/ better statistics of presence of "good" Sox2 motifs across clusters should be provided (ie how many regions do have at least one good motif) and regions lacking Sox2 motifs, particularly at cluster 3 commented and analysed on their own

2c/ using the motifs score rather than their p_value may be important, I would therefore ask the authors to reconsider performing an analysis of all the binding regions identified, ranking them by motif score for either Sox2/Cdx/T and assessing the correlation to binding/nucleosome properties in different cell types and upon genetic interventions as well as their cluster allocation. Observing the expected patterns would provide tremendous support to their claims.

Extra comment:

31I have looked at the comments from Reviewer 3 and to the authors responses. My impression is that the reply is not directly addressing the key point of the dynamics of the events under study ; which is something I was also concerned with. The Reviewer was asking for a comparison of Sox2 ON/OFF in EpiLC but the authors only compared regular EpiLC to CEpiLC. True, they see in normal EpiLC (without Chiron) Sox2 binds robustly at cluster 3, but the real question, I guess, is whether in Sox2 OFF EpiLC the level of Sox2 binding is maintained or not ; which they do not address. Also, I am not sure I fully understand why they show so many regions binding Sox2 preferentially in CEpiLC compared to EpiLC (Fig5C) when in CEpiLC the binding of Sox2 is relatively minor (Fig3E). So, overall, I think this is a difficult decision to take. Both of our key points about the direct involvement of Sox2 are only indirectly and partially addressed, but the paper is overall of good quality. Perhaps tuning down the discussion and adding a paragraph to explain some limitations would be a solution.

Reviewer #3:
None

GUIDELINES FOR SUBMISSION OF NATURE CELL BIOLOGY ARTICLES

ARTICLE FORMAT

32ABSTRACT – should not exceed 150 words and should be unreferenced. This paragraph is the most visible part of the paper and should briefly outline the background and rationale for the work, and accurately summarize the main results and conclusions. Key genes, proteins and organisms should be specified to ensure discoverability of the paper in online searches.

TEXT – the main text consists of the Introduction, Results, and Discussion sections and must not exceed 3500 words including the abstract. The Introduction should expand on the background relating to the work. The Results should be divided in subsections with subheadings, and should provide a concise and accurate description of the experimental findings. The Discussion should expand on the findings and their implications. All relevant primary literature should be cited, in particular when discussing the background and specific findings.

REFERENCES – are limited to a total of 70 in the main text and Methods combined,. They must be numbered sequentially as they appear in the main text, tables and figure legends and Methods and must follow the precise style of Nature Cell Biology references. References only cited in the Methods should be numbered consecutively following the last reference cited in the main text. References only associated with Supplementary Information (e.g. in supplementary legends) do not count toward the total reference limit and do not need to be cited in numerical continuity with references in the main text. Only published papers can be cited, and each publication cited should be included in the numbered reference list, which should include the manuscript titles. Footnotes are not permitted.

Methods should be written concisely, but should contain all elements necessary to allow interpretation and replication of the results. As a guideline, Methods sections typically do not exceed 3,000 words. The Methods should be divided into subsections listing reagents and techniques. When citing previous methods, accurate references should be provided and any alterations should be noted. Information must be provided about: antibody dilutions, company names, catalogue numbers and clone numbers

for monoclonal antibodies; sequences of RNAi and cDNA probes/primers or company names and catalogue numbers if reagents are commercial; cell line names, sources and information on cell line identity and authentication. Animal studies and experiments involving human subjects must be reported in detail, identifying the committees approving the protocols. For studies involving human subjects/samples, a statement must be included confirming that informed consent was obtained. Statistical analyses and information on the reproducibility of experimental results should be provided in a section titled "Statistics and Reproducibility".

All Nature Cell Biology manuscripts submitted on or after March 21 2016, must include a Data availability statement as a separate section after Methods but before references, under the heading "Data Availability". For Springer Nature policies on data availability see <http://www.nature.com/authors/policies/availability.html>; for more information on this particular policy see <http://www.nature.com/authors/policies/data/data-availability-statements-data-citations.pdf>. The Data availability statement should include:

- Accession codes for primary datasets (generated during the study under consideration and designated as "primary accessions") and secondary datasets (published datasets reanalysed during the study under consideration, designated as "referenced accessions"). For primary accessions data should be made public to coincide with publication of the manuscript. A list of data types for which submission to community-endorsed public repositories is mandated (including sequence, structure, microarray, deep sequencing data) can be found here <http://www.nature.com/authors/policies/availability.html#data>.
- Unique identifiers (accession codes, DOIs or other unique persistent identifier) and hyperlinks for datasets deposited in an approved repository, but for which data deposition is not mandated (see here for details <http://www.nature.com/sdata/data-policies/repositories>).
- At a minimum, please include a statement confirming that all relevant data are available from the authors, and/or are included with the manuscript (e.g. as source data or supplementary information), listing which data are included (e.g. by figure panels and data types) and mentioning any restrictions on availability.
- If a dataset has a Digital Object Identifier (DOI) as its unique identifier, we strongly encourage including this in the Reference list and citing the dataset in the Methods.

We recommend that you upload the step-by-step protocols used in this manuscript to the Protocol Exchange. More details can found at www.nature.com/protocolexchange/about.

DISPLAY ITEMS – main display items are limited to 6-8 main figures and/or main tables. For Supplementary Information see below.

FIGURES – Colour figure publication costs \$395 per colour figure. All panels of a multi-panel figure must be logically connected and arranged as they would appear in the final version. Unnecessary figures and figure panels should be avoided (e.g. data presented in small tables could be stated briefly

34in the text instead).

All imaging data should be accompanied by scale bars, which should be defined in the legend. Cropped images of gels/blots are acceptable, but need to be accompanied by size markers, and to retain visible background signal within the linear range (i.e. should not be saturated). The boundaries of panels with low background have to be demarked with black lines. Splicing of panels should only be considered if unavoidable, and must be clearly marked on the figure, and noted in the legend with a statement on whether the samples were obtained and processed simultaneously. Quantitative comparisons between samples on different gels/blots are discouraged; if this is unavoidable, it has to be performed for samples derived from the same experiment with gels/blots were processed in parallel, which needs to be stated in the legend.

Regardless of format, all figures must be vector graphic compatible files, not supplied in a flattened raster/bitmap graphics format, but should be fully editable, allowing us to highlight/copy/paste all text

35and move individual parts of the figures (i.e. arrows, lines, x and y axes, graphs, tick marks, scale bars etc). The only parts of the figure that should be in pixel raster/bitmap format are photographic images or 3D rendered graphics/complex technical illustrations.

Unprocessed scans of all key data generated through electrophoretic separation techniques need to be presented in a supplementary figure that should be labeled and numbered as the final supplementary figure, and should be mentioned in every relevant figure legend. This figure does not count towards the total number of figures and is the only figure that can be displayed over multiple pages, but should be provided as a single file, in PDF or TIFF format. Data in this figure can be displayed in a relatively informal style, but size markers and the figures panels corresponding to the presented data must be indicated.

The total number of Supplementary Figures (not including the “unprocessed scans” Supplementary Figure) should not exceed the number of main display items (figures and/or tables (see our Guide to Authors and March 2012 editorial <http://www.nature.com/ncb/authors/submit/index.html#suppinfo>; <http://www.nature.com/ncb/journal/v14/n3/index.html#ed>). No restrictions apply to Supplementary Tables or Videos, but we advise authors to be selective in including supplemental data.

GUIDELINES FOR EXPERIMENTAL AND STATISTICAL REPORTING

REPORTING REQUIREMENTS – To improve the quality of methods and statistics reporting in our papers we have recently revised the reporting checklist we introduced in 2013. We are now asking all life sciences authors to complete two items: an Editorial Policy Checklist (found here <https://www.nature.com/authors/policies/Policy.pdf>) that verifies compliance with all required editorial policies and a Reporting Summary (found here <https://www.nature.com/authors/policies/ReportingSummary.pdf>) that collects information on experimental design and reagents. These documents are available to referees to aid the evaluation of the manuscript. Please note that these forms are dynamic 'smart pdfs' and must therefore be downloaded and completed in Adobe Reader. We will then flatten them for ease of use by the reviewers. If you would like to reference the guidance text as you complete the template, please access these flattened versions at <http://www.nature.com/authors/policies/availability.html>.

37----- Please don't hesitate to contact NCB@nature.com should you have queries about any of the above requirements -----

Author Rebuttal, first revision:

Reviewer #1

I only have one minor suggestion that the authors would consider moving some details from the main figures to the supplement and would add a model figure to summarize the findings.

We have now included a model summarizing the findings as Figure 8

We have moved B,D,F from figure 2 and F, J-O from Figure 6 to the supplement

Reviewer #2

I think they should more directly explain in the discussion that the experimental system is not sufficient to establish a molecular, direct link, between Sox2 and the attributed effects.

We have added the following text to the Discussion to acknowledge that our work does not unambiguously establish a direct link between SOX2 and BCatenin recruitment/ nucleosome reorganization:

“Further experiments, using for example strategies to acutely deplete SOX2 (Nishimura et al., 2009), will be required to test whether SOX2 occupancy at pluripotency CREs directly drives the nucleosome reconfiguration accompanying the transition from pluripotency to CLE identity. Similar approaches could also be used to test whether SOX2 directly recruits BCatenin to CLE specific CREs, as has been shown previously at WNT-regulated pluripotency genes (Zhang et al., 2013).”

38NucleoATAC should be reanalysed by centering the regions on Sox2/Cdx/T motifs across clusters

As our claim is that SOX2 binding depletes nucleosome occupancy and that CDX/T do not we have performed this reanalysis by centering regions on SOX2 motifs. We have partitioned motifs into groups based on motif score: 1) Those which closely match the consensus motif without a mismatch; 2) Those with up to one mismatch; 3) Those with up to two mismatches. As our claim is that high SOX2 levels in SOX2 ON cells drive nucleosome depletion specifically at cluster 3 sites (but not at sites in other clusters, which we have shown to be constitutively accessible in other experimental conditions/ cell types in figure 5A,F), we have performed further analysis of the average NucleoATAC occupancy profile centred on motifs in cluster 3 peaks only. All of these analyses produce the same result as shown in Figure 5F (where regions are centered on SOX2 peak centers): nucleosome occupancy is depleted in specifically in SOX2 ON cells at motifs in cluster 3 peaks. Moreover, these analyses confirm that nucleosomes are more highly depleted in SOX2 ON at high scoring/high affinity SOX2 motifs within cluster 3 peaks (added as Figure S5G).

better statistics of presence of "good" Sox2 motifs across clusters should be provided (ie how many regions do have at least one good motif) and regions lacking Sox2 motifs, particularly at cluster 3 commented and analysed on their own

We have plotted the percentage of peaks within each cluster with at least one motif that satisfies the thresholds described above (added as Figure S5E). Cluster 3 and Cluster 6 (which both have high levels of SOX2 occupancy in SOX2 ON and Naïve ES cells) have the highest proportion of peaks with motifs meeting the 0 mismatch and 1 mismatch criteria. As described above, we have analysed those motifs without a mismatch separately, and we find them to exhibit the greatest average depletion of nucleosomes in SOX2 ON conditions compared to sets of motifs including 1 or 2 mismatches (Figure S5G). Moreover, we found that cluster 3 peaks on average harbour a greater number of motifs at each of the mismatch thresholds considered in the analysis (added as Figure S5F).

using the motifs score rather than their p_value may be important

We have re-plotted figure 5H using motif scores rather than p-values, and drawn attention to the discreet distribution of values, that represent motifs with either 0, 1, 2 or >2 mismatches.

Perform[] an analysis of all the binding regions identified, ranking them by motif score for either Sox2/Cdx/T and assessing the correlation to binding/nucleosome properties in different cell types and upon genetic interventions as well as their cluster allocation

As our analysis of motif frequencies indicated that cluster 3 SOX2 peaks are composed of a greater number of both high scoring and partially mismatched SOX2 binding motifs we ranked regions by the total score of all motifs within each region. This ranking was used to order heatmaps of nucleosome and SOX2 occupancy profiles across cluster 3 peaks from SOX2 ON cells. In support of our claims, those peaks with greatest total SOX2 motif score exhibited a clearer depletion of nucleosome occupancy and a greater intensity of SOX2 occupancy at SOX2 peak centers, whereas those with lowest total SOX2 motif scores exhibited a more diffuse signal for each measure (added as Figure S5H).

Extra comment:

Perhaps tuning down the discussion and adding a paragraph to explain some limitations would be a solution.

As indicated above, we have added the following text to the Discussion to acknowledge limitations and suggest further experiments:

“Further experiments, using for example strategies to acutely deplete SOX2 (Nishimura et al., 2009), will be required to test whether SOX2 occupancy at pluripotency CREs directly drives the nucleosome reconfiguration accompanying the transition from pluripotency to CLE identity. Similar approaches could

40also be used to test whether SOX2 directly recruits BCatenin to CLE specific CREs, as has been shown previously at WNT-regulated pluripotency genes (Zhang et al., 2013).”

Decision Letter, second revision:

20th December 2021

Dear Dr. Briscoe,

Thank you for submitting your revised manuscript, "Sox2 levels configure the WNT response of epiblast progenitors responsible for vertebrate body formation" (NCB-B44645B). It has been evaluated again by our original reviewer 2 and their comments are below. I am pleased to say that in light of their comments, we shall be happy, in principle, to publish it in Nature Cell Biology, pending minor revisions to comply with our editorial and formatting guidelines.

If the current version of your manuscript is in a PDF format, please email us a copy of the file in an editable format (Microsoft Word or LaTeX)-- we cannot proceed with PDFs at this stage.

Thank you again for your interest in Nature Cell Biology Please do not hesitate to contact me if you have any questions.

Sincerely,
Stelios

Stylianos Lefkopoulos, PhD
He/him/his
Associate Editor, Nature Cell Biology
Springer Nature
Heidelberger Platz 3, 14197 Berlin, Germany

E-mail: stylianos.lefkopoulos@springernature.com

Twitter: @s_lefkopoulos

41Reviewers' Comments:

Reviewer #2:

Remarks to the Author:

I thanks the authors for their high quality revisions. I have no further comment.

2nd February 2022

Dear Dr. Briscoe,

Thank you for your patience as we've prepared the guidelines for final submission of your Nature Cell Biology manuscript, "Sox2 levels configure the WNT response of epiblast progenitors responsible for vertebrate body formation" (NCB-B44645B). Please carefully follow the step-by-step instructions provided in the attached file, and add a response in each row of the table to indicate the changes that you have made. Please also check and comment on any additional marked-up edits we have proposed within the text. Ensuring that each point is addressed will help to ensure that your revised manuscript can be swiftly handed over to our production team.

We would like to start working on your revised paper, with all of the requested files and forms, as soon as possible (preferably within one week). Please get in contact with us if you anticipate delays.

In recognition of the time and expertise our reviewers provide to Nature Cell Biology's editorial process, we would like to formally acknowledge their contribution to the external peer review of your manuscript entitled "Sox2 levels configure the WNT response of epiblast progenitors responsible for vertebrate body formation". For those reviewers who give their assent, we will be publishing their names alongside the published article.

Nature Cell Biology offers a Transparent Peer Review option for new original research manuscripts submitted after December 1st, 2019. As part of this initiative, we encourage our authors to support increased transparency into the peer review process by agreeing to have the reviewer comments, author rebuttal letters, and editorial decision letters published as a Supplementary item. When you submit your final files please clearly state in your cover letter whether or not you would like to participate in this initiative. Please note that failure to state your preference will result in delays in accepting your manuscript for publication.

42Cover suggestions

As you prepare your final files we encourage you to consider whether you have any images or illustrations that may be appropriate for use on the cover of Nature Cell Biology.

Nature Cell Biology has now transitioned to a unified Rights Collection system which will allow our Author Services team to quickly and easily collect the rights and permissions required to publish your work. Approximately 10 days after your paper is formally accepted, you will receive an email in providing you with a link to complete the grant of rights. If your paper is eligible for Open Access, our Author Services team will also be in touch regarding any additional information that may be required to arrange payment for your article.

Please note that *Nature Cell Biology* is a Transformative Journal (TJ). Authors may publish their research with us through the traditional subscription access route or make their paper immediately open access through payment of an article-processing charge (APC). Authors will not be required to make a final decision about access to their article until it has been accepted. Find out more about Transformative Journals

Authors may need to take specific actions to achieve compliance with funder and institutional open access mandates. For submissions from January 2021, if your research is supported by a funder that requires immediate open access (e.g. according to Plan S principles) then you should select the gold OA route, and we will direct you to the compliant route where possible. For authors selecting the subscription publication route our standard licensing terms will need to be accepted, including our self-archiving policies. Those standard licensing terms will supersede any other terms that the author or any third party may assert apply to any version of the manuscript.

For information regarding our different publishing models please see our Transformative Journals page. If you have any questions about costs, Open Access requirements, or our legal forms,

43please contact ASJournals@springernature.com.

Please use the following link for uploading these materials:
[REDACTED]

Best regards,

Nyx Hills
Staff
Nature Cell Biology

On behalf of

Stylios Lefkopoulos
Editor
Nature Cell Biology

Reviewer #2:

Remarks to the Author:

I thank the authors for their high quality revisions. I have no further comment.

Final Decision Letter:

Dear Dr. Briscoe,

I am pleased to inform you that your manuscript, "Sox2 levels configure the WNT response of epiblast progenitors responsible for vertebrate body formation", has now been accepted for publication in Nature Cell Biology. Congratulations to you and your team!

Over the next few weeks, your paper will be copyedited to ensure that it conforms to Nature Cell Biology style. Once your paper is typeset, you will receive an email with a link to choose the

44appropriate publishing options for your paper and our Author Services team will be in touch regarding any additional information that may be required.

Please note that *Nature Cell Biology* is a Transformative Journal (TJ). Authors may publish their research with us through the traditional subscription access route or make their paper immediately open access through payment of an article-processing charge (APC). Authors will not be required to make a final decision about access to their article until it has been accepted. Find out more about Transformative Journals

If your paper includes color figures, please be aware that in order to help cover some of the additional

45cost of four-color reproduction, Nature Portfolio charges our authors a fee for the printing of their color figures. Please contact our offices for exact pricing and details.

If you have not already done so, we strongly recommend that you upload the step-by-step protocols used in this manuscript to the Protocol Exchange (www.nature.com/protocolexchange), an open online resource established by Nature Protocols that allows researchers to share their detailed experimental know-how. All uploaded protocols are made freely available, assigned DOIs for ease of citation and are fully searchable through nature.com. Protocols and Nature Portfolio journal papers in which they are used can be linked to one another, and this link is clearly and prominently visible in the online versions of both papers. Authors who performed the specific experiments can act as primary authors for the Protocol as they will be best placed to share the methodology details, but the Corresponding Author of the present research paper should be included as one of the authors. By uploading your Protocols to Protocol Exchange, you are enabling researchers to more readily reproduce or adapt the methodology you use, as well as increasing the visibility of your protocols and papers. You can also establish a dedicated page to collect your lab Protocols. Further information can be found at www.nature.com/protocolexchange/about

With kind regards,
Stelios

Stylianos Lefkopoulos, PhD
He/him/his
Associate Editor
Nature Cell Biology
Springer Nature
Heidelberger Platz 3, 14197 Berlin, Germany

E-mail: stylianos.lefkopoulos@springernature.com

Twitter: @s_lefkopoulos

** Visit the Springer Nature Editorial and Publishing website at www.springernature.com/editorial-

46nature portfolio

and-publishing-jobs for more information about our career opportunities. If you have any questions please click here.**